# NAADP-regulated two-pore channels drive phagocytosis through endo-lysosomal Ca²⁺ nanodomains, calcineurin and dynamin

Lianne C Davis[*] , Anthony J Morgan & Antony Galione[**]

## Abstract

Macrophages clear pathogens by phagocytosis and lysosomes that fuse with phagosomes are traditionally regarded as to a source of membranes and luminal degradative enzymes. Here, we reveal that endo-lysosomes act as platforms for a new phagocytic signalling pathway in which FcγR activation recruits the second messenger NAADP and thereby promotes the opening of $Ca^{2+}$-permeable two-pore channels (TPCs). Remarkably, phagocytosis is driven by these local endo-lysosomal $Ca^{2+}$ nanodomains rather than global cytoplasmic or ER $Ca^{2+}$ signals. Motile endo-lysosomes contact nascent phagosomes to promote phagocytosis, whereas endo-lysosome immobilization prevents it. We show that TPC-released $Ca^{2+}$ rapidly activates calcineurin, which in turn dephosphorylates and activates the GTPase dynamin-2. Finally, we find that different endo-lysosomal $Ca^{2+}$ channels play diverse roles, with TPCs providing a universal phagocytic signal for a wide range of particles and TRPML1 being only required for phagocytosis of large targets.

**Keywords** calcineurin; dynamin; lysosomes; NAADP; TPC
**Subject Categories** Membrane & Trafficking; Signal Transduction
**The EMBO Journal (2020) 39: e104058**

## Introduction

In professional phagocytic cells such as macrophages, phagocytosis is the process of the engulfment and internalization of pathogens and other large particles (> 0.5 μm) into vesicular phagosomes. Phagocytosis is a multi-step process that involves signal transduction pathways, rearrangements of the actin cytoskeleton and membrane trafficking (Flannagan *et al*, 2012).

During phagocytosis, endo-lysosomes have traditionally been assigned multiple roles as downstream effectors. For example, endo-lysosomes may provide additional membrane to the growing, nascent phagosome (Bajno *et al*, 2000; Czibener *et al*, 2006; Samie *et al*, 2013). Later, phagosomes mature by fusing with endo-lysosomes to transform the phagosome into an acidic and digestive phagolysosome where the enclosed pathogen is degraded (Flannagan *et al*, 2012). Moreover, lysosome upregulation via TFEB activation primes the bactericidal competency of macrophages (Gray *et al*, 2016).

As recently highlighted, a role for early $Ca^{2+}$ signals in phagocytosis remains controversial, with evidence for and against the requirement for $Ca^{2+}$ signals, as well as for and against various "classical" $Ca^{2+}$ signalling pathways ($Ca^{2+}$ influx, $Ca^{2+}$ release from the ER) (Westman *et al*, 2019). In view of this confusion, we wondered if a vital $Ca^{2+}$ signalling pathway had been overlooked: $Ca^{2+}$ can be released from "acidic $Ca^{2+}$-stores" (a spectrum of vesicles with an acidic, $Ca^{2+}$-containing lumen, including lysosomes, endosomes, lysosome-related organelles and secretory vesicles). A major pathway for endo-lysosomal $Ca^{2+}$ release is controlled by the second messenger nicotinic acid adenine dinucleotide phosphate (NAADP), which activates $Ca^{2+}$-permeable two-pore channels (TPCs) located on acidic organelles (Grimm *et al*, 2017; Patel & Kilpatrick, 2018; Galione, 2019), but its role during phagocytosis is currently not known. More broadly, endo-lysosomal $Ca^{2+}$ release is emerging as an important mechanism to combat pathogens, such as during phagosome enlargement and maturation (Samie *et al*, 2013; Dayam *et al*, 2015), mycobacterial clearance (Fineran *et al*, 2016), virus infection (Sakurai *et al*, 2015; Gunaratne *et al*, 2018) and efficient cell killing (Davis *et al*, 2012; Goodridge *et al*, 2019).

In this study, we promote endo-lysosomes from downstream effectors to upstream initiators of essential early signals that recruit a new phagocytosis signalling cascade. We show that it is not the global $Ca^{2+}$ signal, but local endo-lysosomal $Ca^{2+}$ nanodomains that drive phagocytosis; Fcγ receptor engagement triggers local $Ca^{2+}$ release from endo-lysosomes via the NAADP/TPC pathway which activates the $Ca^{2+}$-dependent phosphatase, calcineurin, that, in turn, activates the GTPase, dynamin-2, a protein essential for phagocytosis. Our data indicate that the endo-lysosomal NAADP/TPC pathway drives the phagocytosis and clearance of infective pathogens.

Department of Pharmacology, University of Oxford, Oxford, UK
*Corresponding author. Tel: +44 1865 271850; E-mail: lianne.davis@pharm.ox.ac.uk
**Corresponding author. Tel: +44 1865 271633; E-mail: antony.galione@pharm.ox.ac.uk

# Results

## Fcγ receptor-mediated phagocytosis requires local not global Ca²⁺ signals

Pathogens opsonized by IgG are recognized by the Fc-gamma receptors (FcγR) of macrophages and undergo phagocytosis, involving a rearrangement of the actin cytoskeleton and internalization of the microorganism (Flannagan *et al*, 2012). One of the first detectable signals in response to FcγR ligation is an increase in intracellular $Ca^{2+}$ ($[Ca^{2+}]_i$), but how the FcγR generates $Ca^{2+}$ signals remains incompletely understood and the role of $Ca^{2+}$ in phagocytosis has proven controversial (Westman *et al*, 2019). We therefore investigated $Ca^{2+}$ signalling and its role in phagocytosis in murine bone marrow-derived macrophages (BMDMs) upon FcγR stimulation either during "frustrated phagocytosis" or by phagocytosis of IgG-opsonized beads.

Frustrated phagocytosis can be induced by dropping BMDMs onto IgG-coated coverslips, engaging Fcγ receptors as they come into contact though they are unable to engulf the target. We measured single-cell $[Ca^{2+}]_i$ using the ratiometric $Ca^{2+}$ reporter, fura-2, and investigated the effect of buffering $Ca^{2+}$ sources with the $Ca^{2+}$ chelators, EGTA or BAPTA ($Ca^{2+}$ $K_d$ of 151 and 160 nM, respectively): extracellular $Ca^{2+}$ ($Ca^{2+}_o$) was chelated with EGTA whereas cytosolic $Ca^{2+}$ was buffered with EGTA (or BAPTA) preloaded as their AM-ester precursors.

Contact of BMDMs with the coverslip evoked a rapid, multiphasic $Ca^{2+}$ signal (Fig 1A) but only when the glass was coated with IgG (Δ fura-2 ratio: without IgG 0.04 ± 0.014, $n = 55$; with IgG 0.52 ± 0.02, $n = 46$). Chelation of $Ca^{2+}_o$ with EGTA, to prevent $Ca^{2+}$ influx, reduced but did not abolish the FcγR-induced $Ca^{2+}$ signal in BMDMs (Fig 1A). The residual $Ca^{2+}$ transient, by definition from intracellular $Ca^{2+}$ stores, was suppressed by loading the cytosol with either EGTA/AM or BAPTA/AM (Fig 1A).

Similar results were observed by dropping IgG-opsonized beads onto BMDM (Fig 1B) to evoke $Ca^{2+}$ signals partly generated by release from intracellular stores since they were present when $Ca^{2+}_o$ was removed (Fig 1B). When cells were treated with EGTA/AM or BAPTA/AM, all $Ca^{2+}$ signals were inhibited (Fig 1B). Overall, independent of the method of FcγR engagement, $Ca^{2+}$ signals evoked are composed of both $Ca^{2+}$ release from intracellular $Ca^{2+}$ stores and $Ca^{2+}$ influx across the plasma membrane.

Parallel experiments assessed the consequences of these $Ca^{2+}$-buffering protocols upon bead phagocytosis. In Fig 1E, phagocytosis was determined by counting phagocytosed IgG-fluorescent beads per cell and discarding from the count the external beads labelled with fluorescent anti-IgG (which appear yellow as in Fig 1D, or as a red ring as in Fig 1E, depending on the confocal optical section). Although removal of extracellular $Ca^{2+}$ substantially reduced $Ca^{2+}$ signalling (Fig 1A and B), it had no discernible effect on the ingestion of opsonized beads (Fig 1E), indicating that $Ca^{2+}$ influx is not important for phagocytosis. In contrast, suppression of intracellular $Ca^{2+}$ signals (Fig 1A and B) with the fast $Ca^{2+}$ buffer, BAPTA (AM-loaded), significantly reduced phagocytosis (Fig 1E). As a control for possible off-site effects of BAPTA, BAPTA-FF was used which chelates $Ca^{2+}$ with ~ 400-fold lower affinity ($K_d$ 65 μM) and consequently did not affect phagocytosis; this suggests that BAPTA inhibits by chelating $Ca^{2+}$ (Fig 1E). Remarkably, the slow $Ca^{2+}$ buffer, EGTA (AM-loaded), had no effect upon phagocytosis (Fig 1E) in spite of the suppression of the global cytosolic $Ca^{2+}$ signal (Fig 1A and B). The differential effect of slow and fast $Ca^{2+}$ buffers is diagnostic for processes that depend on *localized* elevations in $[Ca^{2+}]_i$ (i.e. $Ca^{2+}$ micro/nanodomains) because these are suppressed by fast BAPTA, but not slow EGTA (Fig 1C) (Kidd *et al*, 1999). In summary, we confirm that $Ca^{2+}$ is essential for FcγR phagocytosis but the way $Ca^{2+}$ is delivered is crucial: $Ca^{2+}$ influx is not required, whereas local $Ca^{2+}$ release from intracellular stores is essential.

## Endo-lysosomal Ca²⁺ stores drive phagocytosis

We then systematically probed which intracellular $Ca^{2+}$ stores are involved in driving phagocytosis by selectively depleting each store. The largest intracellular $Ca^{2+}$ store is the ER, and this usually underpins large global $Ca^{2+}$ signals. To eliminate the ER $Ca^{2+}$ store, we used ER $Ca^{2+}$-pump inhibitors, thapsigargin or cyclopiazonic acid (CPA), to empty the ER. Experiments were performed in $Ca^{2+}$-free medium (supplemented with 100 μM EGTA) to eliminate consequent capacitative $Ca^{2+}$ entry (although this does not affect phagocytosis, Fig 1E). We first verified that thapsigargin or CPA pre-treatment emptied the ER: in control cells, ionomycin mobilized the ER to give a robust transient $Ca^{2+}$ response; in contrast, after thapsigargin or CPA treatment, the ionomycin response is all but eliminated (Fig 1F) indicating that ER stores had been emptied.

---

**Figure 1. FcγR-mediated phagocytosis requires local not global Ca²⁺ signals.**

A, B  $[Ca^{2+}]_i$ changes in individual BMDMs were measured (A) using fura-2 during frustrated phagocytosis ($n = 46$–113) and (B) by dropping beads onto adherent cells loaded with Calbryte 520 ($n = 64$–159), in 1.8 mM extracellular $Ca^{2+}$ ($+Ca^{2+}_o$) or $Ca^{2+}$-free medium containing 100 μM EGTA ($-Ca^{2+}_o$) with or without prior loading with 25 μM EGTA/AM or 25 μM BAPTA/AM. The maximum fura-2 ratio changes (Δ 350/380) or fluorescence changes normalized to initial fluorescence (Δ$F/F_0$) are plotted.

C  Schematic depicting the buffering ranges of BAPTA (local $Ca^{2+}$) and EGTA (global $Ca^{2+}$).

D  FcγR-mediated phagocytosis was monitored using beads conjugated to Alexa Fluor 488 (in green). External beads were differentially labelled with anti-IgG (in red) and distinguished by their dual labelling (in yellow). Cytochalasin D (10 μM, Cyt D) prevented bead internalization.

E  Bead uptake in 15 min, under conditions as in (A, B), $n = 80$–91 cells with representative images of individual cells delineated by a dotted line.

F–K  Experiments conducted in $-Ca^{2+}_o$: (F) 1 μM thapsigargin (Tg) or 10 μM CPA pre-treatment, all but eliminated the 1 μM ionomycin (iono) response; single-cell $Ca^{2+}$ traces and mean Δ$F/F_0$, $n = 450$–603 cells. (G) FcγR activation (as in B) evoked $Ca^{2+}$ oscillations that were blocked with Tg or CPA treatment, whilst bead phagocytosis was not affected (H), $n = 312$–428 cells. (I) 1 μM bafilomycin A1 (Baf) for 3 h, or $Ca^{2+}$-binding dextran (Cal520-dx) and non-$Ca^{2+}$-binding dextrans (FITC-dx and Texas red-dx) trafficked to lysosomes, minimally affected global $Ca^{2+}$ signals evoked by FcγR activation (J); in parallel, bead uptake was only reduced by bafilomycin A1 or by chelating $Ca^{2+}$ within the lysosome with Cal520-dx (K), $n = 100$–162 cells. All cells are WT BMDM, and all beads are 3-μm-IgG-opsonized.

Data information: Scale bars = 5 μm (D, G, I) or 10 μm (E). Bar charts presented as the mean ± S.E.M. and probability determined using one-way ANOVA (*$P < 0.05$, ***$P < 0.001$). See also Appendix Fig S1.

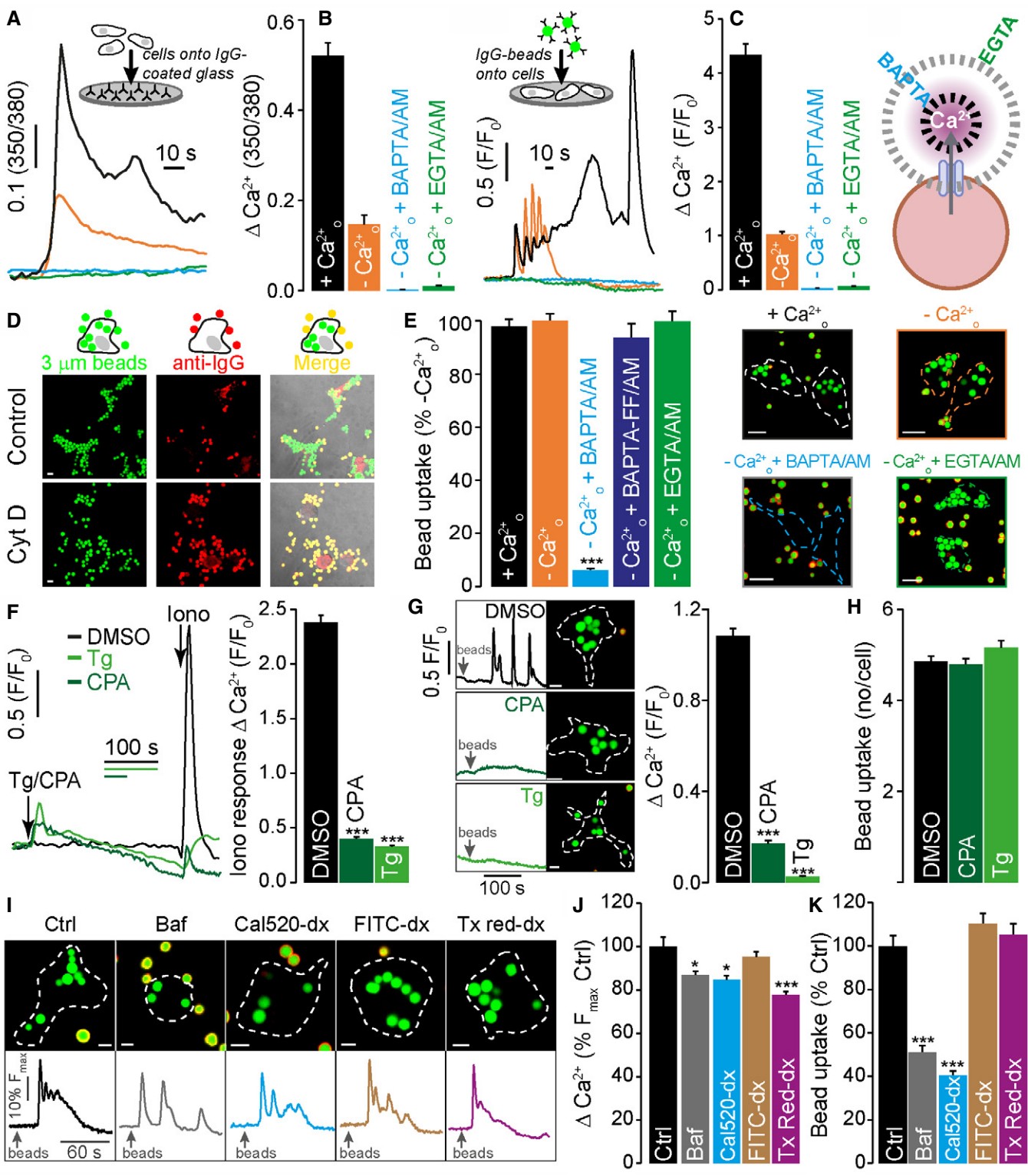

**Figure 1.**

Turning to phagocytosis $Ca^{2+}$ signals, $Fc\gamma R$ activation evoked $Ca^{2+}$ oscillations in DMSO-treated cells, whereas pre-treatment with thapsigargin or CPA almost completely abolished cytosolic $Ca^{2+}$ signals (Fig 1G). This indicates that, as expected, the detectable global $Ca^{2+}$ signals are predominantly derived from the vast ER reservoir. We then tested the effect of such ER-store depletion on phagocytosis. Unexpectedly, neither thapsigargin nor CPA had any impact upon bead uptake (Fig 1H). This indicates that, in spite of its

global magnitude, ER $Ca^{2+}$ release has no role in driving the phagocytosis of (small, 3 μm) particles.

Given that endo-lysosomes are emerging as physically small but functionally important $Ca^{2+}$ stores, we tested for their involvement in phagocytosis. Because the pathways of endo-lysosomal $Ca^{2+}$ filling are unclear (but may operate as $Ca^{2+}/H^+$ exchange) (Morgan *et al*, 2011; Yang *et al*, 2019), there are no drugs that inhibit this $Ca^{2+}$ transporter for directly depleting the vesicle $Ca^{2+}$ content. We use two indirect strategies to lower endo-lysosomal $Ca^{2+}$ storage. First, we chelated the $Ca^{2+}$ content within the vesicle lumen by allowing cells to endocytose dextran-conjugated BAPTA-based $Ca^{2+}$-binding reporters (Lloyd-Evans *et al*, 2008). We used Cal520-dextran ($K_d$ of 320 nM at neutral pH), whereas non-$Ca^{2+}$-binding dextrans FITC or Texas Red were used as controls; all these dextrans traffic to endo-lysosomes (Appendix Fig S1). Endo-lysosomal swelling was only observed in Cal520-Dextran-loaded cells (Appendix Fig S1), consistent with a lysosomal storage phenotype caused by reduced luminal $Ca^{2+}$ (Lloyd-Evans *et al*, 2008). Loading macrophages with dextrans had only a minor effect upon the global (ER) $Ca^{2+}$ signals (Fig 1I and J), and this did not correlate with $Ca^{2+}$-binding ability of the dextrans (there was a small, non-specific effect with Texas Red). In contrast, bead uptake was substantially reduced by the $Ca^{2+}$-binding Cal520-dextran, whereas neither of the non-$Ca^{2+}$-binding dextrans affected phagocytosis (Fig 1I and K).

Complementary to this, bafilomycin A1 (a V-type $H^+$-ATPase inhibitor) can indirectly drain $Ca^{2+}$ from acidic organelles by collapsing the $H^+$ gradient (Morgan *et al*, 2011; Yang *et al*, 2019). Just as with the Cal520-Dx, bafilomycin A1 modestly inhibited $Ca^{2+}$ signals, but had a dramatic impact on bead uptake (Fig 1I–K). Together, the two different strategies provide the first evidence that these small acidic $Ca^{2+}$ stores are important for phagocytosis and we investigated this further.

### NAADP-mediated $Ca^{2+}$ signalling is essential for efficient FcγR-dependent phagocytosis

In examining phagocytosis, past studies have focused on intracellular $Ca^{2+}$ release from the ER and the resulting store-operated $Ca^{2+}$ entry (Westman *et al*, 2019). Since lysosomal $Ca^{2+}$ is important for FcγR phagocytosis (Fig 1I and K), we hypothesized that the NAADP/TPC pathway contributes to FcγR-mediated phagocytosis. NAADP is a second messenger releasing $Ca^{2+}$ from endo-lysosomes, often creating local $Ca^{2+}$ domains (Zhu *et al*, 2010; Morgan *et al*, 2013; Grimm *et al*, 2014; Lin-Moshier *et al*, 2014; Hockey *et al*, 2015) that could explain the differential effects of EGTA and BAPTA upon phagocytosis (Fig 1E).

First, to confirm that BMDMs possess the NAADP-induced $Ca^{2+}$-release pathway, we bath-applied NAADP as its cell-permeant ester form, NAADP/AM and observed robust $Ca^{2+}$ signals which were inhibited by the selective NAADP antagonist Ned-19, or by bafilomycin A1 (Appendix Fig S2A and (Ruas *et al*, 2015a)). This is a hallmark of NAADP-mediated $Ca^{2+}$ release from endo-lysosomes. Turning to FcγR-mediated $Ca^{2+}$ signals, Ned-19 inhibited the $Ca^{2+}$ responses to frustrated phagocytosis (Fig 2A), suggesting that NAADP-induced $Ca^{2+}$ release partly contributes to the global $Ca^{2+}$ signal induced by FcγR stimulation. The corollary of this reduced $Ca^{2+}$ was reduced phagocytosis: internalization of opsonized beads by BMDMs was significantly reduced by Ned-19 compared to DMSO

controls (Fig 2B), suggesting that NAADP-evoked $Ca^{2+}$ signalling contributes to phagocytosis. Together, these data suggest that NAADP-mediated $Ca^{2+}$ signals can drive efficient FcγR-dependent phagocytosis in BMDM.

### Phagocytosis is reduced in TPC knockouts

NAADP acts by gating TPCs, a family of $Ca^{2+}$-permeable cation channels in endo-lysosomes (Grimm *et al*, 2017; Patel & Kilpatrick, 2018; Galione, 2019). We reasoned that as NAADP is important for phagocytosis, its molecular target(s) (TPCs) should also be essential, and TPCs are expressed in phagocytes (Freeman & Grinstein, 2018) and in BMDMs in particular (Ruas *et al*, 2015a). Therefore, various aspects of phagocytosis were monitored in BMDMs from $Tpcn1^{-/-}$ (TPC1-KO), $Tpcn2^{-/-}$ (TPC2-KO), or $Tpcn1/2^{-/-}$ (DKO) mouse lines (Ruas *et al*, 2015b). We have previously shown that TPCs are required for NAADP-induced $Ca^{2+}$ signalling because BMDMs lacking TPCs fail to evoke $Ca^{2+}$ signals in response to NAADP (Ruas *et al*, 2015a).

We first tested whether TPCs are important for the $Ca^{2+}$ signal upon FcγR stimulation during frustrated phagocytosis. Compared to WT BMDM, TPC1- or TPC2-KO BMDMs exhibited substantially reduced $Ca^{2+}$ signals following FcγR engagement, and the degree of inhibition was not any greater in TPC-DKO cells (Fig 2A), confirming that the residual component was TPC-independent. Consistent with this, Ned-19 inhibited the $Ca^{2+}$ response by a similar amount (~ 60%; Fig 2A). In contrast, the knockout of a different lysosomal $Ca^{2+}$ channel, TRPML1, had only a much smaller (~ 20%) effect upon FcγR-mediated $Ca^{2+}$ signals implying that TPCs play a greater role than does TRPML1 in FcγR signal transduction (Fig 2A). Our data suggest for the first time that NAADP-induced $Ca^{2+}$ mobilization from acidic stores via TPCs is a component of the essential $Ca^{2+}$ signal during phagocytosis.

Turning to the phagocytosis of beads, TPC deletion severely inhibited FcγR-mediated phagocytosis at all times after the bead addition (Fig 2C and D). Similarly, the mouse macrophage cell line, RAW 264.7, treated with Ned-19 was also unable to phagocytose opsonized beads normally (Appendix Fig S2C). Importantly, the loss of TPCs does not affect the binding of opsonized beads to the plasma membrane, indicating a post-attachment effect (Appendix Fig S2B).

We evaluated the specificity of lysosomal $Ca^{2+}$ release channels by comparing the role of TPCs with TRPML1. TRPML1 has been implicated in the phagocytosis of large particles (3.9 or 6 μm), but not smaller particles (2.6 or 3 μm) (Samie *et al*, 2013; Dayam *et al*, 2015). Using either 3- or 6-μm opsonized beads, we confirmed that only the phagocytosis of large beads (and not small beads) was dependent on TRPML1 as judged by TRPML1-knockout BMDMs (Fig 2E and F). This contrasted with single TPC-KO macrophages that showed defective phagocytosis of all bead sizes, whether small (Fig 2C and H) or large (Fig 2D). As with $Ca^{2+}$, there was no further inhibition in the TPC-DKO (Fig 2C and D). Taken together, these findings demonstrate that TPCs are critical for phagocytosis over a broad range of particle sizes, whereas TRPML1 knockout only affects phagocytosis of large beads.

### Heterologous expression of TPC1 and TPC2 restored phagocytosis

To confirm that the phagocytic defect in TPC-KO BMDMs is attributable to the loss of TPC expression, we aimed to rescue phagocytosis

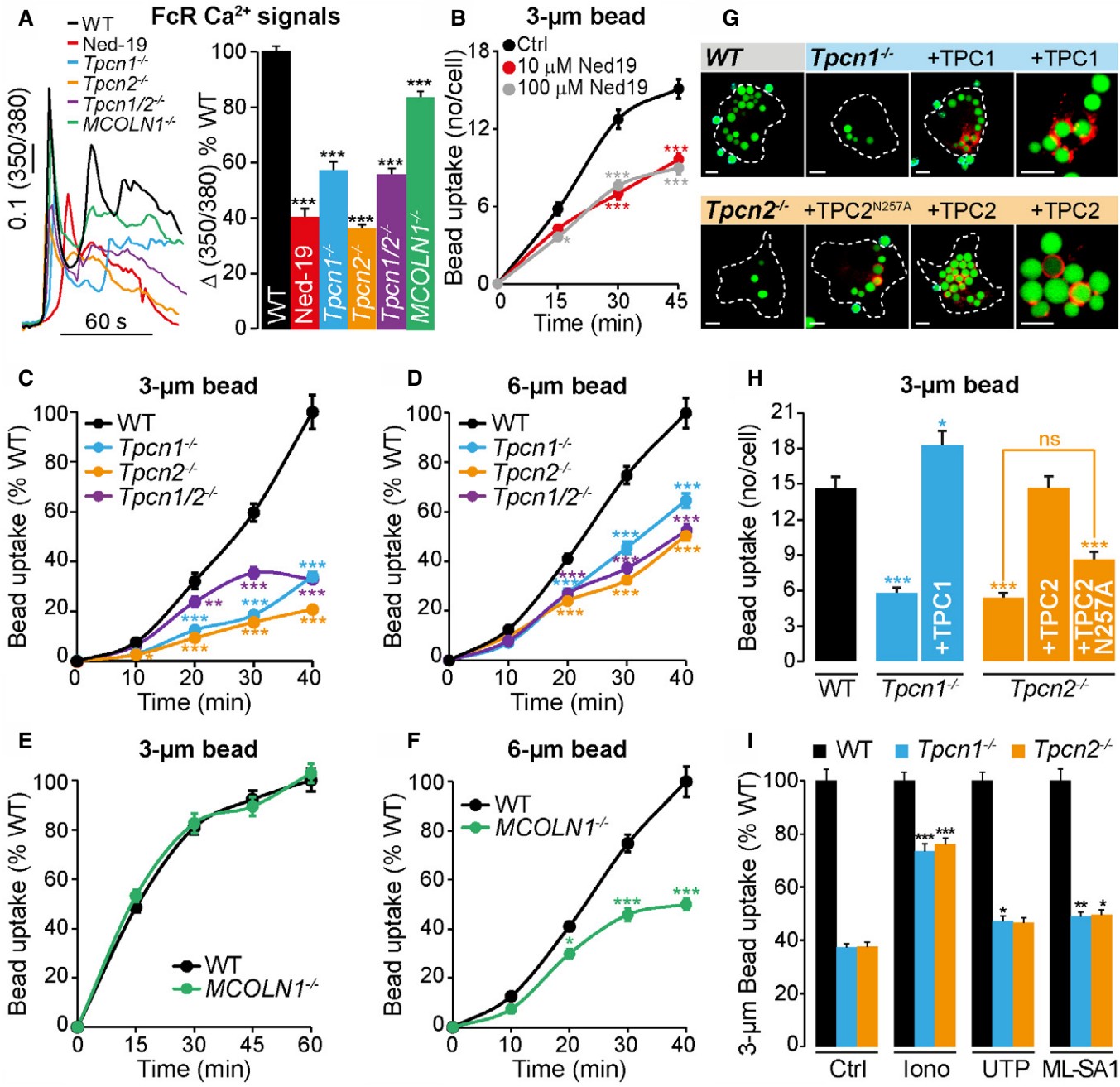

**Figure 2. NAADP and TPCs are essential for efficient FcγR-dependent Ca²⁺ signals and phagocytosis.**

A  The Ca²⁺ signal elicited by frustrated phagocytosis was significantly reduced in Ned-19 (10 μM)-treated WT and in all TPC-KO BMDMs; maximum Ca²⁺ amplitude, expressed as a percentage of WT control ($n$ = 314–641 cells).

B–D  Phagocytosis of opsonized 3-μm (B, C) and 6-μm (D) beads was inhibited by Ned-19 (B) and in BMDMs from $Tpcn^{-/-}$ mice compared to WT ($n$ = 134–280 cells).

E, F  (E) Phagocytosis of 3-μm beads was unchanged in $MCOLN1^{-/-}$ BMDMs, but was impaired using 6-μm beads (F), $n$ = 89–157.

G, H  Heterologous expression of mouse TPCs tagged with TagRFP-T (in red) rescues the phagocytic defect of $Tpcn1^{-/-}$ and $Tpcn2^{-/-}$ BMDM, but a pore-dead TPC2-mutant (N257A) does not, $n$ = 39–94. Images taken 40 min post-3-μm bead addition.

I  Raising cytosolic Ca²⁺ with ionomycin (1 μM) rescues the $Tpcn1^{-/-}$ and $Tpcn2^{-/-}$ phagocytic defect, whilst UTP (100 μM) and ML-SA1 (50 μM) only weakly rescue ($n$ = 84–192 cells).

Data information: Scale bars = 5 μm (G). Graphs presented as the mean ± S.E.M. and probability determined using one-way ANOVA (*** $P < 0.001$, ** $P < 0.01$, * $P < 0.05$ vs WT). See also Appendix Figs S2 and S3.

by exogenously re-expressing TPC1 or TPC2. Expression of TPC1 and TPC2 showed overlap with Lysotracker Green indicative of their acidic organelle localization, although the TPC1 localization was far more variable, as previously reported (Rietdorf *et al*, 2011) (Appendix Fig S3). The corresponding mouse TPC isoform fluorescently tagged with TagRFP-T was transfected into its respective knockout BMDM.

Quantifying bead uptake, the impaired FcγR-mediated phagocytosis in TPC-KO BMDMs was completely rescued by re-expression of TPC1 or TPC2 (Fig 2G and H). Furthermore, expression of a pore-dead mutant channel (TPC2-N257A) (Ruas *et al*, 2015a) failed to rescue the phagocytic defect in TPC2-KO BMDMs (Fig 2G and H), suggesting that TPC2 must be a functional channel to restore phagocytosis. Additional evidence suggested that the phagocytic defect in TPC knockout is a $Ca^{2+}$ deficiency because phagocytosis was rescued in knockout macrophages by indiscriminately elevating cytosolic $Ca^{2+}$ with the $Ca^{2+}$ ionophore, ionomycin (Fig 2I). These data confirm that inhibition of phagocytosis can be attributed to the loss of a $Ca^{2+}$-permeant TPCs in the knockout mice.

Intriguingly, not all $Ca^{2+}$ channels could restore bead uptake. Consistent with little role for $Ca^{2+}$ influx and ER-mediated $Ca^{2+}$ release (Fig 1E and H), the purinoceptor agonist, UTP, which signals via $IP_3$ and store-operated $Ca^{2+}$ entry (Lin & Chen, 1998), only weakly rescued TPC knockout by 10% ($Tpcn1^{-/-}$) and 9% ($Tpcn2^{-/-}$; Fig 2I). More remarkably, activation of the lysosomal TRPML-channel family with the agonist, ML-SA1 (Samie *et al*, 2013), had no effect upon phagocytosis in WT cells (number of beads per cell: Ctrl, $9.8 \pm 0.4$; ML-SA1, $9.7 \pm 0.4$; $n = 84–110$; $P > 0.5$) and barely affected bead uptake in TPC-KO cells (Fig 2I), even though the ML-SA1 evoked TRPML1-dependent $Ca^{2+}$ release (Appendix Fig S4C); this hints that there is selectivity for $Ca^{2+}$ sources that can drive phagocytosis (see below).

### Phagocytic uptake of live bacteria require TPC1 and TPC2

Whilst phagocytic defects in TPC-KO macrophages were observed using opsonized silica beads, we also investigated the phagocytosis of more physiological particles, live bacteria. The internalization of living non-pathogenic bacteria was conducted by incubating BMDMs with *Mycobacterium smegmatis* (3–5 μm bacilli) heterologously expressing the fluorescent protein, mCherry (whose fluorescence persists even in acidic lumina). Internalization was then assessed in three ways: (i) live single-cell confocal imaging of mCherry-tagged *M. smegmatis* taken up into BMDMs (Fig 3A); (ii) bacteria internalized by BMDMs are released from the macrophages by lysis, grown on agar plates and the number of bacterial colonies quantified (Fig 3B and C); and (iii) measuring mCherry fluorescence of BMDM-internalized bacteria using a plate reader (Fig 3D). All three methods showed that bacterial phagocytosis was significantly reduced by BMDMs from TPC knockout mice compared to WT controls (Fig 3A–D) and that internalization of bacteria was time-dependent (Fig 3D). We also infected cells with pathogenic live BCG (Pasteur strain), an attenuated form of *Mycobacterium bovis* (3 μm × 0.3 μm bacilli) and similarly expressing mCherry. Uptake of BCG was also reduced in TPC-KO BMDMs (Fig 3E), suggesting that the pathogenicity of bacteria does not affect the ability of BMDMs to phagocytose. In summary, the reduction in the phagocytic uptake of opsonized beads and living bacteria in TPC-null

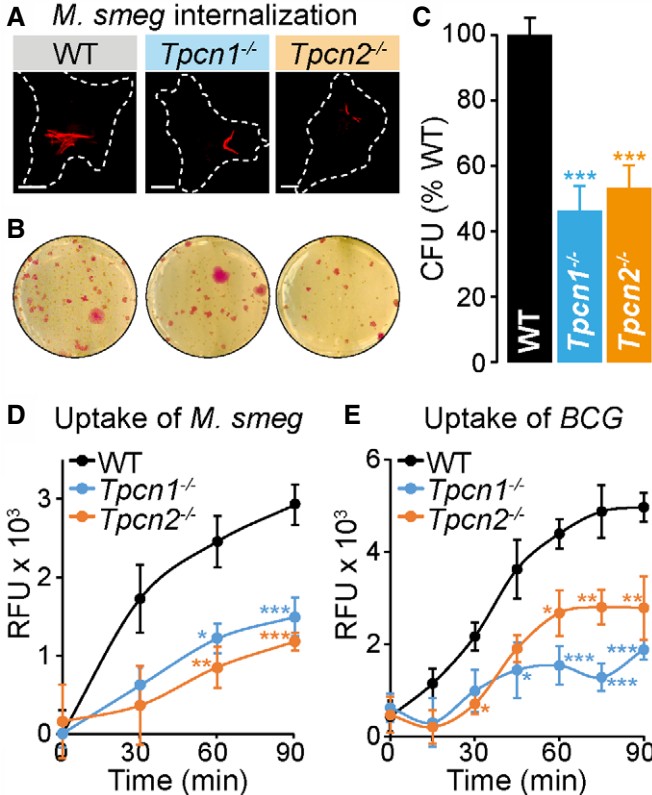

**Figure 3. Phagocytic uptake of live bacteria requires TPC1 and TPC2.**

A    Internalization of bacteria is reduced in $Tpcn^{-/-}$ BMDM, as assessed by single-cell imaging of BMDM incubated with *Mycobacterium smegmatis* tagged with mCherry (in red) for 1 h.

B, C    Internalization of bacteria assessed discontinuously in BMDM populations: (B) The resultant growth of bacteria colonies on agar plates following lysis of BMDM infected with *M. smegmatis* and (C) quantification of colony-forming units (CFU) expressed as a percentage of WT control ($n = 12$);

D, E    Real-time internalization of fluorescent bacteria into BMDM populations: (D) relative fluorescence units (RFU) of internalized mCherry-tagged *M. smegmatis* (E) and mCherry-tagged BCG over time, read using a plate reader ($n = 8$).

Data information: Scale bars = 10 μm (A). Graphs presented as the mean ± S.E.M. and probability determined using one-way ANOVA (*$P < 0.05$, **$P < 0.01$, ***$P < 0.001$).

macrophages, confirms that TPCs are essential for the phagocytic uptake of particles of diverse shape and size. Therefore, $Ca^{2+}$ release from endo-lysosomes via TPCs may be able to generate local $Ca^{2+}$ signals necessary for efficient phagocytosis.

### Endo-lysosomal motility

It is known that endo-lysosomes move towards the phagosome and ultimately fuse with it (Flannagan *et al*, 2012), but we examined the movement of acidic vesicles or TPCs more acutely. First, we performed triple-colour imaging of live BMDMs co-loaded with a red cytosolic $Ca^{2+}$ dye and Lysotracker Green to monitor acidic compartments and then added fluorescent IgG-coated beads (Fig 4A–D). When a bead contacted a cell, it evoked a rapid $Ca^{2+}$

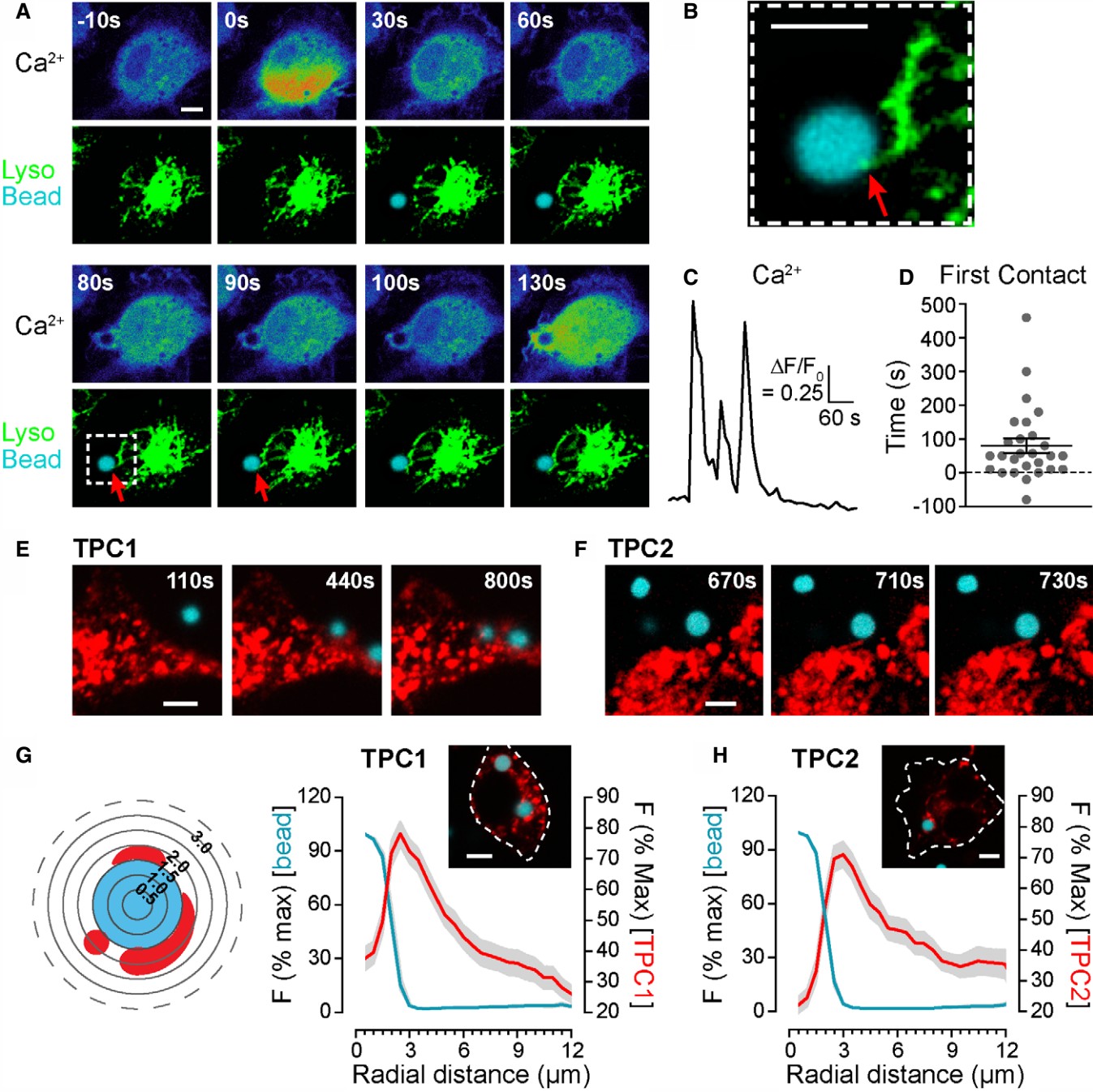

**Figure 4. Rapid intimate apposition of phagocytosed beads and lysosomes or TPCs.**

A–D  BMDMs were co-loaded with 2 μM Calbryte 590/AM (50 min) and 200 nM Lysotracker Green (LTG, 5 min) and, at room temperature, presented with 3-μm opsonized beads labelled with Alexa Fluor 647 (blue). A three-channel image was recorded every 10 s. The LTG brightness has been enhanced *post hoc* to visualize the dimmer, peripheral lysosomes. Time Zero is defined as the frame when the first Ca²⁺ signal was observed. (A) Images represent Ca²⁺ responses or an overlay of LTG and bead fluorescence. Times of each image are relative to the first Ca²⁺ spike. (B) Enlargement of the first contact point between lysosomes and bead (red arrow) as depicted by the dashed box, 80 s post-Ca²⁺. (C) Whole-cell Ca²⁺ responses from the cell in A. (D) Collation of the post-Ca²⁺ contact time, displayed as a scatter plot of individual cells and the mean ± S.E.M. ($n$ = 27 cells).

E–H  RAW264.7 cells were transfected with either TPC1-TagRFP-T or TPC2-TagRFP-T (red) and presented with 3-μm opsonized beads labelled with Alexa Fluor 647 (blue). Acute time-series of bead engagement were collected. Contact between TPCs and beads is depicted in the time-stamped micrographs, where time is arbitrary after the start of the image stack (E, F), i.e. 110 s is still prior to engagement. The spatial relationship between TPCs and beads was quantified using a target graticule of 0.5-μm bands centred on the engulfed bead; the mean fluorescence within each band was normalized to the fluorescence of the maximum band and plotted against this radial distance (G, H); $n$ = 35–36 cells.

Data information: All scale bars = 5 μm. Distribution data are expressed as the mean ± S.E.M.

signal; we defined this $Ca^{2+}$ spike as our reference point for FcR engagement ($t = 0$ s) and recorded the first time after this that lysosomes become closely apposed to the phagosome. We observed that lysosomes make rapid "contact" with the fluorescent bead, on average $81 \pm 21$ s ($n = 27$ cells) after the $Ca^{2+}$ spike. Note that the temporal spread (Fig 4D) encompasses the extremes of short times (when the bead lands directly on lysosomes by chance) as well as long times (when the bead engages the FcR outside the current confocal slice, but takes time to be drawn into the plane of focus). Figure 4A shows an example when peripheral lysosomes moving towards the bead in 80 s and remain around the phagosome. These data indicate that lysosome-phagosome apposition is rapid.

Similarly, tagged TPC1 or TPC2 became juxtapose to the internalized bead at early times after bead phagocytosis (Fig 4E and F). We observed a spectrum of labelling, from the intimate apposition of individual TPC-positive vesicles with the phagosome to the entire phagosome being surrounded by TPCs (Figs 2G and 4G and H). Overall, we find that endo-lysosomes make a rapid intimate apposition with phagosomes.

**Phagocytosis evokes $Ca^{2+}$ nanodomains around endo-lysosomes**

Our data indicated that local and not global $Ca^{2+}$ signals are important for phagocytosis (Fig 1E). In cells loaded with EGTA/AM, global signals are eliminated but local domains are preserved. We found that these residual local domains are driven by NAADP/TPC because phagocytosis was inhibited by Ned-19 in EGTA/AM-loaded cells (Fig 6E). This suggested that the local domains are endo-lysosomal in origin, so we created new approaches to show directly that endo-lysosomal $Ca^{2+}$ nanodomains are generated during, and are important for, phagocytosis. First, we monitored endo-lysosomal $Ca^{2+}$ domains using a genetically encoded $Ca^{2+}$ indicator (GECI) tethered to TPCs.

To mitigate against the prevalent pitfalls of using a high-affinity GECI fusion, we fused a low-affinity, high dynamic range GECI (G-GECO1.2, $K_d$ 1.2 $\mu$M (Zhao *et al*, 2011)) to TPC1 and TPC2 to better discriminate better between local $Ca^{2+}$ and global $Ca^{2+}$ spill-over. Importantly, fusion of the G-GECO1.2 to TPC2 did not alter either TPC targeting to lysosomes (Appendix Fig S4A) or the affinity of G-GECO1.2 for $Ca^{2+}$ as determined *in situ* ($P > 0.2$; Fig 5H). We used RAW 264.7 macrophages because the transfection rate was so low with primary cells that the chance of bead engagement per field proved unusable. Cells expressing TPC1- or TPC2-G-GECO1.2 were co-loaded with a red cytosolic chemical $Ca^{2+}$ dye (Calbryte 590) to simultaneously monitor local and global cytosolic $Ca^{2+}$. Figure 5B and C shows that Fc$\gamma$R-stimulated $Ca^{2+}$ oscillations detected with cytosolic Calbryte 590 are accompanied by oscillations in the TPC-G-GECO1.2 signal. This was not secondary to peri-lysosomal pH oscillations affecting the G-GECO1.2 because we could not detect pH oscillations with a targeted supereeliptic pHluorin (Appendix Fig S4D and F). This unequivocally shows that $Ca^{2+}$ increases in the peri-vesicular domain of lysosomes during phagocytosis.

However, it does not, per se, distinguish between the detection of a privileged local $Ca^{2+}$ domain and a bystander effect of global $Ca^{2+}$ from other sources, which is frequently overlooked. Therefore, as a control to empirically determine this spill-over, we repeated the experiments using unfused, cytosolic G-GECO1.2. Again, oscillations of the G-GECO1.2 signal coincided with the cytosolic Calbryte 590 signals but these cytosolic G-GECO1.2 signals were substantially smaller than those with the TPC fusion protein (Fig 5A and G). In other words, although some of the TPC-G-GECO1.2 signal did derive from global $Ca^{2+}$ spill-over (as detected by the unfused form), TPC-G-GECO1.2 must be detecting locally high $Ca^{2+}$ domains that the cytosolic form cannot.

We then provided additional evidence that the local $Ca^{2+}$ nanodomain and the global $Ca^{2+}$ signal are detected independently of one another during Fc$\gamma$R signalling. First, we selectively inhibited the local $Ca^{2+}$ domain: inhibition of TPC2 activity by either the NAADP antagonist, Ned-19 (Fig 5D and G), or genetic inhibition of TPC2 by a point mutation (D276K) that blocks ion permeation (Wang *et al*, 2012) (Fig 5F and G) each reduced the TPC2-G-GECO1.2 signal to that of cytosolic G-GECO1.2 alone (and left cytosolic Calbryte 590 spikes). Conversely, we exploited EGTA/AM to chelate global but not local $Ca^{2+}$ signals: in cells loaded with EGTA, the cytosolic $Ca^{2+}$ signal measured with Calbryte 590 was all but eliminated, whereas a local TPC2-G-GECO1.2 signal remained (Fig 5E and G). Note that flooding the cell with excess $Ca^{2+}$ at the end of the run overcame the cytosolic buffering and proved a positive control for Calbryte 590 loading.

To further demonstrate that TPC2-G-GECO1.2 detects $Ca^{2+}$ released locally from the lysosome, we eliminated $Ca^{2+}$ release that spills over from the ER by emptying the ER $Ca^{2+}$ store with CPA or thapsigargin in $Ca^{2+}$-free medium. In untreated cells, beads caused robust oscillations in both the cytosolic Calbryte 590 and TPC2-G-GECO1.2 (Fig 5I). As in Fig 1G, pre-emptying the

**Figure 5. Peri-lysosomal $Ca^{2+}$ nanodomains around TPCs.**

A–F    RAW 264.7 macrophages loaded with a cytosolic chemical dye (Cyto-dye) to detect global cytosolic $Ca^{2+}$, and heterologously expressing (B) TPC1-G-GECO1.2 (TPC1-GG) or (C–E) TPC2-G-GECO1.2 (TPC2-GG) to detect peri-endo-lysosomal $Ca^{2+}$ domains, which were not detected by (F) a TPC2 mutant (D276K)-G-GECO1.2 that blocks ion permeation or (A) unfused cytosolic G-GECO1.2 (Cyto-GG). (A–F) Single-cell $Ca^{2+}$ traces upon Fc$\gamma$R activation by opsonized 3-$\mu$m beads in $-Ca^{2+}_o$ expressed as a percentage of the maximum fluorescence of the indicator, determined by adding 2 $\mu$M ionomycin and 10 mM $CaCl_2$ at the end of each experiment (% $F_{max}$) (D) TPC2-G-GECO1.2 activity was inhibited by 10 $\mu$M Ned-19, but was unaffected by (E) chelating global $Ca^{2+}$ with EGTA/AM (100 $\mu$M for 2 min prior to bead addition).

G    Summary data of the first spike from cells A–F (subsequent spikes show the same pattern; Appendix Fig S4B), $n = 87$–274 cells; ***$P < 0.001$ vs Cyto-GG, ###$P < 0.001$ vs TPC2-GG ctrl, †††$P < 0.001$ vs Cyto-GG.

H    Fusion of G-GECO1.2 to TPC2 did not alter the affinity of G-GECO1.2 for $Ca^{2+}$ determined in permeabilized cells ($n = 34$–71 cells).

I, J    (I) Single-cell $Ca^{2+}$ traces upon Fc$\gamma$R activation by opsonized 3-$\mu$m beads in $-Ca^{2+}_o$ expressed as a percentage of $F_{max}$. (I, J) Pre-emptying the ER stores with CPA or thapsigargin (Tg) abolished the global cytosolic $Ca^{2+}$ response evoked by beads, but a TPC2-G-GECO1.2 component remained, $n = 17$–121 cells.

K–M    No local $Ca^{2+}$ signal was detected with TPC2-G-GECO1.2 when $Ca^{2+}$ spiking was evoked by ER-$Ca^{2+}$ release with 100 $\mu$M UTP (K, L) or 10 $\mu$M CPA (M); only a bystander response equivalent to unfused cytosolic G-GECO1.2 was detected ($n = 121$–182 cells; *$P < 0.05$).

Data information: Graphs presented as the mean $\pm$ S.E.M. and probability determined using one-way ANOVA. See also Appendix Figs S4 and S5.

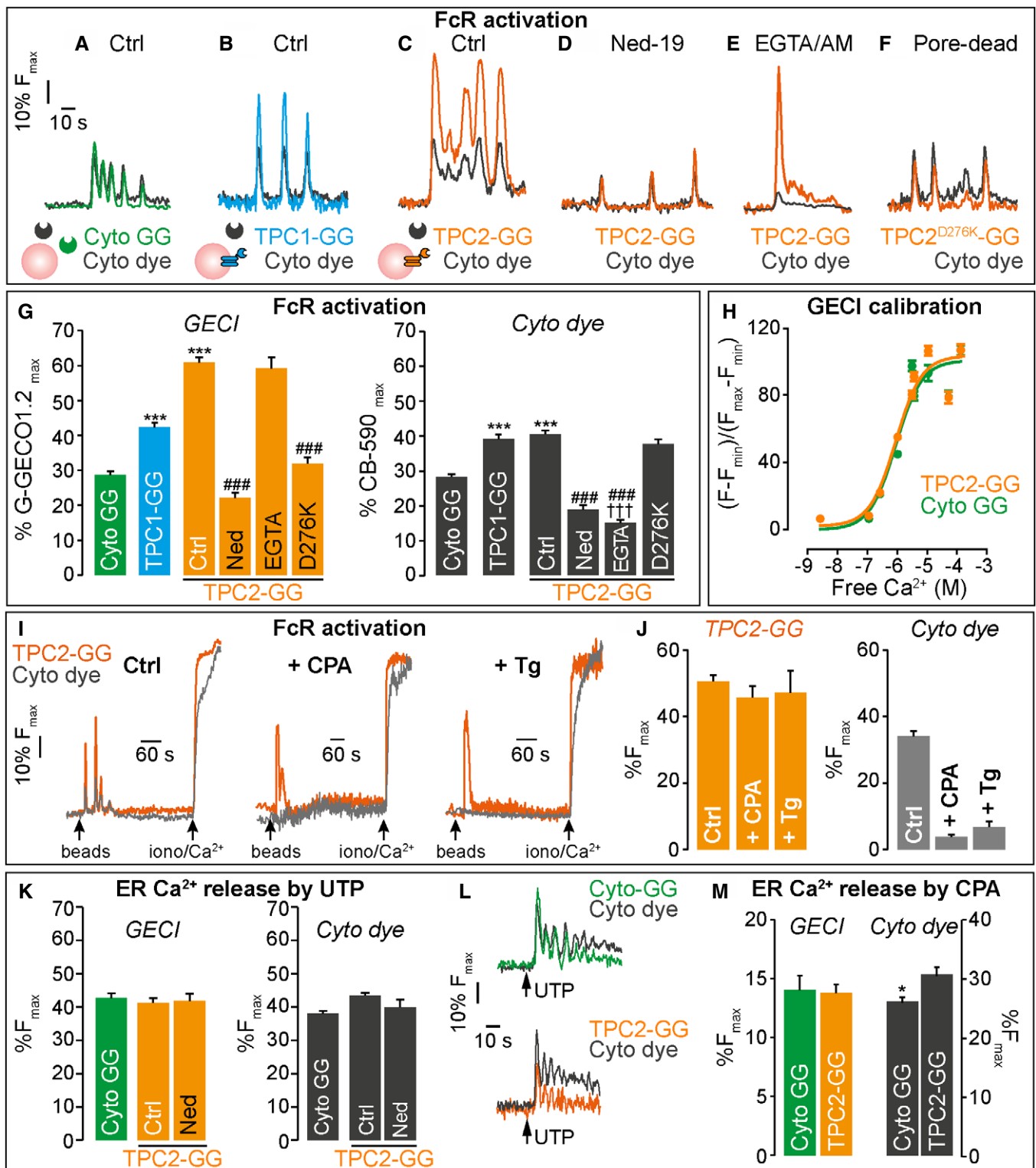

**Figure 5.**

ER stores with CPA or thapsigargin abolished the cytosolic $Ca^{2+}$ response evoked by beads, but remarkably a TPC2-G-GECO1.2 component remained (Fig 5I and J). This strongly suggests that TPC2-G-GECO1.2 is reporting lysosomal $Ca^{2+}$ release that is independent of ER-dependent $Ca^{2+}$ release. Together with the EGTA/AM data (Fig 5E and G), this unequivocally shows that TPC2-G-GECO1.2 can detect local $Ca^{2+}$ responses even in the absence of global $Ca^{2+}$ signals.

Importantly, we show that the local $Ca^{2+}$ domain around TPCs was stimulus-specific: when $Ca^{2+}$ spiking was evoked with UTP, an $IP_3$-coupled agonist (Lin & Chen, 1998), or with the ER $Ca^{2+}$ pump inhibitor, CPA, no local domain was detected with TPC2-G-GECO1.2, only a "bystander response" equivalent to that seen with unfused G-GECO1.2 (Fig 5K–M). It is informative that the strong non-physiological stimulus, ionomycin, experimentally elevates peri-lysosomal $Ca^{2+}$ to even higher levels than does FcR activation, as judged by the TPC2-G-GECO1.2 response (% $F_{max}$) $80 \pm 3\%$ ($n = 65$), and thereby can explain why ionomycin rescues the phagocytic defect in TPC-KO cells (Fig 2I).

Finally, recording TPC2-G-GECO1.2 responses at higher spatial resolution indicates that lysosomes are activated across the entire cell, not merely where the phagosome is formed (Appendix Fig S5). In summary, we have rigorously confirmed that local $Ca^{2+}$ nanodomains around TPCs are selectively generated during phagocytosis across the lysosomal network.

### Peri-lysosomal $Ca^{2+}$ nanodomains drive phagocytosis

Having confirmed the existence of local lysosomal $Ca^{2+}$ nanodomains, we then tested whether they are instrumental in driving phagocytosis. Although the selective inhibition by BAPTA (Figs 1E and 6E) suggests that local $Ca^{2+}$ domains are important, it does not tell us where they are. We therefore selectively collapsed local lysosomal $Ca^{2+}$ domains by targeting $Ca^{2+}$-buffering proteins to the lysosomal surface. To mitigate against potentially chronic effects of peri-lysosomal $Ca^{2+}$-buffering, we engineered a chemical-induced dimerization (CID) system (FKBP12/FRB*) that uses a rapalog (AP21967, a non-inhibitor of mTOR) to acutely bring a calbindin-2 tandem-dimer (CB2x2, 10 $Ca^{2+}$-binding sites) to lysosomes (Fig 6A), as indicated by the increase in Pearson's colocalization coefficient (Fig 6B). With CB2x2 in the cytosol, phagocytosis of beads proceeded as normal. However, when CB2x2 was brought to the lysosome by pre-treatment with rapalog, AP21967, phagocytosis was substantially inhibited (Fig 6C and D). This was not a steric crowding effect, because a similarly sized triple mTagBFP2 construct had no effect upon phagocytosis (Fig 6D). Finally, the effect of CB2x2 correlated with its $Ca^{2+}$-binding capacity because a single point mutation in each of its 5 functional EF-hands (CB2-mutx2) diminished its ability to block phagocytosis (Fig 6D). These data strongly suggest that peri-lysosomal $Ca^{2+}$ nanodomains drive phagocytosis.

### Stimulus-dependent recruitment of different $Ca^{2+}$ channels

Remarkably, our data suggest that the phagocytosis of different-sized particles requires different lysosomal $Ca^{2+}$ channels: TPCs are universally important for the uptake of bacteria (Fig 3), as well as small (3 µm) and large (6 µm) beads alike (Fig 2C and D); in contrast, TRPML1 is only required for the phagocytosis of large particles (Fig 2E and F) (Samie et al, 2013; Dayam et al, 2015). Potentially, this is due to differential activation of these channels by different-sized particles. Given our ability to assess channel activation by recording local $Ca^{2+}$ domains, we explicitly compared TPC2 and TRPML1 activation using their respective G-GECO1.2 fusions; TRPML1-G-GECO1.2 also correctly targeted to the lysosomes (Appendix Fig S4A). When recording global

responses with cytosolic G-GECO1.2, 3- and 6-µm beads evoked similar, small $Ca^{2+}$ spikes (Fig 6F and G), the variable lag reflecting the random time for beads to float down to the cell. In keeping with our hypothesis, both small and large beads activated TPC2 because large TPC2-G-GECO1.2 $Ca^{2+}$ nanodomain signals were observed with each stimulus (Fig 6F and G). Responses were very different for TRPML1-G-GECO1.2: with small 3-µm beads, TRPML1 was not activated and only a small global response was observed; in contrast, the larger 6-µm beads did activate TRPML1 and large local TRPML1-G-GECO1.2 responses were seen (Fig 6F and G). Therefore, this stimulus-specific pattern of channel activation entirely agrees with and explains the differential phagocytic reliance on the lysosomal channel families (Fig 2C–F), but reinforces TPC activation as the universal lysosomal $Ca^{2+}$ nanodomain generator in phagocytosis.

### Dynamin promotes phagocytosis

Having identified that local $Ca^{2+}$ release via TPCs is crucial for phagocytosis, we sought to identify the downstream $Ca^{2+}$-sensitive pathways recruited by TPCs. For many forms of endocytosis (of which phagocytosis is a specialized variant), the $Ca^{2+}$-sensitive GTPase dynamin is required for scission, whereby the internalized vesicle pinches off from the plasma membrane as contractile helical polymers of dynamin wrap around and constrict the neck of a budding vesicle (Antonny et al, 2016). Indeed, such a role for dynamin in phagocytosis has been previously suggested (in addition to its role in the exocytosis of endo-membranes to sites of phagocytosis required for growth of pseudopodia and actin polymerization) (Gold et al, 1999; Di et al, 2003; Marie-Anais et al, 2016). However, the signalling pathway activating this is unknown and so we hypothesized that TPCs couple to dynamin via an intermediate $Ca^{2+}$-dependent protein.

First, to reaffirm a role for dynamin during FcγR-mediated phagocytosis, we pre-incubated WT BMDMs with dynamin inhibitors dynasore, or Dyngo-4a (McCluskey et al, 2013) (a more potent and selective dynasore analog). The dynamin inhibitors significantly repressed bead internalization (Fig 7A), consistent with previous studies that dynamin is essential for efficient FcγR-mediated phagocytosis (Gold et al, 1999; Di et al, 2003). Dynasore and Dyngo-4a also inhibited phagocytosis of beads in RAW 264.7 macrophages (Appendix Fig S2C). As a complement to pharmacological inhibition, expression of a dominant-negative GTPase-deficient mutant of dynamin-2 (K44A) also inhibited phagocytosis in WT BMDM (Fig 7C), as others have reported (Gold et al, 1999; Marie-Anais et al, 2016).

We then postulated that the reduced phagocytic competence of TPC-KO cells was due to a failure to efficiently activate dynamin (itself a $Ca^{2+}$-dependent process—see below). We applied the dynamin activator Ryngo-1-23 in TPC-KO BMDM during phagocytosis in order to activate dynamin in a $Ca^{2+}$-independent manner, by stimulating dynamin oligomerization, basal GTPase activity and fusion-pore closure (Gu et al, 2014; Lasic et al, 2017). Ryngo-1-23 completely rescued the phagocytic defect of TPC-KO macrophages (Fig 7B) whilst having little effect in WT cells (Fig 7B).

We also genetically activated dynamin. Dynamin-2 (the ubiquitous isoform) is constitutively phosphorylated at serine-764 (S764) which inhibits its activity; stimulus-induced dephosphorylation of

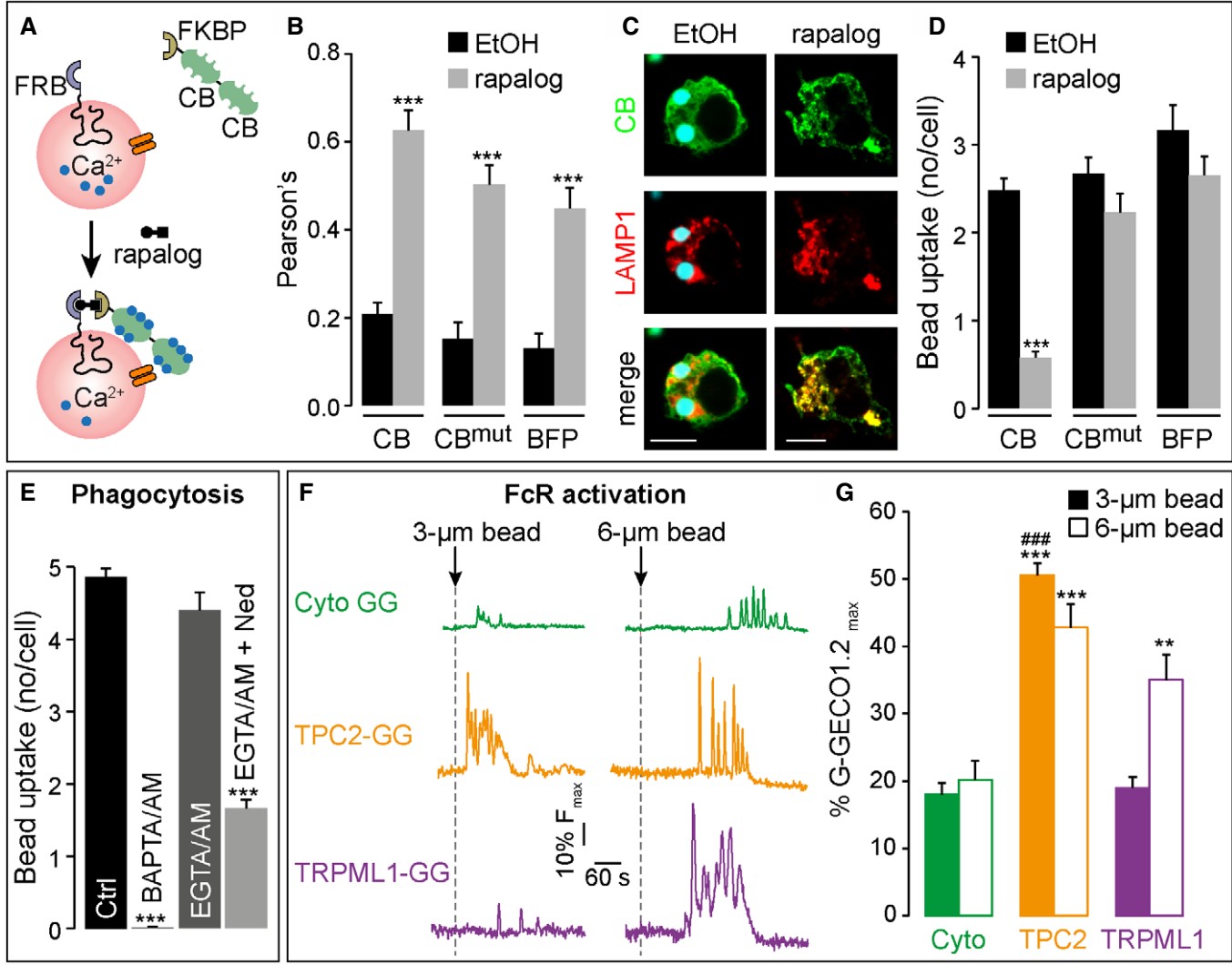

**Figure 6. Bead uptake requires lysosomal Ca²⁺ nanodomains, but different stimuli recruit different lysosomal Ca²⁺ channels.**

A  Cartoon depicting the rapalog CID system (FKBP12/FRB*) to acutely bring tandem Ca²⁺-binding proteins, calbindin-2 (CB), to lysosomes.

B  Translocation of CB, a non-Ca²⁺ binding CB mutant (CBmut) or a triple mTagBFP2 (BFP) to the lysosome (LAMP1), as indicated by the increase in Pearson's correlation coefficient was induced by 250 nM rapalog (30 mins) compared with ethanol (EtOH) control-treated RAW 264.7 cells. $n$ = 11–25 cells; ***$P$ < 0.001.

C  Images of a single RAW 264.7 cell displaying yellow puncta indicative of successful repositioning of CB (in green) to the lysosome (in red) in the presence of rapalog but not EtOH.

D  When CB was brought to the lysosome (+ rapalog), uptake of beads was inhibited, whereas the non-Ca²⁺ binding CB (CB mut; $n$ = 46–187 cells) or triple mTagBFP2 (BFP; $n$ = 38–52 cells) did not block phagocytosis; ***$P$ < 0.001.

E  Bead uptake in BMDMs with and without loading of 25 μM EGTA/AM and 10 μM Ned-19. EGTA-insensitive phagocytosis of beads was inhibited by Ned-19 ($n$ = 75–102 cells) in WT BMDM; ***$P$ < 0.001.

F, G  RAW 264.7 macrophages expressing cytosolic G-GECO1.2 (Cyto-GG), TPC2-G-GECO1.2 (TPC2-GG) or TRPML1-G-GECO1.2 (TRPML1-GG). (F) Single-cell Ca²⁺ traces upon FcγR activation by opsonized 3-μm or 6-μm beads in −Ca²⁺ₒ expressed as a percentage of the maximum indicator fluorescence (% $F_{max}$ = 2 μM ionomycin and 10 mM CaCl₂). (G) Summary of first Ca²⁺ spike; $n$ = 69–121 (3-μm bead), 22–46 (6-μm beads) cells; ***$P$ < 0.001, **$P$ < 0.01 vs Cyto-GG, ###$P$ < 0.001 TPC2-GG vs TRPML1-GG.

Data information: Scale bars = 10 μm (C). Bar charts presented as the mean ± S.E.M. and probability determined using one-way ANOVA.

S764 activates dynamin (Chircop *et al*, 2010, 2011). We therefore expressed in TPC1-KO macrophages a constitutively active form of dynamin-2, by mutating S764 to alanine (mimicking the dephosphorylated state). Similar to Ryngo-1-23, dynamin-2 (S764A) completely rescued bead uptake in TPC1-KO BMDM (Fig 7C), indeed more so than did wild-type dynamin-2 (Fig 7C) which is presumably partially phosphorylated and inhibited. Taken together, the

pharmacological and genetic data affirm that dynamin is crucial for phagocytosis and that its activation rescues the phagocytic defect of TPC-KO cells. Just like the Ca²⁺ rescue (Fig 2I), this confirms that TPC-KO cells are phagocytically competent but TPCs are the conduit for the Ca²⁺-dependent activation of dynamin-2.

The spatial distribution of dynamin is also revealing. Expression of wild-type dynamin-2 or the dynamin-2 S764A mutant could show

an accumulation around the bead circumference as revealed by HA-antibody staining (Fig 7D), suggesting that dynamin-2 accumulates in the phagocytic cup (Gold *et al*, 1999; Di *et al*, 2003). In Fig 7D, dynamin-2 initially is recruited to and encircled the internalizing bead during engulfment but later an intense spot of the dynamin

signal occurred, consistent with scission. Dynamin association with the phagosome is transient (Marie-Anais *et al*, 2016) which is presumably why not all beads are labelled at one time, with totally internalized beads lacking significant dynamin-2 accumulation (Di *et al*, 2003).

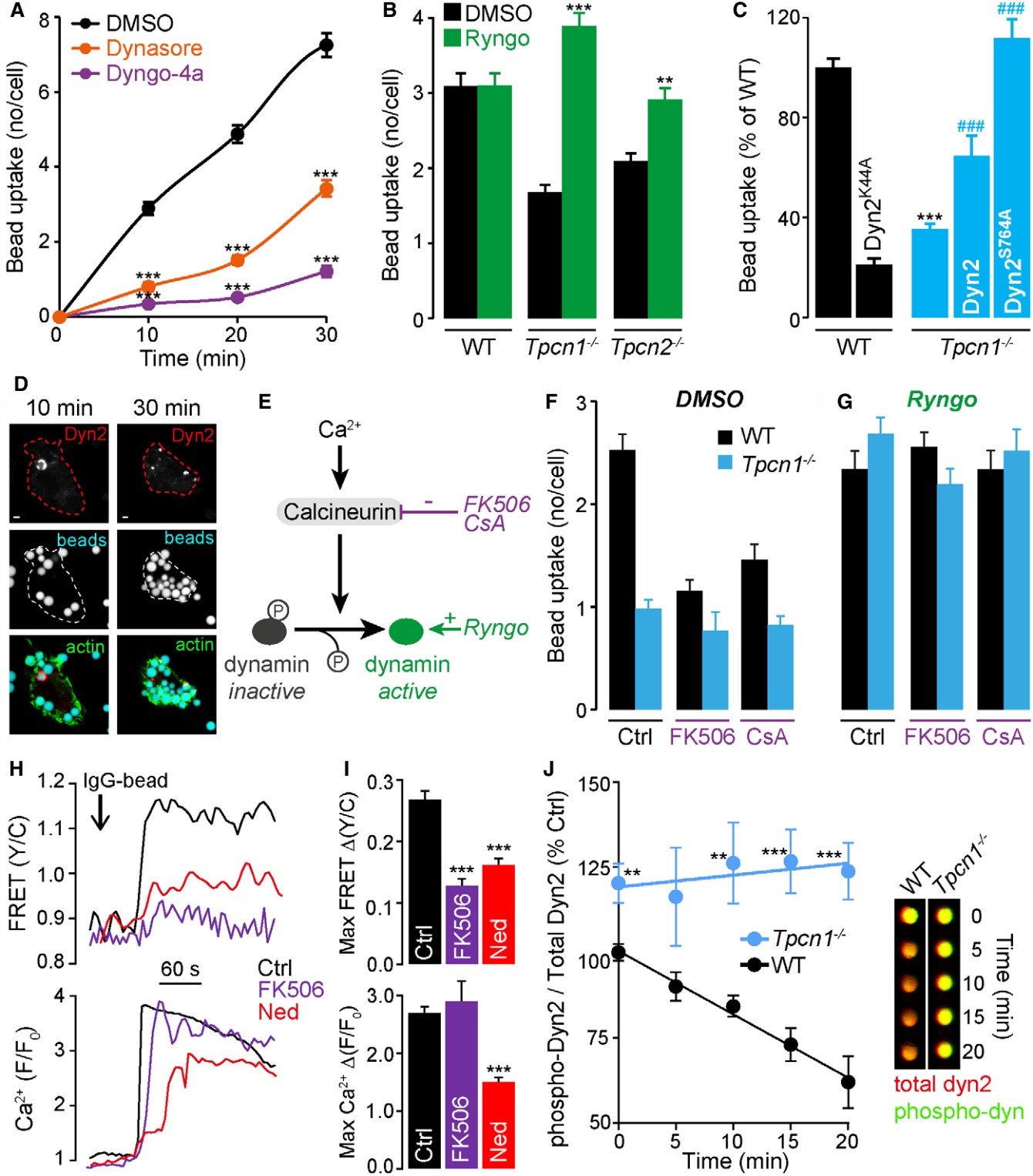

**Figure 7.**

◀

**Figure 7. TPC-mediated Ca$^{2+}$ release recruits calcineurin which in turn activates dynamin.**

A   Dynamin inhibitors (80 μM dynasore or 20 μM Dyngo-4a) inhibit phagocytosis of opsonized 3-μm beads in WT BMDM; $n$ = 101–149 cells, ***$P$ < 0.001.
B   Rescue of the TPC-KO phagocytic defect by the dynamin activator, Ryngo-1-23 (80 μM); $n$ = 128–241 cells, **$P$ < 0.01, ***$P$ < 0.001.
C   In WT BMDM, expression of a dominant-negative dynamin-2 mutant (Dyn2 K44A) reduced phagocytosis ($n$ = 68–184). Expression of wild-type dynamin-2 (Dyn2) or constitutively active dynamin-2 (S764A mutant, mimicking the dephosphorylated state) in *Tpcn1$^{-/-}$* macrophages restored phagocytosis of 3-μm beads after 10 min, $n$ = 10–76 cells; ***$P$ < 0.001 vs WT control, ###$P$ < 0.001 vs *Tpcn1$^{-/-}$* control.
D   Single WT BMDM expressing HA-tagged dynamin-2 (immunolabelled with anti-HA, in red) undergoing phagocytosis of IgG-3-μm beads (blue). Images were taken 10 or 30 min post-addition of beads; actin (green) labelled with Phalloidin Alexa 488.
E   Cartoon indicating sites of action of pharmacological agents.
F   Calcineurin inhibitors, FK506 (10 μM) or cyclosporin A (CsA, 10 μM) inhibited phagocytosis in WT cells, $n$ = 65–159 cells.
G   Ryngo (80 μM) rescued phagocytosis defects in *Tpcn1$^{-/-}$* BMDM even in the presence of FK506 or CsA, $n$ = 103–208 cells.
H, I   Calcineurin activity and cytosolic Ca$^{2+}$ responses simultaneously monitored in single RAW 264.7 using CaNAR2 and jRGECO1a, respectively, upon addition of 3-μm beads. CaNAR2 signals (FRET Y/C) were inhibited by FK506 (10 μM) and Ned-19 (10 μM); (H) single-cell traces and (I) mean maximum responses. $n$ = 65–169; ***$P$ < 0.001 vs control.
J   Phagocytosis induced dephosphorylation of dynamin-2 (ratio of phospho-dynamin to total dynamin-2 expressed as a percentage of WT at time zero) in WT BMDM, but not in *Tpcn1$^{-/-}$* BMDM; **$P$ < 0.01, ***$P$ < 0.001 (time-point matched unpaired *t*-test) $n$ = 10–15. Measured using in-cell western assay, representative wells of plate image: total dyn2 (red) and phospho-dyn2 (green). See also Appendix Figs S6 and S7.

Data information: Scale bars = 2 μm (D). Graphs presented as the mean ± S.E.M. and probability determined using one-way ANOVA.

## Calcineurin activity is required for dynamin activation during phagocytosis

Thus far, our data are consistent with a failure to properly activate dynamin in TPC-KO cells, but they do not provide the link between TPCs and dynamin. The dephosphorylation of dynamin-2 can be regulated by the Ca$^{2+}$-dependent phosphatase calcineurin in HeLa cells (Chircop *et al*, 2010, 2011) (Fig 7E). We therefore hypothesized that local TPC-dependent Ca$^{2+}$ release activates calcineurin, which in turn dephosphorylates and activates dynamin. Indeed, in HeLa cells, BAPTA/AM prevents dynamin-2 dephosphorylation (Chircop *et al*, 2010).

First, we tested whether calcineurin is required for FcγR-mediated phagocytosis in WT cells by applying the chemically unrelated inhibitors of calcineurin activity, FK506 and cyclosporin A (CsA) (Li *et al*, 2011). Accordingly, phagocytosis was significantly reduced in WT BMDM treated with both FK506 and CsA (Fig 7F); indeed, the calcineurin inhibition functionally mimicked TPC deletion in terms of its magnitude. In contrast, inhibition of calcineurin in TPC1-KO BMDMs had no further effect on their already reduced phagocytic capacity (Fig 7F) suggesting that the residual component was both calcineurin- and TPC-independent.

Second, if calcineurin is indeed upstream of dynamin, then direct activation of dynamin should be able to by-pass the blockade by the calcineurin inhibitors. Therefore, we applied the dynamin activator, Ryngo-1-23, which circumvents the dephosphorylation of dynamin. In WT cells, Ryngo-1-23 was indeed able to reverse the inhibition by FK506 or CsA (Fig 7G). In TPC1-KO cells, a similar rescue was observed by Ryngo-1-23 (Fig 7G). Together, the data suggest that (i) a major component of the phagocytosis mechanism shares a reliance upon calcineurin and TPCs; (ii) dynamin activation by Ryngo-1-23 acts downstream of calcineurin.

## FcγR activation stimulates calcineurin activity in live macrophages

The previous data are consistent with the order of events being Ca$^{2+}$ release via NAADP/TPC leading to activation of calcineurin and the dephosphorylation (and hence activation) of dynamin. To confirm that Ca$^{2+}$-release via NAADP/TPC activates calcineurin, we used a genetically encoded fluorescence resonance energy transfer (FRET)-based reporter to monitor calcineurin activity (CaNAR2 (Mehta *et al*, 2014)). This is the first example of real-time monitoring of calcineurin activity in living macrophages. First, we validated the use of the FRET reporter in live single cells by simultaneously monitoring cytosolic Ca$^{2+}$ responses (with a red GECI) and calcineurin activity with CaNAR2. Upon ionomycin addition, increased FRET was observed indicative of calcineurin activation, and this was reversed by chelating extracellular Ca$^{2+}$ with EGTA; importantly, CaNAR2 signals were inhibited by the calcineurin inhibitor FK506 (Appendix Fig S6A, B and D). As expected, the Ca$^{2+}$ signal was unaffected by FK506 (Appendix Fig S6A–C). Furthermore, we confirmed that lysosome-mediated Ca$^{2+}$ release resulted in calcineurin activation (Appendix Fig S6E and F).

Having confirmed a link between lysosomal Ca$^{2+}$-release and calcineurin, we used the biosensor to directly probe the role of FcγR-induced Ca$^{2+}$ signals in modulating endogenous calcineurin activity in RAW 264.7 macrophages (Fig 7H and I). Upon IgG-bead addition, cytosolic Ca$^{2+}$ signals were accompanied by an increase in the FRET signal, indicative of calcineurin activation (Fig 7H). As expected, inhibition of calcineurin with FK506 significantly inhibited FRET but was without effect on the Ca$^{2+}$ signal (Fig 7H and I). Finally, we could confirm a new specific role for NAADP in the stimulation of calcineurin because the NAADP antagonist, Ned-19, significantly reduced both Ca$^{2+}$ signals and calcineurin activity induced by IgG-bead activation of FcγR (Fig 7H and I).

## FcγR-mediated phagocytosis requires TPC-dependent dephosphorylation of dynamin-2

Having linked NAADP/TPC to calcineurin, the final requirement was to link these to dynamin phosphorylation. That the constitutively active, un-phosphorylated dynamin-2 (S764A) rescued phagocytosis in TPC1-KO macrophages (Fig 7C) was consistent with this scheme, but was indirect evidence. We therefore directly assessed dynamin-2 phosphorylation using a phospho-specific antibody for dynamin. The specificity of the phospho-serine antibody was confirmed when dynamin-2 immunoreactivity in COS-7 cell lysates was

eliminated by the S764A mutation (Appendix Fig S7A and B) or in WT BMDM lysates treated with lambda protein phosphatase (Appendix Fig S7C).

During the internalization of opsonized 3-μm beads, the phosphorylation state of dynamin-2 was assessed by both immunoblotting (Appendix Fig S7E and F) and by an in-cell western assay (Fig 7J). Using the phospho-specific antibody for dynamin, we revealed that FcγR-mediated phagocytosis parallels a time-dependent dynamin dephosphorylation in WT macrophages (rapid, in ≤ 1 min; Appendix Fig S7E and F). Strikingly, not only was dynamin dephosphorylation not observed in TPC1-KO cells, but basal dynamin-2 phosphorylation levels in TPC1-KO macrophages were enhanced compared with WT (Fig 7J and Appendix Fig S7E). The data finally confirm that dynamin dephosphorylation is indeed driven by TPCs during phagocytosis.

### Lysosomal positioning is essential for phagocytosis and dynamin activation

Having shown that lysosomes rapidly appose the phagosome (Fig 4), we hypothesized that lysosome positioning was important for driving bead uptake. Therefore, we prevented lysosomal movement by immobilizing lysosomes at the microtubule-organizing centre (MTOC), away from the phagosome. This was effected acutely by a CID (FKBP12/FRB*) system: acute addition of rapalog, AP21967, tethers lysosomes to a dynein-binding protein that moves vesicles to the MTOC (Bentley et al, 2015). The striking clustering of lysosomes at the MTOC (Fig 8A) was only observed in the presence of the entire tripartite complex of FKBP12-rapalog-FRB* because it was not observed when one component (either rapalog or FRB*) was omitted (Fig 8A and B). Moreover, this clustering was selective for lysosomes because neither the ER nor mitochondria were affected (Fig 8C). Consistent with our movement hypothesis, phagocytosis of 3-μm beads was markedly inhibited by immobilizing lysosomes, but only by the tripartite combination that successfully locked lysosomes at the MTOC (Fig 8A and B).

If lysosomal positioning is essential for activating dynamin, then direct activation of dynamin should rescue the rapalog effect. Accordingly, both the chemical activator, Ryngo-1-23 and expression of constitutively active dynamin-2-S764A, acting independently of lysosomes, completely restored bead uptake in cells with immobilized lysosomes (Fig 8D). This rescue is of importance because it is consistent with lysosomes being essential for providing a dynamin activation signal and not simply by providing additional membrane for the growing phagosome; that is, with lysosomes locked at the MTOC, other membrane sources must be sufficient for enlarging the phagosome.

Superficially, our results appeared to differ from a previous study where lysosomes were essential for providing membrane to the phagosome (Samie et al, 2013). However, in that study, 6-μm beads were used whose volume is eight-fold greater than that of a 3-μm bead, so we hypothesized that lysosomes only provide additional membrane for larger particles. Accordingly, and in contrast to 3-μm beads, when the phagocytosis of 6-μm beads was inhibited by immobilizing lysosomes at the MTOC (Fig 8E), it was not rescued by dynamin activation (neither Ryngo-1-23 nor dynamin-2-S764A). This suggests that cells cannot phagocytose larger particles without lysosomal movement, presumably via membrane provision (Samie

et al, 2013), because membranes are limiting. This reveals that lysosomes contribute to phagocytosis via multiple, coincident pathways, but the TPC/dynamin pathway is of universal importance. Taken together, our data indicate that TPCs have an essential role in driving early phagosome formation.

## Discussion

The purpose of this study was to delineate the signal transduction pathway that couples FcγR engagement to particle phagocytosis in macrophages. The role of $Ca^{2+}$ in phagocytosis has long proven controversial (Westman et al, 2019). Our results may not only resolve this controversy but define a new pathway of endo-lysosomal $Ca^{2+}$ signalling during phagocytosis and delineate the downstream decoding elements.

### FcγR-mediated phagocytosis depends on endo-lysosomal $Ca^{2+}$ stores

The $Ca^{2+}$ signals that accompany phagocytosis are a complex summation, and interaction of, simultaneous parallel pathways. The substantial global signals measured with cytosolic $Ca^{2+}$ reporters are predominantly comprised of $Ca^{2+}$ release from the ER (via $IP_3Rs$) and $Ca^{2+}$ entry across the plasma membrane (via Orai) (Demaurex & Nunes, 2016; Westman et al, 2019), but these potentially mask local $Ca^{2+}$ nanodomains that are invisible in global recordings. We have shown that ER/$Ca^{2+}$ influx ($IP_3R$/Orai) pathways are not required but that the endo-lysosomal NAADP/TPC axis generates local $Ca^{2+}$ nanodomains that are essential for phagocytosis. At phagocytosis, this elevates endo-lysosomes from the classical downstream terminal degradative compartments to upstream early signal generators.

The emerging consensus is that endo-lysosomal TPCs are both $Na^+$- and $Ca^{2+}$-permeable channels but, unusually, with the $Na^+$/$Ca^{2+}$ permeability ratio dependent on the activating ligand, be it $PI(3,5)P_2$ lipid or NAADP, the latter favouring $Ca^{2+}$ fluxes (Ruas et al, 2015a; Gerndt et al, 2020). The NAADP/TPC axis is increasingly implicated in health and disease (Patel & Kilpatrick, 2018), with precedence for its acting via local $Ca^{2+}$ domains (Zhu et al, 2010; Morgan et al, 2013; Grimm et al, 2014; Lin-Moshier et al, 2014; Hockey et al, 2015), but its role in phagocytosis was unknown. We now show that the TPC system provides essential early upstream signals to activate calcineurin/dynamin.

Specifically, the first evidence that the depressed phagocytosis in TPC-KO macrophages was a $Ca^{2+}$ defect was its rescue by the $Ca^{2+}$ ionophore, ionomycin. However, the striking fact that other endogenous $Ca^{2+}$ channels could not substitute for TPCs (namely $IP_3R$, Orai, TRPMLs) implied that there was signal compartmentation and that TPCs generated unique $Ca^{2+}$ nanodomains to drive phagocytosis. The differential effects of EGTA and BAPTA (Kidd et al, 1999) on bead uptake were further evidence for local $Ca^{2+}$ signalling (Samie et al, 2013; Sun et al, 2020)—replicating and resolving previous interpretational discrepancies (Westman et al, 2019)—but we formally monitored and manipulated endo-lysosomal $Ca^{2+}$ nanodomains using genetically targeted $Ca^{2+}$ reporters and $Ca^{2+}$ buffers. The approaches revealed that, even when the global cytosolic $Ca^{2+}$ signals were entirely repressed, local TPC-

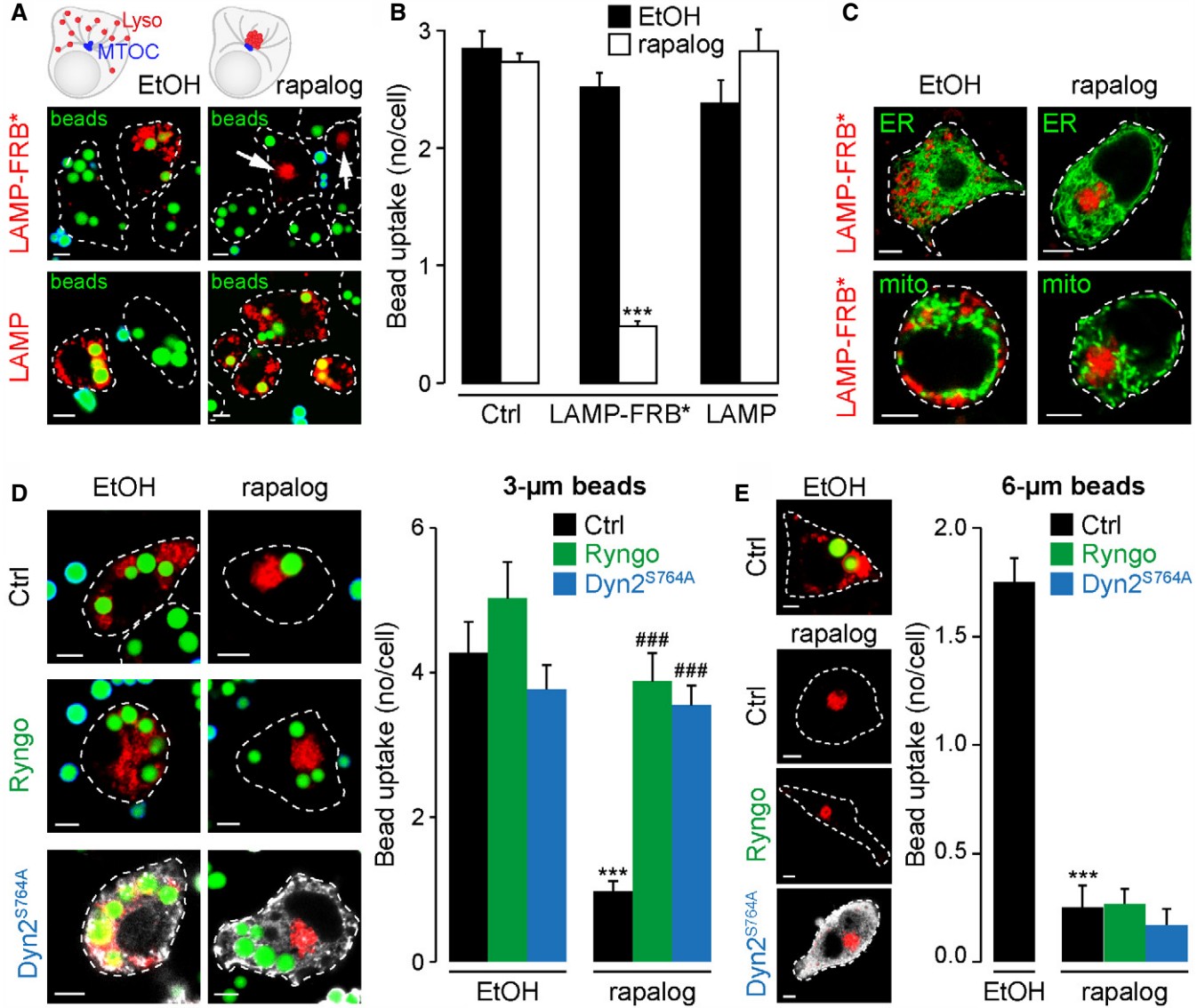

**Figure 8. Lysosomal clustering inhibits dynamin-dependent phagocytosis.**

A–C   RAW 264.7 macrophages expressing the dynein-binding protein (mTagBFP2-BicD2-FKBP12) for rapalog-induced lysosome repositioning to the MTOC, also
co-expressed HsLAMP1-mCherry with or without a C-terminal FRB* (LAMP-FRB* or LAMP, in red). Cells were treated with 0.05% ethanol or 250 nM rapalog for 2 h.
Only the tripartite complex of FKBP12-rapalog-FRB induced clustering of lysosomes at the MTOC (A). This clustering was selective for lysosomes because ER and
mitochondria (KDEL-GFP and MitoTracker Green) were not affected (C). (A, B) Phagocytosis of IgG-3-μm beads was reduced by immobilizing lysosomes at the
MTOC, $n = 80$–564 cells; ***$P < 0.001$.

D, E   (D) Activation of dynamin-2 with Ryngo-1-23 or expression of constitutively active dynamin-2 (S764A mutant, expression confirmed by the greyscale image)
restored phagocytosis of IgG-3-μm beads ($n = 34$–51 cells), but had no effect on the phagocytosis of IgG-6-μm beads, $n = 36$–72 cells (E); ***$P < 0.001$ EtOH Ctrl vs
rapalog Ctrl, ###$P < 0.001$ rapalog Ctrl vs rapalog Ryngo/Dyn2$^{S764A}$.

Data information: All scale bars = 5 μm. Bar charts presented as the mean ± S.E.M. and probability determined using one-way ANOVA.

dependent $Ca^{2+}$ nanodomains were still generated and bead
uptake proceeded normally.

These endo-lysosomal $Ca^{2+}$ nanodomains represent an extreme
form of $Ca^{2+}$ signal compartmentation because they do not appear
to spill-over to neighbouring channels (cf. plasma membrane chan-
nels (Bastian-Eugenio *et al*, 2019)). Thus, the G-GECO1.2 can only
detect endo-lysosomal $Ca^{2+}$ release when fused to the very channel
through which $Ca^{2+}$ is passing; with 3-μm beads, G-GECO1.2

fusions of channels that are not activated (the pore-dead TPC2
(D276K) or TRPML1) cannot detect $Ca^{2+}$ nanodomains of neigh-
bouring endogenous TPCs on the same vesicle (Figs 5 and 6)—at
least, not above the small cytosolic spill-over. The pore-dead mutant
TPC2(D276K) does not appear to act as a dominant-negative to
repress the endogenous TPCs because it does not mimic the Ned-19
or TPC-KO inhibition of the global $Ca^{2+}$ signal (Fig 2). The para-
digm of local $Ca^{2+}$ signalling in immune cells has precedents

(e.g. Nunes *et al*, 2012; Wolf *et al*, 2015; Diercks *et al*, 2018), and macrophages join cytotoxic T cells in requiring local NAADP/TPC $Ca^{2+}$ nanodomains (Davis *et al*, 2012).

### Different channel nanodomains drive different pathways

Extreme $Ca^{2+}$ domain compartmentation might therefore explain why different $Ca^{2+}$ channels on the same organelle (TPC2/TRPML1) can play different roles at phagocytosis: the TPC pathway couples to dynamin activation and is a *universal* requirement for all phagocytosis substrates tested (small beads, large beads, different bacteria—Figs 2 and 3), whereas TRPML1 is specialist, being activated by and required for large particles only (Figs 2, 6 and 8) (Samie *et al*, 2013; Dayam *et al*, 2015); large particles require more membrane to be enveloped and lysosomes provide the supplement, via their exocytotic fusion with the plasma membrane (Bajno *et al*, 2000; Braun *et al*, 2004; Czibener *et al*, 2006; Samie *et al*, 2013; Dayam *et al*, 2015), presumably in a Synaptotagmin-7 (Syn7)-dependent manner (Czibener *et al*, 2006; Samie *et al*, 2013). In this model, NAADP activation of TPCs occurs with all particle sizes, whereas $PI(3,5)P_2$ levels would only increase with large particles to additionally recruit TRPML1 (Samie *et al*, 2013).

How can such exquisite $Ca^{2+}$ nanodomains be segregated and differentially decoded? We first estimate the size of a $Ca^{2+}$ nanodomain from the selective inhibition by BAPTA over EGTA. The length constants of BAPTA and EGTA (see Materials and Methods) are approximately 7–15 nm and 220–500 nm, respectively, so the diameter of the $Ca^{2+}$ nanodomains must lie between these two ranges, which broadly agrees with 5- to 50-nm $Ca^{2+}$ domains at a channel mouth (Naraghi & Neher, 1997). As to inter-channel separation, the density of lysosomal channels on a macrophage lysosome is unclear when both higher (Freeman & Grinstein, 2018) and lower (Ruas *et al*, 2015a) expression levels of TPCs have been reported, but it could be as low as 20 channels per lysosome (Fameli *et al*, 2014). Assuming a lysosomal diameter of 0.5–2.0 μm and TPC diameters of 8 nm (She *et al*, 2019), even 500 TPCs would only occupy 0.4–6.4% of a single lysosome's surface area (see Materials and Methods). Therefore, there appears to be ample space to segregate TPC- and TRPML1-dependent $Ca^{2+}$ nanodomains even without considering surface $Ca^{2+}$-binding proteins that could buffer $Ca^{2+}$ and "ring-fence" nanodomains.

How these segregated $Ca^{2+}$ nanodomains are selectively and differentially decoded is unclear. We hypothesize that $Ca^{2+}$-dependent decoding molecules (calcineurin and Syn7) are intimately associated with their cognate channel—possibly in a protein complex—and selectively respond to the local high $Ca^{2+}$ nanodomains. Indeed, calcineurin can be tethered to locations via anchor proteins (Li *et al*, 2011) and Syn7 co-localizes with TRPML1 in large phagosomes (Samie *et al*, 2013). Such selective decoding would normally necessitate a low $Ca^{2+}$-affinity, and whilst calcineurin does indeed possess low-affinity $Ca^{2+}$-binding sites (Klee *et al*, 1998), it is less clear how high-affinity Syn7 might maintain unique fidelity with TRPML1. These hypotheses await formal confirmation.

### The trigger mode and $Ca^{2+}$ globalization

Small lysosomal $Ca^{2+}$ release secondarily recruits a large explosive $Ca^{2+}$ release from the ER via $Ca^{2+}$-induced $Ca^{2+}$ release (the "trigger hypothesis") which serves to amplify and globalize the initial, local

lysosomal trigger (Galione, 2019). That is, the endo-lysosomal $Ca^{2+}$ store is too small *per se* to *directly* contribute to global $Ca^{2+}$ spikes, so it contributes *indirectly* by recruiting the ER $Ca^{2+}$ store. This apparently occurs at phagocytosis, because the global $Ca^{2+}$ response is disproportionately reduced by inhibiting the small "invisible" lysosomal $Ca^{2+}$ release (by bafilomycin A1, luminal $Ca^{2+}$ chelation, TPC-KO, Ned-19; Figs 1 and 2). Additional evidence for a lysosome-ER dialogue at phagocytosis is that the normal response pattern of multiple $Ca^{2+}$ oscillations (reliant on successive rounds of triggering) becomes a single lysosomal spike when coupling to the ER is inhibited (Fig 5E and I).

Compared with other cell types that rely on the "trigger" mode of $Ca^{2+}$ signalling (Galione, 2019), macrophages are unusual in not requiring it for acute phagocytosis because inhibiting the secondary recruitment of the ER amplifier with EGTA/AM or CPA did not impact bead uptake (Fig 1E and H) or the lysosomal $Ca^{2+}$ nanodomain (Fig 5E and I). Nonetheless, the inhibition of phagocytosis bead uptake by Ned-19 appears to be mostly via its effects on local $Ca^{2+}$ nanodomains and not global signals because Ned-19 inhibited bead uptake even in cells where global signals were suppressed with EGTA/AM (Fig 6). We hypothesize that different modes of $Ca^{2+}$ signalling are employed for different aspects of macrophage activation: local $Ca^{2+}$ nanodomains (without amplification) acutely drive particle uptake, whereas the global, energetically expensive $Ca^{2+}$ signals (via $IP_3R$/Orai) must serve other roles at phagocytosis, e.g. stimulation of mitochondrial ATP synthesis, or longer term changes in cytokine secretion or gene expression.

### A new phagocytosis pathway to dynamin-2

Dynamin-2 is crucial for phagocytosis at multiple steps (Marie-Anais *et al*, 2016), but how FcγR engagement couples to dynamin activation has remained obscure; we suggest that NAADP/TPCs/endo-lysosomes bridge that transduction gap. In other contexts, the $Ca^{2+}$-dependent phosphatase, calcineurin, dephosphorylates and activates dynamin (Chircop *et al*, 2010, 2011); we now show that phagocytosis uses the NAADP/TPC pathway to rapidly (seconds) recruit calcineurin, as monitored with a live-cell FRET reporter (Fig 7), which is temporally incongruous with NFAT transcription, e.g. non-opsonic phagocytosis (Fric *et al*, 2014). Calcineurin activation appeared necessary for bead phagocytosis because two calcineurin inhibitors, FK506 and CsA, inhibited bead uptake (Fig 7). To the best of our knowledge, there are surprisingly no reports of these calcineurin inhibitors on the early (minutes) phase of phagocytosis. Inhibiting calcineurin repressed bead uptake by a comparable degree to TPC deletion/blockade and was non-additive with it, suggesting a common pathway (Fig 7). Conversely, activating calcineurin with ionomycin (Appendix Fig S6, likely because it generates high TPC2-G-GECO1.2 $Ca^{2+}$ domains) rescued the calcineurin defect in TPC-KO cells and restored phagocytosis (Fig 2). Finally, we argue that calcineurin is upstream of dynamin because conditions where calcineurin activation is repressed (by FK506 or TPC-KO or their combination) manifests as a failure to activate dynamin: these conditions are rescued by the dynamin activator, Ryngo-1-23 (Fig 7). This broadens the role of calcineurin in phagocytosis to being an acute $Ca^{2+}$-decoder, not just a activator of transcription.

Dynamin plays multifarious roles at phagocytosis, driving events that are both early (e.g. phagocytic cup formation) and late (e.g.

phagosome closure/scission) (Marie-Anais *et al*, 2016). Since phospho-dynamin is the inactive form (Chircop *et al*, 2011), we hypothesized that TPC/calcineurin led to dephosphorylation and activation. Thus, TPC-KO cells fail to phagocytose properly because they do not dephosphorylate and activate dynamin-2. First, phagocytosis could be restored in TPC-KOs by activating dynamin-2 (most strongly with a non-phosphorylated dynamin mutant-S764A; Fig 7B and C). Second, directly monitoring dynamin phosphorylation using immunoblots (Appendix Fig S7) or in-cell Westerns (Fig 7) revealed a rapid (within 1 min, Appendix Fig S7) phagocytosis-dependent dephosphorylation of dynamin-2 in WT cells; crucially, this dephosphorylation is TPC-dependent as it was eliminated in TPC-KO cells (Fig 7).

Although our data do not allow us to pinpoint which dynamin-dependent aspect(s) of phagocytosis are stimulated by TPCs, we add a new GTPase to the TPC signalling pathway (cf. Rab7 (Lin-Moshier *et al*, 2014)). Indeed, a TPC/dynamin axis may extend to other membrane engulfment events that exhibit a dual dependence upon TPCs and dynamin, e.g. EGF receptor endocytosis (Sousa *et al*, 2012; Grimm *et al*, 2014; Kilpatrick *et al*, 2017) and cholera toxin subunit B internalization (Lajoie *et al*, 2009; Ruas *et al*, 2014). It may also be clinically significant since human pathogens use dynamin-dependent endocytosis to initiate infection or deliver toxins (Harper *et al*, 2013).

### Lysosome signalling at phagocytosis

We introduce new signalling concepts to phagocytosis: first, that endo-lysosomes are early signal generators via the NAADP/TPC axis; second, this axis is a missing early pathway to dynamin

activation; third, that different endo-lysosomal $Ca^{2+}$-channel nanodomains play different roles. This extends the roles for this small crucial organelle in immune cells (Inpanathan & Botelho, 2019).

In the context of the literature, we posit the following model (Fig 9). When opsonized particles engage and activate the FcR, this early extracellular interaction presumably occurs *prior* to phagosome formation and yet the very first $Ca^{2+}$ signal is dependent on the NAADP/TPC axis (as are subsequent spikes—Figs 2 and 5; Appendix Fig S4). This rapid global $Ca^{2+}$ signal masks the local underlying essential lysosomal $Ca^{2+}$ nanodomains generated by TPCs. These nanodomains are observed across the lysosomal network (Appendix Fig S5), and not just at those lysosomes near the bead engagement site; this is not surprising given that the cytosolic messenger, NAADP, diffuses so rapidly (Churchill & Galione, 2000) that it would fill the entire cell in a second to widely activate TPCs.

We suggest that TPC $Ca^{2+}$-nanodomains rapidly stimulate calcineurin and thence dynamin-2. However, our data do not allow us to be definitive about the spatio-temporal choreography since FcR-mediated phagocytosis is a multi-step process with dynamin involved in early (e.g. phagocytic cup formation) and late (e.g. phagosome closure/scission) events (Marie-Anais *et al*, 2016).

Temporally, it is relevant that both early (focal exocytosis/pseudopodia formation (Samie *et al*, 2013)) and late (vesicle scission (Cabeza *et al*, 2010)) dynamin-dependent events may exhibit a co-reliance on $Ca^{2+}$ (albeit with some controversy (Di *et al*, 2003)). We speculate that the first, pre-phagosomal TPC $Ca^{2+}$ signal activates the early dynamin-dependent events (since dynamin

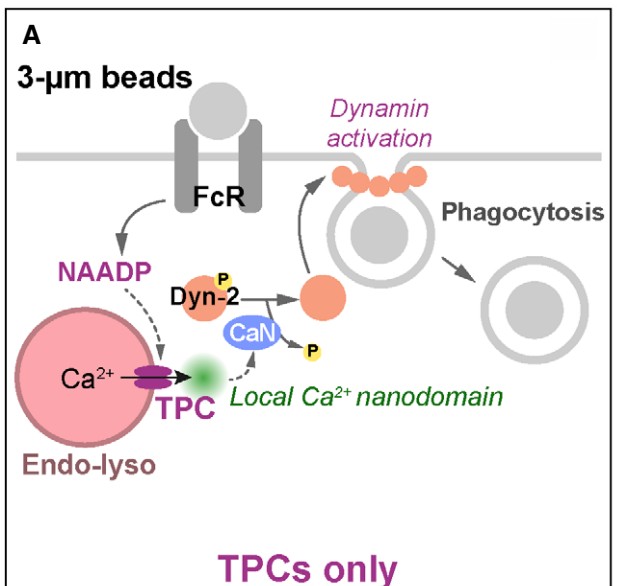
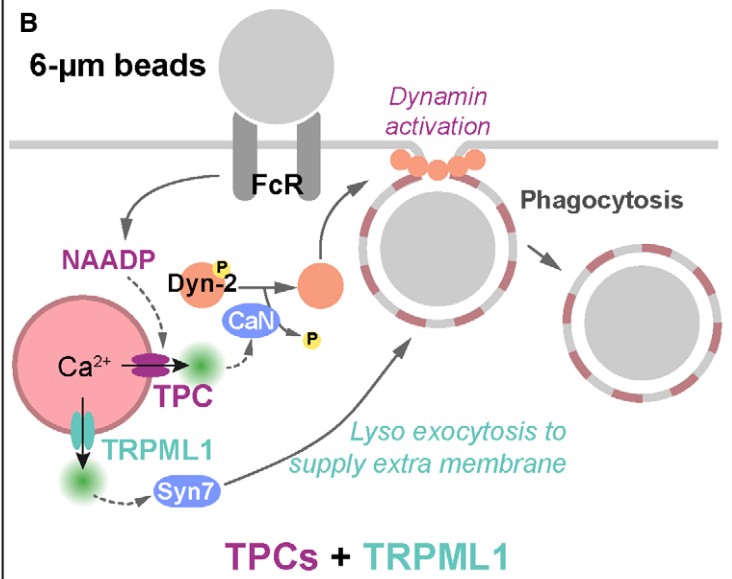

**Figure 9.** **Scheme depicting the multiple roles of endo-lysosomal $Ca^{2+}$ in phagocytosis.**

A   Engagement of the FcR by opsonized particles of multiple sizes and shapes (small or large beads, rod bacteria) commonly recruits the second messenger, NAADP, which activates TPC channels across the macrophage endo-lysosomal network. Local $Ca^{2+}$ nanodomains around TPCs activate the $Ca^{2+}$-dependent protein phosphatase, calcineurin (CaN), which dephosphorylates and thereby activates dynamin (Dyn-2). For clarity, only one dynamin-dependent event is shown (phagosome scission) but others are not excluded.

B   TRPML1 $Ca^{2+}$ nanodomains are only activated by large (6 μm) beads, a specialist secondary signal to additionally drive lysosomal exocytosis (via Synaptotagmin 7, Syn 7) and thereby provide supplementary lysosomal membrane (red-dashed lines) to the larger phagosome.

dephosphorylation can occur within 1 min) and that successive TPC $Ca^{2+}$ spikes sustain the later ones.

Spatially, a sub-population of lysosomes and/or TPCs rapidly appose the phagosome (Fig 4). The dual dependence of phagocytosis on TPC1 and TPC2 agrees with both apposing the phagosome, even though they are on different acidic vesicle populations. We hypothesize that TPC1 (broadly expressed throughout the endo-lysosomal system (Ruas *et al*, 2015a) operates on the early- and recycling-endosomes that regulate phagosome size (Flannagan *et al*, 2012), whereas TPC2 acts from its restricted late-endosome/lysosome locus. In terms of $Ca^{2+}$, lysosomes are rapidly recruited to release $Ca^{2+}$ across the entire cell (Appendix Fig S5) and not just around the site of bead engagement/phagosome formation.

If both TPCs and their $Ca^{2+}$ signals are observed across the cell, what is the role of the phagosomal apposition and TPC movement? First, we do not formally know whether the rapid apposition of phagosomes and lysosomes is truly a directed movement or merely a stochastic, chance interaction, but our data are consistent with a requirement for lysosomal movement: bead uptake is blocked by lysosomal immobilization (Fig 8). The simplest hypothesis would be that the nearest endo-lysosomes are the first to encounter the phagosome by their motility (whether by chance or design) to deliver $Ca^{2+}$ (and/or downstream signals) to where they are locally needed; the fact that distal lysosomes are activated is merely a safe-guarding "redundancy". Alternatively, the pan-activation of lysosomes across the cell could "crowd-source" the activation of cytosolic Dyn-2 which then vectorially migrates to the sites of phagocytosis (Marie-Anais *et al*, 2016). It is less clear how lysosomal movement would be required in the latter model, but we cannot exclude it.

With large beads, we envision that the above core TPC model still holds, but is extended to include the additional recruitment of lysosomal TRPML1 channels that drive lysosomal exocytosis and plasma membrane expansion to coat larger particles (Fig 9) (Samie *et al*, 2013; Sun *et al*, 2020). Given that focal exocytosis precedes bead uptake (Di *et al*, 2003), it is likely that this is a very early event.

Following phagosomal scission, lysosomes contribute to the maturation of the phagosome in multiple ways. First, TRPML1 drives phagosome-lysosome fusion (Dayam *et al*, 2015) (a role for TPCs is currently unknown). Second, lysosome fusion delivers the acidic and hydrolytic environment of the phagolysosome lumen (Flannagan *et al*, 2012), the "classical" terminal role of lysosomes in phagocytosis; consequently, TPCs appear in the phagosome proteome (Campbell-Valois *et al*, 2012; Dill *et al*, 2015; Guo *et al*, 2015), and a phagocytosis genome-wide CRISPR screen (Haney *et al*, 2018). Third, lysosomal $Ca^{2+}$ through TRPML1 activates transcription (via TFEB) to boost long term the phagocytic capacity of macrophages to degrade and kill pathogens (Gray *et al*, 2016).

Finally, engulfed pathogens can subvert the host machinery to prolong their own survival, and this can involve endo-lysosomes. Pathogens can suppress phagosome-lysosome fusion, and this may occur via perturbing endo-lysosomal $Ca^{2+}$ homeostasis, as in tuberculosis (Fineran *et al*, 2016). In view of our present work, it is possible that pathogenic disruption of lysosomal $Ca^{2+}$ could also reduce clearance by ablating the primary endo-lysosomal signal for phagocytosis. Therefore, TPC activators may prove a future therapeutic strategy for overcoming pathogenic block. During the revision of

this manuscript, a paper was published implicating TPCs in the resolution of phagosomes (Freeman *et al*, 2020) which complements our work and underscores the emerging importance of TPCs in immune surveillance.

## Materials and Methods

### Mice husbandry

Wild-type (WT) C57BL/6 mice, mouse lines carrying $Tpcn1^{tm1Dgen}$ ($Tpcn1^{-/-}$), $Tpcn2^{Gt(YHD437)Byg}$ ($Tpcn2^{-/-}$) and $Tpcn1^{tm1Dgen}/Tpcn2^{Gt(YHD437)Byg}$ ($Tpcn1/2^{-/-}$) mutant alleles (Ruas *et al*, 2015b) and $Mcoln1^{tm1Sasl}$ ($Mcoln1^{-/-}$) (Venugopal *et al*, 2007) mice (The Jackson Laboratory) were housed in the Biomedical Science Building (University of Oxford). Animal use was approved by the University of Oxford's Local Ethical Review Committee and was permitted by a UK Home Office licence in accordance with UK law (the Animals [Scientific Procedures] Act 1986).

### Cell culture and transfection

Bone marrow was extracted from the hind leg bones of male 12- to 16-week-old mice (at least 6 mice per genotype) and were plated in RPMI containing 20% v/v L929-conditioned medium, 10% v/v FCS, 2 mM glutamine, 100 U/ml penicillin and 100 μg/ml streptomycin. Adherent macrophage progenitors were allowed to differentiate for 7 days before use, at 37°C under 5% $CO_2$. RAW 264.7 murine macrophages were a kind gift from Frances Platt (University of Oxford) and were maintained in DMEM supplemented with 10% v/v FCS, 2 mM glutamine, 100 U/ml penicillin and 100 μg/ml streptomycin. BMDM and RAW 264.7 were transfected using JetPRIME (Polyplus transfection), in ratios of 1 μg of endotoxin-free plasmid DNA with 2 μl of JetPRIME, and incubated for 24–48 h before analysis.

### Plasmids

Constructs carrying cDNAs for mouse *Tpcn1* (GenBank BC058951), mouse *Tpcn2* (BC141195), human *Tpcn1* (BC136796), human *Tpcn2* (BC063008) and human TRPML1 (NM_020533) were all generated in-house: mouse TPC1-HA-TagRFP-T, mouse TPC2-HA-TagRFP-T, mouse TPC2(N257A)-HA-TagRFP-T, human TPC1-G-GECO1.2, human TPC2-G-GECO1.2, human TPC2(D276K)-G-GECO1.2. Mutant cDNAs were constructed by site-directed mutagenesis. Human TRPML1-G-GECO1.2 contained an additional C-terminal ER-export sequence (FCYENEV) to improve lysosomal targeting. The calcium-buffering protein constructs for rapalog-induced repositioning were designed in-house: FKBP12-(Calbindin-2)$_2$-mTagBFP2-HA utilized human calbindin-2 (GenBank BC015484) from the IMAGE clone 3847342; FKBP12-(Calbindin-2-EFmut)$_2$-mTagBFP2-HA was constructed using a DNA string GeneArt synthesis mutating all five functional EF-hands (12th [-Z] Glu to Gln in each). The triple mTagBFP2 construct (FKBP12-(mTagBFP2)$_3$-HA) was generated so that a similarly sized protein could also be repositioned in the cell, to account for any steric hindrances. The rapalog-FRB* (T2098L mutant for use with the rapalog AP21967, that does not inhibit mTOR) was targeted to the cytosolic face of lysosomes by fusion with human LAMP1 to produce LAMP1-mCherry-FRB*. The dynein-

binding protein for rapalog-induced organelle repositioning to MTOC (mTagBFP2-BicD2-FKBP12) was constructed in-house using pBa-tdTomato-flag-BicD2 594-FKBP (from Gary Banker and Marvin Bentley, Addgene plasmid #64205) (Bentley *et al*, 2015). To measure peri-lysosomal pH ratiometically, mCherry-SEpHluorin-LAMP1 was produced in-house, by inserting human LAMP1 into mCherry-SEpHluorin (from Sergio Grinstein, Addgene plasmid #32001 (Koivusalo *et al*, 2010)). The following plasmids were obtained from Addgene: CMV-G-GECO1.2 (#32446) and CMV-B-GECO1 (#32448) from Robert Campbell (Zhao *et al*, 2011), pcDNA3-Cyto-CaNAR2 from Jin Zhang (#64729) (Mehta *et al*, 2014), pGP-CMV-NES-jRGECO1a from Douglas Kim (#61563) (Dana *et al*, 2016), K44A HA-dynamin-2 pcDNA3.1 from Sandra Schmid (#34685) and HA-mDyn2 pcDNA3 from Pietro De Camilli (#36264) (Ferguson *et al*, 2007)—this plasmid was also used to produce HA-mDyn2 (S764A) pcDNA3 by site-directed mutagenesis (in-house). KDEL-GFP was a gift from Sergio Grinstein (University of Toronto).

## Intracellular Ca$^{2+}$ measurements

Cells were loaded with 2 μM Fura-2/AM, 2 μM Calbryte 520/AM or 2 μM Calbryte 590/AM in the presence of 0.03% w/v Pluronic F-127 in extracellular medium (ECM, mM: 121 NaCl, 5.4 KCl, 0.8 MgCl$_2$, 1.8 CaCl$_2$, 6 NaHCO$_3$, 25 HEPES, 10 Glucose) supplemented with essential amino acids, for 45–60 min at room temperature, followed by a 15 min de-esterification. Alternatively, cells were transfected with plasmids to express cytosolic GECIs or TPC-tethered GECIs. For experiments conducted in Ca$^{2+}$-free medium (−Ca$^{2+}_o$), cells were washed once with Ca$^{2+}$-free ECM supplemented with 1 mM EGTA, followed by two washes in Ca$^{2+}$-free ECM plus 100 μM EGTA and experiments conducted in this same medium. Unless otherwise stated, cytosolic Ca$^{2+}$ was chelated by loading BMDMs with 25 μM EGTA/AM or 25 μM BAPTA/AM for 45 min at room temperature in the presence of 0.03% w/v Pluronic F-127 in ECM, followed by a 15 min de-esterification.

Cells loaded with fura-2 were imaged using an Olympus IX71 microscope equipped with a 40× UApo/340 objective and excited alternately by 350- and 380-nm light using a Cairn monochromator; emission was collected at 480–540 nm. Autofluorescence was determined at the end of each run by addition of 1 μM ionomycin with 4 mM MnCl$_2$ to quench fura-2. Cells loaded with Calbryte 520 were imaged with ex 470 ± 20 nm, em 525 ± 25 nm, respectively. An image was collected every 2–3 s.

RAW 264.7 cells expressing TPC1/2-G-GECO1.2 and co-loaded with Calbryte 590 were imaged using a Nikon A1R laser-scanning confocal equipped with a Plan Apo VC 20× DIC N2 objective in resonant-scanner mode, with an image collected every 0.533 s. In channel series, G-GECO1.2 and Calbryte 590 were excited at 488 and 561 nm, and emission was centred at 525 and 595 nm, respectively. Experiments were conducted on a heated stage at approx. 32°C.

Images were analysed using custom-written Magipix software (Ron Jacob, King's College London, UK) on a single-cell basis.

## *In situ* calibration of G-GECO1.2

The $K_d$ for Ca$^{2+}$ of G-GECO1.2 proteins was determined *in situ* in permeabilized Cos-7 cells equilibrated with intracellular-like media (ICM) buffered to different free [Ca$^{2+}$]. The basic ICM was (mM):

10 NaCl, 140 KCl, 20 HEPES, pH 7.2 at room temperature. To this ICM, 5 mM of a Ca$^{2+}$ chelator was added ("Free"); to half of this chelator solution, 5 mM CaCl$_2$ was added ("Total") and all solutions were re-adjusted to pH 7.2 with KOH. Different free [Ca$^{2+}$] were generated by mixing the "Free" and "Total" solutions in different ratios, as calculated using Winmax chelator (Dr C. Patton, Stanford, USA). To generate a broad range of free [Ca$^{2+}$], it was necessary to use different Ca$^{2+}$ chelators with different Ca$^{2+}$ affinities (DiBr-BAPTA, BAPTA, HEDTA and nitrilotriacetic acid). Where possible, we overlapped the common free [Ca$^{2+}$] ranges of different chelators to minimize potential errors from inter-chelator variation.

Briefly, Cos-7 cells transiently expressing cytosolic G-GECO1.2 or human TPC2-G-GECO1.2 were washed 3× with an ICM at a fixed free [Ca$^{2+}$], and the final ICM addition was supplemented with 30 μM sulforhodamine B as a permeabilization marker (MW 559). Cells were imaged with standard green and red channels. The plasma membrane was discretely permeabilized with 60 μg/ml β-escin for ~ 5 min, breaching indicated by the entry of small MW marker, sulforhodamine B, while the large MW protein G-GECO1.2 was retained (even the cytosolic form). As extracellular Ca$^{2+}$ equilibrated, the GECI fluorescence reached a plateau and this value was normalized to the basal fluorescence prior to permeabilization ($F_0$). A plot of the free [Ca$^{2+}$] versus $F/F_0$ values was fitted to a sigmoidal dose–response (GraphPad Prism 5) to determine the *in situ* $K_d$ of G-GECO1.2 either cytosolic or fused to TPC2. The calculated Ca$^{2+}$ $K_d$ was as follows: 0.98 μM (95% confidence interval: 0.77–1.25 μM) for cyto-G-GECO1.2 and 0.81 μM (95% confidence interval: 0.64–1.04 μM) for human TPC2-G-GECO1.2.

## Opsonization and coupling of fluorophores to beads

50 μg of carboxylated silica beads (3.0 μm diameter) were incubated with 25 mg/ml cyanamide in PBS pH 7.2 for 45 min. 5 mg of fatty acid-free BSA and 1 mg of mouse IgG were added to the beads in 0.1 M sodium borate pH 8 (coupling buffer) and incubated with agitation for 6 h. The beads were washed in 250 mM glycine in PBS pH 7.2, followed by washing in coupling buffer. The beads were incubated with 1 μg Alexa Fluor 647 succinimidyl ester in coupling buffer for 1 h, followed by washing in 250 mM glycine in PBS pH 7.2 and stored at 4°C in PBS containing 2% sodium azide.

## Phagocytosis of beads

Macrophages were seeded onto ethanol-washed 16-mm diameter glass coverslips in 24-well cell culture dishes. Where specified, cells were pre-treated with 10 μM cytochalasin D for 15 min or 10 μM *trans*-Ned-19 for 30 min prior to addition of beads and remained present throughout the experiment. IgG-opsonized 3-μm silica beads covalently-coupled to Alexa Fluor 647 were added to each well (~ 1 × 10$^7$ beads/ml) in RPMI medium, and the plate centrifuged at 300 *g* for 1 min. Cells were incubated at 37°C to phagocytose, after which they were placed on ice to arrest phagocytosis and washed 3× with ice-cold PBS. Residual extracellular beads were labelled with 5 μg/ml Alexa Fluor 488-conjugated goat anti-mouse F(ab')$_2$ antibody for 5 min on ice and washed extensively. The cells were fixed with 4% w/v paraformaldehyde in PBS for 10 min at room temperature and coverslips mounted onto microscope slides using ProLong Gold. Cells were imaged using a Zeiss LSM510 Meta

confocal laser-scanning microscope equipped with a 63× objective, in multitrack mode, using the following excitation/emission parameters (nm): Alexa 647 (633/> 650) and Alexa 488 (488/505–530). At least 10 images from random areas of multiple coverslips were imaged. The number of beads internalized per cell was counted blindly to ensure unbiased assessment and at least 100 cells per genotype/treatment were counted.

### IgG-bead binding to the FcγR

To check FcγR binding capacity, BMDMs were seeded at $1\times10^5$ cells/well of a black-welled µclear flat-bottomed 96-well plate (Greiner Bio-one) the day before the experiment. Cells were pre-treated with 10 µM cytochalasin D for 15 min to prevent bead internalization, before addition of IgG- and Alexa Fluor 488-coupled 3-µm beads in extracellular medium (ECM mM: 121 NaCl, 5.4 KCl, 0.8 $MgCl_2$, 1.8 $CaCl_2$, 6 $NaHCO_3$, 25 HEPES, 10 Glucose) containing 10 µM cytochalasin D. The beads were allowed to bind for 15 min at 37°C, after which fluorescence was measured in a Novostar plate reader at ex 450 nm, em 520 nm. Cells were extensively washed (typically 8-times) with ECM containing 10 µM cytochalasin D in order to remove non-bound beads until the fluorescence reached a stable minimum value. Cellular autofluorescence was subtracted and bead binding expressed as the percentage of relative fluorescence units of WT BMDM.

### Phagocytosis of bacteria

*Mycobacterium smegmatis* (mc2155 strain expressing mCherry) and *BCG* (Pasteur strain expressing mCherry) were provided by Paul Fineran and Frances Platt (University of Oxford). Mycobacteria were grown on 7H11 agar plates (supplemented with oleic albumin dextrose catalase) before transfer to 7H9 liquid media (supplemented with albumin dextrose catalase) and grown to log phase ($OD_{600\ nm}$ 0.6–1.0) at 37°C. BMDMs were plated in a 24-well plate at $1 \times 10^5$ cells/well the day before analysis. Cells were washed twice with RPMI containing 10% v/v FCS (without antibiotics) before infection with mycobacterium at a multiplicity of infection of 10 for 2 h at 37°C. Afterwards, the cells were washed with fresh medium and treated with 200 µg/ml amikacin for 1 h to kill non-engulfed bacteria. Following more washes, BMDMs were lysed in sterile milli-Q $H_2O$ for 10 min at 37°C and plated onto 7H11 agar plates (containing 50 µg/ml kanamycin to select for mCherry-expressing mycobacterium) at different dilutions using the pour-plate technique. The plates were incubated for 6 days at 37°C, and images digitized on a flat-bed scanner and colony-forming units were counted. Alternatively, cells in 96-well plates were infected with Mycobacteria and washed (as above). Internalized mCherry-expressing bacteria detected using a Novostar plate reader (BMG LABTECH) using excitation/emission 570/620 nm.

### Immunofluorescence staining

Cells were fixed in 4% w/v paraformaldehyde in PBS and permeabilized/blocked with 0.1% w/v saponin/5% w/v goat serum in PBS. Antibody incubations were performed in PBS/0.01% w/v saponin/5% w/v goat serum. The primary antibody anti-HA (rat monoclonal 3F10; Roche) was used at 100 ng/ml, and a goat anti-IgG conjugate

of Alexa 546 was used as the secondary antibody. Cells were viewed on a Zeiss 510 Meta confocal microscope using excitation 543 nm and > 560 nm emission parameters.

### Buffering peri-lysosomal $Ca^{2+}$

RAW 264.7 macrophages in CellView 4-compartment glass-bottom dishes (Greiner Bio-One) were transfected with 0.3 µg HsLAMP1-mCherry-FRB* together with 0.8 µg (i) FKBP12-(Calbindin-2)2-mTagBFP2-HA or (ii) FKBP12-(Calbindin-2-EFmut)2-mTagBFP2-HA or (iii) FKBP12-(mTagBFP2)3-HA, per compartment using jetPRIME and allowed to express for 24 h. Cells were treated with the rapalog 250 nM AP21967 (Takara) to cause dimerization of FRB* and FKBP12 proteins, or 0.05% v/v ethanol (control) for 45 min at 37°C in DMEM. Cells were washed once with $Ca^{2+}$-free ECM supplemented with 1 mM EGTA, followed by two washes in $Ca^{2+}$-free ECM with 100 µM EGTA. Phagocytosis of Alexa 647-conjugated 3 µm-IgG-beads was conducted in $Ca^{2+}$-free ECM supplemented with 100 µM EGTA and 250 nM AP21967 for 10 min at 37°C.

### Intracellular calcineurin activity measurements

RAW 264.7 macrophages were transfected with a FRET reporter for calcineurin activity, CaNAR2, and the genetically encoded $Ca^{2+}$ indicator, jRGECO1a, and imaged 48 h later. Cells were pre-treated with 10 µM FK506 or 10 µM *trans*-Ned-19 for 30 min at room temperature in ECM, before addition of 2 µM ionomycin or mouse IgG-coupled 3-µm beads. Images were acquired on a Zeiss 510 Meta confocal microscope using excitation 543 and > 560 nm emission parameters for detection of jRGECO1a. Dual Cerulean3/YPet emission ratio imaging was performed using excitation 458 nm and emission measured at 475–525 nm for Cerulean3 and 530–600 nm for YPet. jRGECO1a fluorescence changes were normalized to initial fluorescence ($\Delta F/F_0$) and YPet/Cerulean3 (Y/C) emission ratio changes (FRET) from CaNAR2 ($\Delta Y/C$).

### In-cell western assay for detection of dynamin-2

BMDM were seeded into a 96-well U-bottom, suspension culture plate (Greiner Bio-one). FcγR-mediated phagocytosis of rabbit IgG-3 µm beads proceeded at 37°C for various times. Phagocytosis was arrested by placing the plate on ice and the cells washed 3× with ice-cold PBS. Cells were fixed and permeabilized with ice-cold methanol for 10 min at RT. Following block in Odyssey® blocking buffer (LI-COR), cells were incubated with dynamin-2 (G-4) mouse monoclonal (Santa Cruz) and dynamin [pSer778] sheep polyclonal (Novus Biologicals). Anti-dynamin [pSer778] is a phospho-specific antibody to phospho-Ser 778 in dynamin-1 (Tan *et al*, 2003). Ser-778 is conserved in the dynamin-2 phospho-box at position Ser-764 (Chircop *et al*, 2011). Dual-immunolabelling was detected using IRDye® 680RD donkey anti-mouse IgG and IRDye® 800CW donkey anti-goat IgG polyclonals (LI-COR) and scanned using an Odyssey infrared scanner in the 700 and 800 nm channels.

### Western blotting

Total-protein lysates were prepared by solubilizing cells in RIPA buffer (150 mM NaCl, 5 mM EDTA, 50 mM Tris pH 8, 1% v/v

NP40, 0.5% w/v sodium deoxycholate, 0.1% w/v SDS) supplemented with 1× Halt protease and phosphatase inhibitors (Thermo Scientific). For protein phosphatase treatment, a pellet of WT BMDM was resuspended in NEBuffer (New England BioLabs) and sonicated on ice. Lambda protein phosphatase (100–400 U) and MnCl$_2$ (0.75 mM) were added to the lysates and incubated at 30°C for 1 h before Western blotting. Proteins were resolved by SDS–PAGE (50 μg/lane) using 4–12% TruPAGE gels (Sigma) and transferred to PVDF membranes. Membranes were blocked in Odyssey® blocking buffer (LI-COR) and incubated with dynamin-2 (G-4) monoclonal (Santa Cruz), HA (3F10) monoclonal (Roche), dynamin [pSer778] (Novus Biologicals) or actin (JLA20) monoclonal (DSHB). Secondary antibodies were anti-mouse, anti-rat and anti-goat conjugated to IRDyes® (LI-COR). Immunoreactive bands were detected by fluorescence using an Odyssey infrared scanner in the 700 and 800 nm channels.

### Lysosomal recruitment to the MTOC

RAW 264.7 macrophages were transfected with LAMP1-mCherry-FRB* and the dynein-binding protein for CID-mediated repositioning to the MTOC (mTagBFP2-BicD2-FKBP12). Cells were treated with 250 nM AP21967 (Takara) or 0.05% ethanol (control) for 45 min at 37°C in DMEM, washed once with Ca$^{2+}$-free ECM supplemented with 1 mM EGTA, followed by two washes in Ca$^{2+}$-free ECM with 100 μM EGTA and phagocytosis of Alexa 647-conjugated 3 μm-IgG-beads conducted in this same medium.

### Statistical analysis

Each experimental condition was performed on at least three preparations on multiple days. Data are presented as mean ± S.E.M. and analysed by either Student's *t*-test (unpaired) for two conditions, or a one-way ANOVA for multiple conditions with significance determined as $P < 0.05$ using GraphPad Prism or Instat programs. Imaging data are analysed as pooled single-cell values. Normality was tested using a Kolmogorov–Smirnov test (Instat). Normally distributed data were analysed using a Tukey–Kramer post-test, otherwise a non-parametric post-test (Kruskal–Wallis) was applied. Graphs were annotated using the following conventions: $P > 0.05$ (ns), $P < 0.05$ (*), $P < 0.01$ (**) and $P < 0.001$ (***).

### Calculations

Length constants for EGTA and BAPTA were calculated using the equations and variables in (Kidd *et al*, 1999) except using a Ca$^{2+}$ diffusion coefficient of 233 μm$^2$/s (Kasai & Petersen, 1994). We assumed a 20–100-fold concentrating of extracellular chelator/AM so that the cytosolic concentration rose to the millimolar range (1–5 mM). This gave length constants of 221–494 nm (EGTA) and 7–15 nm (BAPTA) over this concentration range.

To estimate the relative space occupied by TPCs on a vesicle, we first calculated the surface area (SA) of the vesicle sphere ($4\pi r^2$) from a diameter of 0.5–2.0 μm (785,398–12,566,371 nm$^2$). An 8-nm diameter TPC (She *et al*, 2019) was assumed to be circular ($\pi r^2$) and occupy an area of ~ 100 nm$^2$. Therefore, 500 channels occupy 50,265 nm$^2$ which is 0.4–6.4% of the total vesicular SA.

## Data availability

The authors declare that there are no primary datasets and computer codes associated with this study.

**Expanded View** for this article is available online.

### Acknowledgements
We thank Margarida Ruas (University of Oxford) for genotyping and for helpful discussions. We thank Frances Platt and Paul Fineran (University of Oxford) for the kind gifts of the RAW 264.7 cell line and the cultures of *M. smegmatis* and *BCG* and Siamon Gordon (University of Oxford) for useful discussions. We thank the Oxford Biomedical Science Building staff for mice housing and care. This work was supported by The Wellcome Trust (102828/Z/13/Z).

### Author contributions
Conceptualization: LCD, AJM, AG; Methodology: LCD, AJM; Investigation: LCD, AJM; Writing: LCD, AJM, AG; Funding acquisition: AG.

### Conflict of interest
The authors declare that they have no conflict of interest.

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
