## [Review Process File · The EMBO Journal]

NAADP-regulated two-pore channels drive phagocytosis through endolysosomal Ca²⁺ nanodomains, calcineurin and dynamin

Lianne Davis, Anthony Morgan, and Antony Galione
DOI: [10.15252/embj.2019104058](https://doi.org/10.15252/embj.2019104058)

Review Timeline:

Submission Date:	19th Nov 19
Editorial Decision:	13th Dec 19
Revision Received:	20th Mar 20
Editorial Decision:	17th Apr 20
Revision Received:	22nd Apr 20
Accepted:	29th Apr 20

Editor: Elisabetta Argenzio

Transaction Report:

Thank you for submitting your manuscript entitled "NAADP/TPCs drive phagocytosis through endo-lysosomal Ca²⁺ nanodomains, calcineurin and dynamin" [EMBOJ-2019-104058] to The EMBO Journal. Your study has been sent to three referees for evaluation, whose reviews are enclosed below.

As you can see, the referees concur with us on the overall interest of your findings. However, they also raise several critical points that have to be addressed before they can support the publication of your work in The EMBO Journal. In particular, referee #1 stresses that both the presence of lysosomes near the sites of phagosome formation and that these phagosomes are those that release calcium have to be tested. In addition, this reviewer requests you to show that Fc-receptor stimulation generates NAADP locally and to examine the effects of Bafilomycin on phagocytosis. Referee #2 asks to investigate the role of TRPML1 and TPCs in generating Ca²⁺ nanodomain and in phagocytosis. Also, the referees point out inconsistencies with the existing literature that you should clarify and/or discuss.

Given the overall interest of your study, I would like to invite you to revise the manuscript in response to the referee reports. I should note that conclusively addressing these and all the other referees' points is essential for publication in The EMBO Journal.

When preparing your letter of response to the referees' comments, bear in mind that this will form part of the Review Process File and will be available online to the community. For more details on our Transparent Editorial Process, please visit our website:
http://emboj.embopress.org/about#Transparent_Process.

We generally grant three months as standard revision time. As a matter of policy, competing manuscripts published during this period will not negatively impact on our assessment of the conceptual advance presented by your study. Nevertheless, please contact me as soon as possible upon publication of any related work.

I thank you again for the opportunity to consider this study for publication and will be happy to answer any questions about the submission of the revised manuscript to The EMBO Journal. I look forward to your revision.

Referee #1:

The manuscript by Davis et al describes an unappreciated role for lysosomes in phagocytosis. Specifically, the authors propose that NAADP generated during activation of Fc receptors stimulates the release of calcium from lysosomes, which in turn activates calcineurin to dephosphorylate and activate dynamin, that is seemingly required for phagosome sealing. The authors used an array of elegant, powerful techniques to document most of the preceding steps in the proposed sequence.

While the results and conclusions are novel, there are some important gaps that need to be bridged

and, more importantly, there are a large number of apparent internal inconsistencies or inconsistencies with data published by others. These need to be addressed.

-there is no evidence that lysosomes are present in the vicinity of forming phagosomes, so that a very localized perilyosomal calcium elevation stimulates dynamin locally. The authors need to show that lysosomes are present near sites of phagosome formation and that only those phagosomes and not distant ones release calcium.

-why does calcium released locally by ER and entering the cell via Orai1 not suffice to activate calcineurin, when the contribution of these pathways is greater and they are unquestionably activated locally at sites of phagocytosis by well established mechanisms. This is particularly puzzling because addition of ionomycin does restore phagocytosis in TPC KO cells.

-what is the evidence that Fc receptor stimulation, whether via CD38 or by other means generates NAADP locally? Without this evidence, the model is not convincing.

-the authors state that they used dextran Cal520 because "there are no drugs for directly depleting vesicle calcium content". Elsewhere they use bafilomycin to deplete the vesicular calcium selectively. These statements are incompatible. Does bafilomycin inhibit phagocytosis? Multiple groups have used this inhibitor but have not reported effects on phagocytosis.

-that the Kd of dextran Cal520 at neutral pH is 320 nM is irrelevant; what is its Kd at the acidic pH prevailing in lysosomes? If, as expected, it is much higher, it may be ineffective. Even if able to bind calcium at the acidic pH, why does the continuous uptake of calcium, whether by Ca/H exchange or via endocytosis, saturate its binding sites?

-TPC channels are thought to be activated by PI(3,5)P2, which is present constitutively in late endosomes and lysosomes. It is not clear how calcium would be retained under these conditions and its release stimulated by NAADP. The authors do not discuss the regulation of TPCs by phospholipids.

-Figures 1 and 2 seem internally inconsistent. In Fig 1 no calcium changes are measurable after thapsigargin, implying that the contribution of lysosomes is barely measurable, yet Ned19 causes big changes in Fig 2, as does the elimination of TPC1 or TPC2.

-the authors report that TPC1 and TPC2 both colocalize to a similar extent with LysoTracker-positive organelles. This seems inconsistent with a number of papers from others in the literature and even with their own data. The extent of colocalization of TPC1 with LysoTracker in Fig EV3 seems modest (and could be entirely fortuitous) and in Fig EV4 the colocalization of TPC1 with LAMP is negligible. These findings are more compatible with the published findings that TPC1 is in a Rab5-positive early compartment that would not be LysoTracker positive.

-if TPC1 and TPC2 are in fact in different compartments, it is difficult to envisage how knocking out either one causes a $\approx 75\%$ reduction calcium changes. It would also require both early and late endosomes or lysosomes to be present at sites of phagocytosis.

-the authors report that TPC1 and TPC2 surround the lysosome. However, in one case (TPC2) the channels line the phagosomal membrane, while in the other (TPC1), vesicles cluster in the vicinity. Discussing both phenomena in the same context is inappropriate. Moreover, it is not at all clear how TPC2 insertion into a sealed, presumably late phagosome is relevant to the proposed role of the channel in forming the phagosome.

-Figure 4 is most puzzling. Why are the calcium oscillations measured in the cytosol perfectly synchronized with perilyosomal changes, when the cytosolic changes are demonstrated earlier to be attributable to the ER and Orai1? And why does Ned19 dampen the cytosolic oscillations by 50% when the contribution of TPCs is shown earlier to be too small to detect? And if TPC1 and TPC2 are in fact in different compartments, how are their oscillations synchronized?

-how can calcium oscillations persist in cells loaded with dextran Cal520 in Fig 1, yet be almost eliminated by Ned19 in Fig 4?

-the inability of TPC-GG to detect the perilyosomal calcium when released by a neighboring channel is unprecedented. What accounts for this extreme behavior?

-in Fig 5d, why is dynamin exclusively localized to one bead (which seems to have been already internalized) while it is missing from most other beads, including ones that are nearer the membrane and in the process of being ingested, judging from the actin staining?

-why does activation of dynamin for phagocytosis require calcium-induced dephosphorylation, while all other dynamin dependent processes, like endocytosis, proceed in the absence of calcium changes?

-cyclosporine and FK506 had been used in experiments involving phagocytosis without any reported inhibitory effects (e.g. Cell 2007, 130:37). Why the apparent discrepancy?

-where is calcineurin in cells at rest and why doesn't the much larger and more localized calcium concentration change produced by release from ER and Orai1 suffice to activate it?

-how can 60% of the phosphodynamin be selectively dephosphorylated by calcineurin activated locally by peri-lysosomal calcium?

-how can calcineurin remain phosphorylated to a level that is >100% of the control in TPC1 KO cells, yet the cells phagocytose with 50% efficiency? Does this not imply that dephosphorylation is not essential for phagocytosis? And why is there no dephosphorylation caused by TPC2 present in the TPC1 KO? If TPC1 and 2 are present in the same compartment, shouldn't calcium be released via TPC2?

-why does phosphodynamin continue to decrease gradually over 20 min, if the calcium transient are over within a minute or two? It is essential to establish what fraction of the dynamin is phosphorylated prior to stimulation, since a rather small (<10%) change occurs within the period required for phagocytosis. Not all the dynamin could be inactive prior to phagocytosis, to the extent that endocytosis is ongoing.

Referee #2:

In this important manuscript the authors describe a new mechanism by which the lysosomes drive phagocytosis. The authors provide ample and convincing evidence to conclude that stimulation of the macrophages Fc receptor releases Ca²⁺ from the endolysosomal organelles through the NAADP-activated TPC channels to generate a near lysosomal Ca²⁺ signal that is independent of global Ca²⁺ homeostasis and release. This Ca²⁺ nanodomain activated calcineurin that

dephosphorylates dynamin. The dephosphorylated dynamin pinch to internalized the engulfed particle, including pathogens, independent of their size.

The novel findings substantially add to understanding the role of the lysosomes in phagocytosis and understanding this fundamental biological process. However, I have several fairly minor comments.

An issue that is not fully resolved is the role of TRPML1 and the TPCs in generating the nanodomain Ca^{2+} signal and in phagocytosis. The two channels are expected to reside in the same organelles and perhaps share the same Ca^{2+} pool. To resolve this issue, in the discussion the authors suggest two different roles for the channels that maybe mediated by neighboring organelles. This should be tested more directly using the available pharmacological activators and inhibitors of TRPML1. The authors should test if prior treatment of the cells with TRPML1 activators and inhibitors affects phagocytosis stimulated by FcR.

Another clarification with regard to stimulation of the FcR is involvement of NAADP in its function. The crucial requirement for Ca^{2+} in the lysosomal nanodomain can be further demonstrated by showing that stimulation with ATP does not affect phagocytosis.

There is significant overlap between the results and discussion sections, and both are very long and can be reduced by 20-30% without losing clarity of conveying the significance of the story.

Referee #3:

This paper reports extensive findings that demonstrate a key role for Ca nanodomains generated by endo-lysosomal two-pore channels (TPCs) in driving phagocytosis by macrophages. This mechanism is novel; helps resolve conflicting reports in the literature; and, as phrased the authors "promotes endo-lysosomes from downstream effectors to upstream initiators ... of phagocytosis". The findings will be highly relevant to studies of phagocytosis and endocytosis in multiple cell types. The paper is a pleasure to read, and guides the reader through a complex sequence of experiments with an elegant, expository writing style that is becoming sadly rare in the current scientific literature.

The authors employ a variety of complementary techniques and experimental systems to demonstrate that FcγR-induced phagocytosis is regulated by the release of endo-lysosomal Ca^{2+} through TPCs including: buffering of intracellular Ca^{2+} by exogenously loaded Ca^{2+} buffers with different binding kinetics and affinities; buffering of endo-lysosomal Ca^{2+} by passive uptake of a dextran-conjugated Ca^{2+} indicator and by recombinant expression of calcium binding protein targeted to the outer endo-lysosomal membrane; the use of cells from TPC1 and TPC2 knockout mice; inhibiting TPC activity with Ned-19; and generation of tethered TPC-GEC1 Ca^{2+} reporters. The authors further extend their investigation to identify downstream proteins dynamin and calcineurin as critical players regulating phagocytosis in BMDMs.

Although caveats can be raised to some approaches (e.g. inference of Ca microdomains from comparative actions of BAPTA vs EGTA; 'bleed-through' of global Ca to the tethered Ca probe; and issues raised below), taken together the authors' multi-pronged approach, utilizing diverse methodologies to address each step in the phagocytosis pathway, presents compelling evidence to

support their conclusions.

We present the following comments as suggestions for further improvement of the paper, not as essential requirements for new work before the paper could be published.

Substantive comments

1. In Fig 2C, phagocytosis of 3 μm beads in Tpcn1^{-/-} or Tpcn2^{-/-} BMDM cells was significantly reduced as compared to WT cells, but substantial phagocytosis still remained (20% and 35% of WT respectively at 40 min). It would be beneficial to test FcyR-induced phagocytosis in Tpcn1/Tpcn2 double knockout cells. This should be relatively straightforward since the authors have previously shown that TPCs are required for NAADP-induced Ca²⁺ signaling because BMDMs lacking TPCs fail to evoke Ca²⁺ signals in response to NAADP (Ruas et al., 2015a) .

2. In Fig 4a-k, much effort was taken to characterize Ca²⁺ signals detected by G-GECO1.2 tethered to TPCs in order to differentiate local Ca²⁺ signals arising from the release of Ca²⁺ from endo-lysosomes from 'bleed-through' of global Ca²⁺ signals. A more direct approach would be to deplete ER Ca²⁺ stores (with TG or CPA) and then invoke phagocytosis by FcyR activation as shown in Fig 1G. Under this scenario, the obfuscating global Ca²⁺ signals caused by ER Ca²⁺ release would be abolished and the local Ca²⁺ signals arising from lysosomal Ca²⁺ release, which as the authors contend are essential to phagocytosis, should be retained and be more specifically resolved by their GECI-TPC reporter. This would further strengthen the conclusion that endo-lysosomal Ca²⁺ nanodomains drive phagocytosis.

3. Although the experimental evidence strongly points to endo-lysosomal Ca²⁺ nanodomains initiating phagocytosis by activating calcinurin, it is difficult to envision how this mechanism may work. How is it that a small and exquisitely confined Ca²⁺ domain (apparently localized to individual TPCs) can selectively activate calcinurin whereas a much larger global Ca²⁺ elevation evoked by release through IP3Rs does not? The notion of Ca²⁺ 'nanodomains' is typically considered in situations where Ca²⁺ channels are located at contact sites that hold them in immediate apposition to an effector or transporter (e.g. Ca²⁺ transfer from ER to mitochondria), but how could that be the case here? The authors suggest that the Ca²⁺ signals are restricted to each forming phagosome, and present evidence that movement of lysosomes toward the phagosome is required for phagocytosis. But if that is what underlies the spatial localization of the signaling it would seem the movement must occur before the transmission of the Ca²⁺ signal?

It would be interesting if the authors could present their thoughts on these questions.

Minor comments

1. Referencing Fig 2G, the authors state "The redistribution of TPCs from endo-lysosomes suggests that the acidic Ca²⁺ stores themselves rearrange, concentrating around and close to the phagosome..." but do not provide evidence to support this statement, i.e. What was the distribution of TPCs prior to phagocytosis and how was this redistribution quantified?

2. The finding that ER Ca²⁺ release by UTP produces a small signal in the GECI-TPC reporter not greatly different than the signal detected by untethered cytosolic GECI (Fig 4I,J), which the authors attribute to a 'bystander effect', is opposite to the findings of Atakpa, Thillaiappan et al, 2018 (cited by the authors; page 13) who, using the same G-GECO1.2 (albeit tethered to a different lysosomal protein) reported IP3R-mediated Ca²⁺ release selectively delivers Ca²⁺ to lysosomes resulting in

much larger signals in the tethered vs untethered GECI. Some discussion as to the cause of these differences would be beneficial.

3. All n numbers are reported as the number of cells analyzed and data in bar and point graphs are presented as mean {plus minus} sem per Experimental Methods. Please indicate the number of times each experiment was performed for each experimental condition tested and clarify whether statistics were performed on the number of cells analyzed or on the number of experimental repetitions (the mean of the means). Also, indicate in the Experimental Methods the statistical test/s used to determine if the data are normally distributed (an underlying assumption when applying student's t test or ANOVA).

4. Check scale bars on image panels. Many panels showing 3 μm diameter beads have the beads appear considerably larger than that as judged by the scale bars. (e.g. Fig 1I, Fig 2G, etc).

5. The numerical y-axes on several graphs are simple labeled "Phagocytosis" (e.g. Fig 1H,K, Fig 2B,H,I, ect.). Clarify what the units on the axis represent: e.g. if it is bead uptake per cell, explicitly state in figure or figure legend.

6. Page 15, top. The statement that the D276k mutant is not acting as a dominant negative is difficult to understand at this point in the text without further explanation; although this is clearly elucidated in the Discussion.

7. Page 29, discussion of 'trigger' hypothesis of amplified Ca^{2+} release through IP3Rs. Although the global Ca^{2+} signal was blocked by EGTA, this does not eliminate the possibility of coupling via a Ca^{2+} nanodomain if TPCs and IP3Rs were in close apposition.

8. Page 20. Heading "Fc γ R-mediated phagocytosis induces TPC-dependent dephosphorylation of dynamin". Reword, as it is the dephosphorylation of dynamin that activates phagocytosis.

Our Responses to Refs

We thank all three referees for their helpful remarks which have improved the manuscript. We shall deal with each referee in turn.

Referee #1

The manuscript by Davis et al describes an unappreciated role for lysosomes in phagocytosis. Specifically, the authors propose that NAADP generated during activation of Fc receptors stimulates the release of calcium from lysosomes, which in turn activates calcineurin to dephosphorylate and activate dynamin that is seemingly required for phagosome sealing. The authors used an array of elegant, powerful techniques to document most of the preceding steps in the proposed sequence.

While the results and conclusions are novel, there are some important gaps that need to be bridged and, more importantly, there are a large number of apparent internal inconsistencies or inconsistencies with data published by others. These need to be addressed.

(1) there is no evidence that lysosomes are present in the vicinity of forming phagosomes, so that a very localized perilyosomal calcium elevation stimulates dynamin locally. The authors need to show that lysosomes are present near sites of phagosome formation and that only those phagosomes and not distant ones release calcium.

Phagosome Contact

We thank the referee for the suggestion. We include new imaging data at higher spatial resolution. This shows that lysosomes make rapid contact with beads (81 ± 21 s after the first Ca^{2+} signal).

Incidentally, this extends decades of work showing that lysosomes move towards (and ultimately fuse with) the phagosome, although we now show that this juxtaposition occurs very rapidly.

Distal, Proximal, Local

(We presume that the referee means “only those *phagosomes...*” rather than *lysosomes*.)

We feel that the referee has made assumptions that our model neither invokes nor requires. If we understand correctly, the referee envisions that *only* those lysosomes that are proximal to the bead generate a Ca^{2+} signal (and a concomitant, local activation of dynamin). However, we do not invoke this, it is not required for, nor fits in with the sequence of events:

- (i) The initial Ca^{2+} signal is generated upon FcR engagement, when the beads are still *extracellular*; therefore, TPCs are activated (**Fig.2,4**) and contribute to the Ca^{2+} spike (**Fig.2**) *before phagosome formation*.
- (ii) As we argue below (Point 3), NAADP will diffuse globally within seconds of receptor engagement (even if produced by a point source). Therefore, selective activation of lysosomes near the phagosome is *not* to be expected; lysosomes across the entire cell will be activated. Confirming this, we have now imaged cells with higher spatial resolution and this shows that **lysosomes are activated across the cell, both proximal and distal to bead engagement sites (Fig.EV6)**.
- (iii) Note: although lysosomes across the entire cell are activated, the Ca^{2+} signal of each channel is, by definition, local. In our model, these collective local signals activate CaN (and then Dynamin).

- (iv) How these local Ca^{2+} signals are *functionally transferred* to the phagosomes is currently unknown. Indeed, **each of the following different models is compatible with our data, but does not require the model proposed by the referee:**
- All lysosomes are activated, but only those proximal to the phagosome are used (*'redundancy'*).
 - A protein (e.g. CaN and/or dynamin) that associates with lysosomes (transiently or chronically) is released in an activated form from the lysosomes and diffuses to phagosomes (*'crowd sourcing'*). Certainly, dynamin moves to and accumulates at the sites of phagosome closure [1], but it does not mean (or require) that these are the *only* sites of Dynamin activation.
- (v) Furthermore, as we have already stated in the manuscript, dynamin plays several roles in phagocytosis and our data do not allow us to connect the lysosomal signals to any particular downstream dynamin-dependent event(s), be it scission or actin rearrangement.

In summary, thank the referee for requesting the additional experiments which we include, but these do not agree with the referee's model of polarized lysosomal activation.

Further, considerable work will be required to elucidate how the local Ca^{2+} /CaN/dynamin axis is integrated, but our basic conclusions hold.

We have also included an explicit hypothesis at the end of the Discussion to help orientate the reader.

(2) why does calcium released locally by ER and entering the cell via Orai1 not suffice to activate calcineurin, when the contribution of these pathways is greater and they are unquestionably activated locally at sites of phagocytosis by well established mechanisms. This is particularly puzzling because addition of ionomycin does restore phagocytosis in TPC KO cells.

We do not agree that they are 'unquestionably activated and locally greater' in MØ. First, we are not alone in showing that Ca^{2+} influx is dispensable for phagocytosis (reviewed in [2]). Our own data are very clear: eliminating Ca^{2+} influx or the global ER Ca^{2+} response has absolutely no impact on phagocytosis.

To reinforce that IP_3 /Orai do not couple to CaN and phagocytosis in these cells, we include **new data** using a physiological agonist (**UTP**) that only couples to IP_3 /Orai (and not to the lysosome – Fig. 4). There are two points:

1. UTP evokes a global Ca^{2+} signal but does *not* activate CaN like ionomycin does (see below, $n = 86-96$ cells). This shows that IP_3/Orai is not an efficient stimulus of CaN in these cells.

2. UTP does not rescue bead uptake in TPC KO (new Fig. 2). This again shows that IP_3/Orai cannot support phagocytosis.

We do not feel that data are at all ‘puzzling’ because the Ca^{2+} signalling field has long accepted the idea that different Ca^{2+} sources are ‘non-equivalent’ i.e. they can uniquely ‘pair-up’ with their own effectors (signal compartmentation), as exemplified by different NFAT isoforms [3-6], mitochondria [7], Ca^{2+} -activated K^+ channels [8,9]. In order for this fidelity to be best maintained, the Ca^{2+} sensor is: (a) apposed to the Ca^{2+} source; (b) of low Ca^{2+} affinity. In this way, only those sensors within the high Ca^{2+} nanodomain will be activated. Global Ca^{2+} values do not efficiently attain the activation threshold.

The referee is also puzzled by the fact that ionomycin is effective. However, ionomycin (a blunt Ca^{2+} ionophore) is a strong, non-physiological Ca^{2+} stimulus that can mimic physiological stimuli by artificially driving locally high $[\text{Ca}^{2+}]$ throughout the cell [4]. In our experiments, ionomycin generates a high peri-lysosomal $[\text{Ca}^{2+}]$: **ionomycin stimulates a TPC2-G-GECO1.2 response that exceeds even FcγRIIA activation (% $F_{\text{Max}} = 80 \pm 3\%$ ($n = 65$ %)); UTP does not do this (Fig.4). This can entirely explain why ionomycin activates CaN but UTP does not. We have included the ionomycin data in the text (page 17).**

We do not yet formally understand why CaN uniquely responds to peri-endo-lysosomal Ca^{2+} but, by analogy with other examples in the Ca^{2+} field, it can be most readily explained by the *apposition/low affinity* model stated above (via an association with TPCs). Indeed, calcineurin (CaN) has low affinity Ca^{2+} -binding sites [10,11] making it less efficiently activated by global Ca^{2+} . However, determining the precise mechanism of decoding is well beyond the scope of the current ms, but we **now elaborate our model in the Discussion.**

(3)what is the evidence that Fc receptor stimulation, whether via CD38 or by other means generates NAADP locally? Without this evidence, the model is not convincing.

Our model neither invokes nor requires local NAADP (see also Point 2). Local Ca^{2+} responses are usually driven by the distribution of the *channels* and *not* the messenger [12] e.g. global uncaging of IP_3 evokes highly localized Ca^{2+} events because of *channel* functional heterogeneity [13]. Indeed, the rapid diffusion of NAADP ($50-150 \mu\text{m}^2/\text{s}$ [14]) would mean that it is likely uniform across a $10\text{-}\mu\text{m}$

diameter $M\emptyset$ within 1 second, as long argued for other second messengers [12], but local Ca^{2+} nanodomains would still be formed at endo-lysosomes.

Furthermore, the experiment is technically impossible because we have no single-cell assay for NAADP (the field has not yet identified an NAADP-binding protein to produce a genetically encoded NAADP probe). **We include these remarks in our model.**

(4) the authors state that they used dextran Cal520 because "there are no drugs for directly depleting vesicle calcium content". Elsewhere they use bafilomycin to deplete the vesicular calcium selectively. These statements are incompatible. Does bafilomycin inhibit phagocytosis? Multiple groups have used this inhibitor but have not reported effects on phagocytosis.

First, there is no incompatibility as this was deliberately phrased: we remain correct in saying that there are no *direct* inhibitors of Ca^{2+} uptake into lysosomes (there is no lysosomal 'thapsigargin'; we do not even know the pathway of Ca^{2+} filling with certainty); bafilomycin A1 *indirectly* depletes Ca^{2+} (it collapses the pH gradient first). Nonetheless, **we have re-phrased this section to avoid further ambiguity.**

We also now include **new data with Bafilomycin A1 (Fig. 1) which mimics the Cal520Dx effects on both Ca^{2+} and bead uptake. Two convergent strategies implicate acidic stores such as lysosomes.**

(5) that the K_d of dextran Cal520 at neutral pH is 320 nM is irrelevant; what is its K_d at the acidic pH prevailing in lysosomes? If, as expected, it is much higher, it may be ineffective. Even if able to bind calcium at the acidic pH, why does the continuous uptake of calcium, whether by Ca/H exchange or via endocytosis, saturate its binding sites?

Without the ability to directly and dynamically monitor luminal $[Ca^{2+}]$ (currently untenable for the entire field), we concede that it can never be directly proven that Cal520 has been effective.

However, we contend that it has been effective for the following reasons:

- (i) Non- Ca^{2+} binding dextrans were without effect on phagocytosis or lysosomal morphology.
- (ii) When lysosomes were loaded with Cal520-Dx they showed a classical 'lysosomal storage' morphology (swelling) which is a hallmark of reduced luminal Ca^{2+} [15]. This phenotype also correlates with the Ca^{2+} affinity of the dextran (low affinity does not induce a storage phenotype whereas a high affinity does induce it [15]).
- (iii) To offset the referee's concerns, we have calculated the pH-dependence of Cal520-Dx and show that it can indeed reduce luminal $[Ca^{2+}]$ in the acidic compartments (see *Calculation* below).
- (iv) Regarding the 'continuous uptake' and 'saturation' we disagree with the referee's interpretation:
 - a. If the suggestion of continuous Ca^{2+} uptake and the saturation of luminal buffers were true, then this should hold for *any* Ca^{2+} -storing organelle: therefore the ER would never reach a point of equilibrium, it would keep filling *ad infinitum*. Clearly that does not occur. This is because the filling level of any store is a balance between uptake, leaks and luminal buffering, and these are not linear, unchanging parameters:
 - i. The ER/SR Ca^{2+} store is suggested to exhibit a highly complex, non-linear, dynamic interaction between channels, pumps and buffers [16].
 - ii. The ER Ca^{2+} -ATPase is itself regulated by the ER luminal $[Ca^{2+}]$ and the pump slows down when the stores are full [17,18]. Leaks are also subject to regulation.

- b. Specifically for the endo-lysosomal system, Ca^{2+} uptake via endocytosis is not a unidirectional Ca^{2+} uptake system: following endocytosis of extracellular Ca^{2+} (1mM) the Ca^{2+} is rapidly and actively *extruded* from the endosomes down to the micromolar range [19,20]. Our understanding of how these extrusion pathways and Ca^{2+} leaks vary along the endo-lysosomal pathway is limited, so the assumptions that uptake is ongoing and unlimited is unlikely.
- c. We do not know what effect(s) luminal $[\text{Ca}^{2+}]$ has on the Ca^{2+} -filling rate. For example, Luminal Ca^{2+} is known to regulate lysosomal Ca^{2+} -channel activity [21,22] (i.e. leak) and/or lysosomal $\Delta\Psi$ which might all impact *net* uptake.
- d. Saturation. Ca^{2+} will not only bind to, but dissociate from the Cal520-Dx. The system is likely to reach a new equilibrium, as has been observed when ER luminal Ca^{2+} is chelated with luminal TPEN [23], an analogous system.

Calculation of lysosomal $[\text{Ca}^{2+}]$

BAPTA-based Ca^{2+} dyes exhibit a predictable shift in the K_d at the lysosomal pH [24,25]. The luminal dextran concentration = the extracellular loading-concentration = 1mM (assuming 10mg/ml of a 10kDa dextran, at 1:1 stoichiometry of dye:dextran). The resting lysosomal $[\text{Ca}^{2+}]$ is around 500 μM [26]. Using the Ca^{2+} K_d of 320nM (at neutral pH), and the *same H^+ -binding constants as the parent BAPTA*, we can calculate (using the standard program to calculate bound:free $[\text{Ca}^{2+}]$, Winmaxchelator) that the luminal $[\text{Ca}^{2+}]$ in the presence of Cal520Dx is:

We highlight 2 values of the macrophage lysosomal pH, 4.52 (our own determination using calibrated Oregon-Green Dextran – unpublished) or 4.71 [27]. As can be seen from the graph, the luminal $[\text{Ca}^{2+}]$ would be reduced by 60-75% (from 500 μM to 120-200 μM), depending on the pH value. This supports a proof-of-principle that the dextran will chelate luminal $[\text{Ca}^{2+}]$.

The next issue to consider is the relationship between the luminal $[\text{Ca}^{2+}]$ and the Ca^{2+} release i.e. will this decrease of 60-75% manifest itself as a reduced Ca^{2+} release? The answer is yes: lysosome-dependent diseases, such as NPC [15] or TB infection of $M\phi$ [28] reduce the luminal lysosomal $[\text{Ca}^{2+}]$ by the slightly lower 40-60% and this manifests as reduced Ca^{2+} release. Therefore, this predicted reduction in lysosomal Ca^{2+} will significantly reduce lysosomal Ca^{2+} release.

In summary, we feel our approach is a valid one. **We are happy to include the above estimates in the ms if it is required.**

(6)TPC channels are thought to be activated by PI(3,5)P₂, which is present constitutively in late endosomes and lysosomes. It is not clear how calcium would be retained under these conditions and its release stimulated by NAADP. The authors do not discuss the regulation of TPCs by phospholipids. We apologise for not discussing lipid-regulation, but this was not really the focus of our paper.

However, we (and others in the field) would strongly disagree with the implication that lysosomal channels are ‘constitutively activated’ by resting lipids:

- (a) The implication that endo-lysosomes are *not* Ca²⁺ stores (‘unclear how Ca²⁺ would be retained’), goes against an international cohort of scientists who have shown that Ca²⁺ is indeed stored within endo-lysosomes as assessed by fluorescence, inductively coupled plasma mass spectrometry, and X-ray microprobe analyses [26,29,30]. **Constitutively active channels would deplete the endo-lysosomal Ca²⁺ stores, so this constitutive leak cannot be occurring significantly.**
- (b) **Resting PI(3,5)P₂ levels are likely to be low** because cell stimulation enhances them, and this may then activate channels to release Ca²⁺ [31].
- (c) Both TRPMLs and TPCs can be activated by PI(3,5)P₂ and yet agonists of these channels (e.g. MLSA1 [32,33], antidepressants [34], NAADP/AM [35,36], NAADP [37-39]) **acutely evoke Ca²⁺ release. This would be impossible if the channels were already open and the stores depleted.** These conclusions are commonly shared across the endo-lysosomal Ca²⁺ community, even by those that invoke PI(3,5)P₂ as a channel modulator.
- (d) PI(3,5)P₂ stimulation of TPCs predominantly would stimulate Na⁺ fluxes (whereas NAADP stimulates stronger Ca²⁺ fluxes [35,40]). **This makes Ca²⁺ depletion even less likely.**
- (e) TPCs and TRPML1 are differentially activated during phagocytosis (**Figs. 2, 5**). However, PI(3,5)P₂ indiscriminately acts upon *both* the TPC family as well as the TRPML1 [26,41].
 - a. Therefore, PI(3,5)P₂ cannot be functioning as the only ‘second messenger’ to TPCs as it would affect both channel families. We (and others) view PI(3,5)P₂ as a permissive factor (analogous to PI(4,5)P₂ for some plasma membrane ion channels and transporters).
 - b. Even the Xu group (who first demonstrated PI(3,5)P₂ activates TPCs) latterly concede that it could be a permissive factor [34]; indeed, channel activators *synergise* with PI(3,5)P₂ [34].
- (f) Lysosomal storage pathologies result when the lysosomal [Ca²⁺] drops (or if there are disease-inducing mutations of channels) e.g. reduced metabolite trafficking, abnormal lysosomal morphology and motility [15]. If the channels were constitutively active, these abnormalities would be observed. They are not, until there are disease-inducing gain-of-function mutations [42].

The *physiology* of endo-lysosomal lipids in channel modulation is not well understood, for example:

- (i) We do not know what the resting ‘free’ concentration of PI(3,5)P₂ is in the lysosomal membrane.
- (ii) Phosphoinositides of the plasma membrane are segregated into domains – we do not know whether lipid compartmentation also occurs in endo-lysosomes to reduce the interaction with resting channels.

In view of these uncertainties, we feel there is enough for the reader to deal with and do not feel that a discussion is relevant to the paper. **We have, however, included an additional line stating that TPCs can be gated by PI(3,5)P₂ (page.27).**

(7) Figures 1 and 2 seem internally inconsistent. In Fig 1 no calcium changes are measurable after thapsigargin, implying that the contribution of lysosomes is barely measurable, yet Ned19 causes big changes in Fig 2, as does the elimination of TPC1 or TPC2.

We apologise for the confusion. However, the results are entirely consistent with the so-called ‘trigger model’ of lysosomal Ca^{2+} signalling that has been upheld by the community for the past 20 years.

Trigger Hypothesis

In this model, lysosomes provide the local, initial trigger; the ER is then recruited by CICR as the secondary amplifier [43,44]. Note that, in Ca^{2+} -free medium, global Ca^{2+} recordings are almost exclusively from the ER (the lysosomal store volume is too small to support these large excursions in mammalian cells [43] – **Fig.1, 4**).

By analogy with other cell types where lysosomes and ER are simultaneously recruited by a single stimulus [43,44], FcR-mediated Ca^{2+} responses appear a mixture of NAADP and IP_3 -dependent responses. Therefore, the Ca^{2+} signals are a mixed signal, comprised of:

- (a) Pure lysosomal Ca^{2+} nanodomains.
- (b) Pure IP_3 -dependent ER Ca^{2+} release.
- (c) ER Ca^{2+} -release triggered by the lysosome (trigger component).

Specific Points

- Therefore, with thapsigargin/CPA (b) and (c) are removed entirely (**Fig. 1**). The local lysosomal Ca^{2+} nanodomains (a) are undetectable using a cytoplasmic Ca^{2+} indicator (but detected using TPC2-G-GECO1.2 **Fig. 4**).
- Ned-19/TPC-KOs removes the lysosome-dependent signals (a,c), but leaves (b).

This is why the inhibition of lysosomal Ca^{2+} release has a ‘disproportionate’ effect upon global Ca^{2+} signals. **We have described this in the Discussion (page 30).**

(8) the authors report that TPC1 and TPC2 both colocalize to a similar extent with LysoTracker-positive organelles. This seems inconsistent with a number of papers from others in the literature and even with their own data. The extent of colocalization of TPC1 with LysoTracker in Fig EV3 seems modest (and could be entirely fortuitous) and in Fig EV4 the colocalization of TPC1 with LAMP is negligible. These findings are more compatible with the published findings that TPC1 is in a Rab5-positive early compartment that would not be LysoTracker positive.

We apologise for the confusion. TPC2 has a narrower distribution, almost exclusively in the late endosome/lysosome; the **vesicle distribution of TPC1 is broader** as has been previously reported, from recycling endosomes to lysosomes [45]. The primary BMDM data indeed appeared as these higher correlation populations.

We now include new data with TPC1/2 expression in RAW cells with a much larger sample number. As can be seen from **Fig. EV3B,C**, the correlation between TPC1 and LysoTracker is far more scattered with individual macrophages showing stronger and weaker correlation (like other cell types [45]). The BMDM data reflected a higher proportion of the stronger correlates.

-if TPC1 and TPC2 are in fact in different compartments, it is difficult to envisage how knocking out either one causes a $\approx 75\%$ reduction calcium changes. It would also require both early and late endosomes or lysosomes to be present at sites of phagocytosis.

We sympathise with the referee's difficulty regarding the similar effects of single KOs upon global Ca^{2+} signals. There are several issues:

- As we raise below in **point 18**, we do not currently understand why there is no isoform redundancy in the system – phagocytosis seems to require both to function properly. It may reflect TPC1 and TPC2 acting in *series*, but we currently do not know why.
- We now quantify new data to show that TPC1/2 both appose the phagocytosis sites (**Fig. EV7**).

(9) the authors report that TPC1 and TPC2 surround the lysosome. However, in one case (TPC2) the channels line the phagosomal membrane, while in the other (TPC1), vesicles cluster in the vicinity. Discussing both phenomena in the same context is inappropriate. Moreover, it is not at all clear how TPC2 insertion into a sealed, presumably late phagosome is relevant to the proposed role of the channel in forming the phagosome.

We apologise for the confusion and have re-phrased the description of the distribution pattern as well as including new data.

The pattern of vesicular apposition to the phagosomal membrane is variable at earlier times (later times indeed show the insertion that the referee refers to). We would like to reinforce that, even if insertion has occurred, it means that vesicles must have – by definition – approached the phagosome and so this remains relevant.

Prior to insertion, we can observe single vesicles, small clusters of vesicles or an encirclement of the entire phagosome. Overall, we quantify a close apposition of either TPC1 or TPC2 that is juxtapose to the bead boundary, but not within the phagosomal membrane (**Fig. EV7**).

(10) Figure 4 is most puzzling. Why are the calcium oscillations measured in the cytosol perfectly synchronized with perilyosomal changes, when the cytosolic changes are demonstrated earlier to be attributable to the ER and Orai1? And why does Ned19 dampen the cytosolic oscillations by 50% when the contribution of TPCs is shown earlier to be too small to detect? And if TPC1 and TPC2 are in fact in different compartments, how are their oscillations synchronized?

- **Synchronization.** There are two relevant issues: the Trigger Hypothesis and Bidirectional Feedback.
 - We refer the referee to Point (7) regarding the Trigger Hypothesis. The synchronization is entirely consistent with the fact that lysosomes provide the 'pacing' trigger that then entrains the ER Ca^{2+} release. Unfortunately, we do not have sufficient temporal resolution to confirm that TPC-GECI signals precede cytosolic Ca^{2+} .
 - Indeed, in experiments where the ER is eliminated (either by CPA or EGTA/AM), we no longer observe Ca^{2+} oscillations (**Fig. 4**) but rather a single local TPC2-G-GECO1.2 Ca^{2+} spike. This is consistent with measuring the lysosomal trigger but without the oscillatory amplifier.
 - Furthermore, the dialogue between lysosomes and the ER is *two-way* and complex: one influences the other in complex feedback loops [46-48], which are likely to mutually support synchronization.

- Orai? These experiments are in Ca^{2+} -free medium so there is no Orai component.
- Ned-19: As discussed above (and Point 7, Trigger Hypothesis), dampening is due to reducing the triggering from the ER.
- Different compartments. We have not and cannot measure TPC1 and TPC2 responses at the same time (we only have used one colour GECl, G-GECO1.2) so we do not formally know that they are synchronized.

(11) how can calcium oscillations persist in cells loaded with dextran Cal520 in Fig 1, yet be almost eliminated by Ned19 in Fig 4?

In Fig. 4d, cytosolic Ca^{2+} oscillations persist in presence of Ned-19, they are not ‘eliminated’.

If the referee simply means that Ned-19 inhibits more efficiently than does Cal520-Dx, then we would agree. Because of solubility (and expense) we cannot use a higher concentration of the dextran, so we do not know where we are on a concentration-inhibition curve.

- There is no reason to assume that these concentrations of Ned-19 and Cal520-Dx are inducing the same degree of inhibition of the Ca^{2+} signal.
- Neither is there a linear relationship between the global Ca^{2+} signal and bead uptake.

(12) the inability of TPC-GG to detect the perilyosomal calcium when released by a neighboring channel is unprecedented. What accounts for this extreme behavior?

We are pleased that the referee appreciates the remarkable compartmentation of the system.

However, segregation of channels in a membrane is not unprecedented. For example, in the plasma membrane, Orai1 is differentially segregated away from neighbouring Ca^{2+} -channel families in the same membrane: the GECl of Orai1-GCaMP3 does not detect Ca^{2+} influx stimulated through P2X4 channels whereas it does detect Ca^{2+} from TRPV1 [49]. This underscores channel compartmentation *even in the same membrane system*. Our data are amongst the first to demonstrate this for endo-lysosomes, and that this has downstream consequences.

We further argue that the compartmentation of the TPC Ca^{2+} signals is not surprising: the local $[\text{Ca}^{2+}]$ near the channel mouth will, by definition, be higher than the surrounding cytosol.

A

The graph above is taken from Neher's work on Ca^{2+} diffusion (Fig. 2A) [50] showing that the $[\text{Ca}^{2+}]$ rapidly falls off with the distance from the channel mouth. Depending on the model, it could fall by 99% when only ~5nm (rapid) or 50nm (linear) away from the channel mouth. For comparison, the TPC dimers are ~ 8nm across [51,52].

The density of TPC channels on the curved lysosomal surface (~500-2000nm diameter) is unclear but may be as low as only 20 channels per lysosome [53]. We have calculated that even **500** channels on a vesicle would only occupy **0.4 to 6.4 %** of the vesicle surface area, depending on the vesicle diameter (**values now included in the Discussion**). This allows ample space to segregate channels.

Moreover, Ca^{2+} -binding proteins are known to be expressed on the lysosome surface and these local Ca^{2+} buffers could further 'isolate' or limit the Ca^{2+} nanodomains, if placed in between channels. The field simply does not know.

We also include new data explicitly demonstrating that TRPML1-G-GECO1.2 and TPC2-G-GECO1.2. are differentially activated by different stimuli – TRPML1 is only activated by large beads, whereas TPC2 is activated by small and large beads (Fig. 5). This further reinforces our compartmentation model and also that different channels are differentially recruited to serve different roles.

(13)in Fig 5d, why is dynamin exclusively localized to one bead (which seems to have been already internalized) while it is missing from most other beads, including ones that are nearer the membrane and in the process of being ingested, judging from the actin staining?

We apologise for the potential confusion, but there are two issues that need to be considered:

- (a) These are confocal slices, and dynamin association with beads (especially at the pinching-off site) would not be expected to be parfocal across the entire field.
- (b) Dynamin association with phagosomes is *transient* [1]. One would not expect all beads to be associated with dynamin at any given time since they would be ingested at different times.

(14)why does activation of dynamin for phagocytosis require calcium-induced dephosphorylation, while all other dynamin dependent processes, like endocytosis, proceed in the absence of calcium changes?

We disagree with the referee that 'all other dynamin processes' are Ca^{2+} -independent. The following are a sample over 20 years showing Ca^{2+} /dynamin inter-dependence and we could only refer to a subset of these in our ms:

- Phagocytosis can be considered a specialized form of endocytosis and there is a substantial literature over decades showing that endocytosis can likewise be dependent on Ca^{2+} , calcineurin [CaN] and dynamin.
 - Neuronal dynamin-dependent endocytosis is Ca^{2+} (and CaN) dependent [54,55].
 - This can be via localized Ca^{2+} nanodomains as evidenced by the differential effect of EGTA and BAPTA, [56] just as we have observed in phagocytosis.
 - Indeed, calcineurin and dynamin are known to dynamically form a complex to drive endocytosis [57].
- Cytokinesis requires Ca^{2+} and calcineurin-dependent dephosphorylation of Dyn-2 [58].
- Internalization of the CFTR [59] or GABA_B receptors [60] depends on Ca^{2+} /CaN and dynamin.
- Endocytosis of the H⁺-ATPase in osteoclasts is via Ca^{2+} -sensitive dynamin pathway [61].

- Pancreatic-cell endocytosis is Ca^{2+} - and dynamin-dependent [62].

Space restrictions mean that we have only highlighted a few examples in the Discussion.

(15)cyclosporine and FK506 had been used in experiments involving phagocytosis without any reported inhibitory effects (e.g. Cell 2007, 130:37). Why the apparent discrepancy?

We would disagree that phagocytosis is always reported to be refractory to CsA/FK. The issue is not straightforward.

Whether phagocytosis is inhibited or not by CsA/FK appears to be dependent on a number of experimental factors including (but not restricted to) the type of phagocyte, time-course, in/ex vivo, experimental regimens.

- For example, phagocytosis by granulocytes is *indeed* inhibited by CsA or FK [63,64].

We do agree that other studies have failed to observe an inhibition of phagocytosis by CsA/FK. However, please note:

- To the best of our knowledge, there are (surprisingly) no studies that have replicated the conditions of our experiments i.e. **short** time courses, 3-6- μm particles, ex vivo BMDMs.
- Many studies investigating CsA/FK506 have been conducted over *long time courses* (1-24 **hours**). An inhibition may have been missed if the *initial rate* of phagocytosis was suppressed (but longer times allowed the same end-point to be attained eventually).
- These drugs have often been tested against phagocytosis *in vivo* (which raises whole-animal, multicellular complications and uncertainties about the [drug] exposed to the $\text{M}\emptyset$) and usually over long times courses (see above).

In conclusion, we are confident of our own data which has been replicated many times, and can only suggest differences due to the various experimental protocols.

(16)where is calcineurin in cells at rest and why doesn't the much larger and more localized calcium concentration change produced by release from ER and Orai1 suffice to activate it?

Ca^{2+} from ER/Orai may not be 'larger and localized'. Our TPC-GEI constructs clearly show (Fig. 4) that the local endo-lysosomal Ca^{2+} domains are larger (and more local) than when UTP (IP_3 and Orai) is stimulated.

The referee touches on the key point of 'location'. As discussed on previous point (2), it has been long recognised in the Ca^{2+} field that not all Ca^{2+} sources couple to the same downstream decoders and this paradox has been resolved by the realization that Ca^{2+} signals are compartmentalized; it is not the global but the *local Ca^{2+} domains* that are key [65].

Therefore, our working hypothesis (and ongoing work) is that CaN associates with TPCs for activation by the Ca^{2+} nanodomains. Although this is well beyond the scope of the current ms and is a study in itself (see Point 2), we have **made mention of this in a revised Discussion to help orientate the reader.**

(17) how can 60% of the phosphodynamin be selectively dephosphorylated by calcineurin activated locally by peri-lysosomal calcium?

These concerns are countered by the fact that there are numerous precedents in the literature that *local* lysosomal Ca^{2+} release is the unique stimulus that evokes *whole-cell* changes in protein phosphorylation (e.g. calcineurin, mTORC1, ERK, JNK, CaMKK2) [33,66-69] or 'global' protein activity (e.g. oncogenes) [69].

These then translate into downstream consequences such as autophagy & transcription factor (TFEB) activation [33], differentiation [70], angiogenesis [68], to name but a few.

It is already accepted that local lysosomal Ca^{2+} can couple to global consequences. We add phagocytosis to that list.

(18) how can calcineurin remain phosphorylated to a level that is >100% of the control in TPC1 KO cells, yet the cells phagocytose with 50% efficiency? Does this not imply that dephosphorylation is not essential for phagocytosis? And why is there no dephosphorylation caused by TPC2 present in the TPC1 KO? If TPC1 and 2 are present in the same compartment, shouldn't calcium be released via TPC2?

(We assume that the referee means the phosphorylation of *dynamamin*, not of calcineurin.)

We correlate the dephosphorylation of dynamamin with the *TPC-dependent* component. Yes, there is a TPC-independent component, but we are not claiming that CaN/Dyn is responsible for that.

TPC1 and TPC2 are not always in the same compartment. Although some TPC1 is found in lysosomes, it is also found in other compartments (recycling, early endosomes [45]) that could contribute Ca^{2+} signals. Indeed, it is already known that these other compartments contribute to the phagosomal membrane.

We sympathise with the referee regarding the lack of redundancy between TPC1 and TPC2. Phagocytosis seems to depend upon both isoforms; eliminating *both* isoforms in double knockout cells (DKO) does not produce any further inhibition (**new Fig.2**), this shows that the residual phagocytosis is TPC-independent (and not due to the residual isoform in the single KO). We currently do not understand why. Potential explanations:

- Each isoform acts in *series* within the same pathway, and not in parallel.
- Alternatively, TPC1 and TPC2 act in parallel but coincidentally converge on Dynamamin activation. This is ongoing work.

We allude to this in the Discussion.

(19) why does phosphodynamin continue to decrease gradually over 20 min, if the calcium transient are over within a minute or two? It is essential to establish what fraction of the dynamamin is phosphorylated prior to stimulation, since a rather small (<10%) change occurs within the period required for phagocytosis. Not all the dynamamin could be inactive prior to phagocytosis, to the extent that endocytosis is ongoing.

The only reason why Ca^{2+} transients are so brief is that most of our Ca^{2+} measurements were conducted in Ca^{2+} -free medium to investigate the Ca^{2+} -release phase in isolation, as we detailed (cf. Fig. 1 in Ca^{2+} -containing).

(Note: unless stated otherwise, the phagocytosis (bead uptake) and phosphorylation experiments were conducted in normal (Ca^{2+} -containing) medium. Since Ca^{2+} influx has absolutely no role in phagocytosis (Fig. 1), our results show that Ca^{2+} influx would only serve to sustain the endo-lysosomal stores and not to act as a direct Ca^{2+} source to stimulate phagocytosis.)

We are unclear why the referee feels there are kinetic discrepancies.

- In Fig. 6J, the dephosphorylation is significant by **5 mins** and clearly ongoing thereafter. In the Western blot (Fig. EV8E,F) a significant dephosphorylation is observed within **1 min**. These are entirely compatible with the phagocytosis kinetics (and the time for lysosomes to make contact with beads = 81s after Ca^{2+}).
 - Moreover, the earliest time point we have measured bead uptake is 10 min (**Fig. 2**; which shows a significant, if variable, degree of bead uptake) and this equates to a ~20% drop in *global* phosphorylation.
- It should also be born in mind that our phosphorylation assays are measured in *populations* of cells whereas bead uptake and Ca^{2+} signals are recorded in *single cells*. The population measurements are a spatio-temporal averaging, with single cells reacting with different time courses. Furthermore, we do not know whether all the dynamins of a single cell is equally important – there might be a critical subpopulation. Currently, there is no reliable means of resolving single-molecule subcellular dynamin phosphorylation status, to the best of our knowledge.
- Fraction phosphorylated. Absolute quantitation would require laborious mass spectrometry (MS) because other approaches would rely on the efficacy of antibody binding. However, a qualitative assessment of the Western Blot (Fig. EV8) suggests that 50% of the basal dynamin is dephosphorylated upon phagocytosis.
 - The resting phosphorylation status is technically impossible to assess without MS. Of course, it is likely that global dynamin is partly dephosphorylated under basal conditions (e.g. as indicated by the higher phosphorylation in TPC1-KO cells), but again, heterogeneity makes it impossible to determine whether the *relevant* dynamin population is being measured.

In summary, we do not agree that our kinetics are inconsistent. Moreover, it is technically demanding to determine the absolute fraction of relevant dynamin that is basally phosphorylated and this would not enhance the paper.

Referee #2

In this important manuscript the authors describe a new mechanism by which the lysosomes drive phagocytosis. The authors provide ample and convincing evidence to conclude that stimulation of the macrophages Fc receptor releases Ca^{2+} from the endolysosomal organelles through the NAADP-activated TPC channels to generate a near lysosomal Ca^{2+} signal that is independent of global Ca^{2+} homeostasis and release. This Ca^{2+} nanodomain activated calcineurin that dephosphorylates dynamin. The dephosphorylated dynamin pinch to internalized the engulfed particle, including pathogens, independent of their size.

The novel findings substantially add to understanding the role of the lysosomes in phagocytosis and understanding this fundamental biological process. However, I have several fairly minor comments.

An issue that is not fully resolved is the role of TRPML1 and the TPCs in generating the nanodomain Ca^{2+} signal and in phagocytosis. The two channels are expected to reside in the same organelles and perhaps share the same Ca^{2+} pool. To resolve this issue, in the discussion the authors suggest two different roles for the channels that maybe mediated by neighboring organelles. This should be tested more directly using the available pharmacological activators and inhibitors of TRPML1. The authors should test if prior treatment of the cells with TRPML1 activators and inhibitors affects phagocytosis stimulated by FcR.

We thank the referee for the suggestions. However, the pharmacology of TRPML1 is not as clear cut as we would like. The only TRPML1 inhibitor to which we have access (ML-SI3) is also a potent NAADP/TPC blocker (our unpublished data), so we would not be able to conclude anything. It was for this reason that we used the cleaner genetic approach of the TRPML1-knockout mice. We feel this is more definitive and hope the referee understands the decision.

We have also attempted to rescue TPC-KO cells by artificially stimulating TRPMLs using the agonist, ML-SA1. In keeping with different channels for different roles, we now show that ML-SA1 fails to rescue TPC-KO (**Fig.2**). That is, TRPMLs cannot substitute for TPCs.

We do *not* have to invoke the idea of ‘neighbouring organelles’, for we regard TRPML1 and TPC2 as probably being on the same lysosomal vesicles. Rather, we suggest that different stimuli (i.e. different sized beads) differentially recruit the different channels. We now include new data (**Fig. 5**) directly comparing channel activation as assessed with TPC2-G-GECO1.2 and TRPML1-G-GECO1.2.

We show that TRPML1 is only activated by large beads (whereas TPC2 is activated by small and large beads).

How this extreme Ca^{2+} signalling compartmentation is decoded is currently unclear but will form the basis of ongoing work.

Another clarification with regard to stimulation of the FcR is involvement of NAADP in its function. The crucial requirement for Ca^{2+} in the lysosomal nanodomain can be further demonstrated by showing that stimulation with ATP does not affect phagocytosis.

We thank the referee for the suggestion, though we used the physiological macrophage purinoceptor agonist UTP (not ATP). We have now confirmed that UTP cannot rescue TPC KO cells (**Fig.2I**).

There is significant overlap between the results and discussion sections, and both are very long and

can be reduced by 20-30% without losing clarity of conveying the significance of the story.

We apologise for this and have tried our best to shorten the Discussion. However, new data and points raised by all referees have meant further clarification was required.

Referee #3

This paper reports extensive findings that demonstrate a key role for Ca^{2+} nanodomains generated by endo-lysosomal two-pore channels (TPCs) in driving phagocytosis by macrophages. This mechanism is novel; helps resolve conflicting reports in the literature; and, as phrased the authors "promotes endo-lysosomes from downstream effectors to upstream initiators ... of phagocytosis". The findings will be highly relevant to studies of phagocytosis and endocytosis in multiple cell types. The paper is a pleasure to read, and guides the reader through a complex sequence of experiments with an elegant, expository writing style that is becoming sadly rare in the current scientific literature.

The authors employ a variety of complementary techniques and experimental systems to demonstrate that Fc γ R-induced phagocytosis is regulated by the release of endo-lysosomal Ca^{2+} through TPCs including: buffering of intracellular Ca^{2+} by exogenously loaded Ca^{2+} buffers with different binding kinetics and affinities; buffering of endo-lysosomal Ca^{2+} by passive uptake of a dextran-conjugated Ca^{2+} indicator and by recombinant expression of calcium binding protein targeted to the outer endo-lysosomal membrane; the use of cells from TPC1 and TPC2 knockout mice; inhibiting TPC activity with Ned-19; and generation of tethered TPC-GECI Ca^{2+} reporters. The authors further extend their investigation to identify downstream proteins dynamin and calcineurin as critical players regulating phagocytosis in BMDMs.

Although caveats can be raised to some approaches (e.g. inference of Ca^{2+} microdomains from comparative actions of BAPTA vs EGTA; 'bleed-through' of global Ca^{2+} to the tethered Ca^{2+} probe; and issues raised below), taken together the authors' multi-pronged approach, utilizing diverse methodologies to address each step in the phagocytosis pathway, presents compelling evidence to support their conclusions.

We present the following comments as suggestions for further improvement of the paper, not as essential requirements for new work before the paper could be published.

Substantive comments

1. In Fig 2C, phagocytosis of 3 μm beads in Tpcn1 $^{-/-}$ or Tpcn2 $^{-/-}$ BMDM cells was significantly reduced as compared to WT cells, but substantial phagocytosis still remained (20% and 35% of WT respectively at 40 min). It would be beneficial to test Fc γ R-induced phagocytosis in Tpcn1/Tpcn2 double knockout cells. This should be relatively straightforward since the authors have previously shown that TPCs are required for NAADP-induced Ca^{2+} signaling because BMDMs lacking TPCs fail to evoke Ca^{2+} signals in response to NAADP (Ruas et al., 2015a).

We thank the referee for the suggestion and include new data (Ca^{2+} and bead uptake) with DKO cells (Figs. 2A,C,D - purple). There is no further inhibition by removing both TPC isoforms which suggests that the residual component is truly TPC-independent and not due to the residual isoform.

2. In Fig 4a-k, much effort was taken to characterize Ca^{2+} signals detected by G-GECO1.2 tethered to TPCs in order to differentiate local Ca^{2+} signals arising from the release of Ca^{2+} from endo-lysosomes from 'bleed-through' of global Ca^{2+} signals. A more direct approach would be to deplete ER Ca^{2+} stores (with TG or CPA) and then invoke phagocytosis by Fc γ R activation as shown in Fig 1G. Under this scenario, the obfuscating global Ca^{2+} signals caused by ER Ca^{2+} release would be abolished and the local Ca^{2+} signals arising from lysosomal Ca^{2+} release, which as the authors contend are essential

to phagocytosis, should be retained and be more specifically resolved by their GECI-TPC reporter. This would further strengthen the conclusion that endo-lysosomal Ca^{2+} nanodomains drive phagocytosis.

We thank the referee for the suggestion and include new data (Fig. 4I-J) where the ER Ca^{2+} store is pre-depleted with CPA and the cells stimulated with beads. As the referee predicted, FcR stimulation evokes only a local lysosomal Ca^{2+} response (detected with TPC2-G-GECO1.2) but not a global Ca^{2+} response (detected with cytosolic Calbryte590). This is consistent with TPC2-G-GECO1.2 measuring local lysosomal Ca^{2+} nanodomains.

3. Although the experimental evidence strongly points to endo-lysosomal Ca^{2+} nanodomains initiating phagocytosis by activating calcinurin, it is difficult to envision how this mechanism may work. How is it that a small and exquisitely confined Ca^{2+} domain (apparently localized to individual TPCs) can selectively activate calcinurin whereas a much larger global Ca^{2+} elevation evoked by release through IP_3Rs does not? The notion of Ca^{2+} 'nanodomains' is typically considered in situations where Ca^{2+} channels are located at contact sites that hold them in immediate apposition to an effector or transporter (e.g. Ca^{2+} transfer from ER to mitochondria), but how could that be the case here? The authors suggest that the Ca^{2+} signals are restricted to each forming phagosome, and present evidence that movement of lysosomes toward the phagosome is required for phagocytosis. But if that is what underlies the spatial localization of the signaling it would seem the movement must occur before the transmission of the Ca^{2+} signal?

We apologise for not adequately explaining our working model. We agree with the referee that apposition of Ca^{2+} source and effector is the most likely explanation for such signalling fidelity. Therefore, we hypothesise that calcineurin associates with endo-lysosomes (and specifically with TPCs) where it will sense the locally high Ca^{2+} nanodomains.

Our new data indicate that lysosomes do move rapidly towards the bead/phagosome (~81s after the first Ca^{2+} signal, Fig.EV7). Whilst we do suggest that lysosomal motility is likely to be important (Fig.7), we do not currently understand how this is involved in generating selective signals. Indeed, as mentioned to Referee 1, our model does not strictly require targeting of the lysosomes to the phagosome (e.g. a factor could diffuse from activated lysosomes to the phagosome).

Given these uncertainties (and the amount of additional work required to resolve this), we hope that a brief expansion of the model in the Discussion will be sufficient to orientate the reader.

It would be interesting if the authors could present their thoughts on these questions.

Minor comments

1. Referencing Fig 2G, the authors state "The redistribution of TPCs from endo-lysosomes suggests that the acidic Ca^{2+} stores themselves rearrange, concentrating around and close to the phagosome..." but do not provide evidence to support this statement, i.e. What was the distribution of TPCs prior to phagocytosis and how was this redistribution quantified?

We apologise for the oversight. We now include new data imaging lysosomal movement at higher spatial resolution and we now quantify both the timing (Fig. EV4A-D) and the spatial relationship between bead and TPCs (Fig. EV4G,H). This indicates that lysosomes do appose the nascent phagosome in 81 ± 21 s (n=27).

2. The finding that ER Ca^{2+} release by UTP produces a small signal in the GECI-TPC reporter not greatly different than the signal detected by untethered cytosolic GECI (Fig 4I,J), which the authors

attribute to a 'bystander effect', is opposite to the findings of Atakpa, Thillaiappan et al, 2018 (cited by the authors; page 13) who, using the same G-GECO1.2 (albeit tethered to a different lysosomal protein) reported IP₃R-mediated Ca²⁺ release selectively delivers Ca²⁺ to lysosomes resulting in much larger signals in the tethered vs untethered GECI. Some discussion as to the cause of these differences would be beneficial.

The referee makes an astute point. We can only suggest cell-type differences (HeLa versus MØ). We have no reason to assume that the lysosome-ER membrane contact sites (MCSs) and/or relative IP₃R clustering is the same between MØ and HeLa cells. It is noteworthy that in HeLa cells it was predominantly a sub-population of lysosomes in MCSs that saw these high [Ca²⁺].

Our discussion is already quite long, so we have opted to remove this reference and will discuss this in a follow-up paper.

3. All n numbers are reported as the number of cells analyzed and data in bar and point graphs are presented as mean ± sem per Experimental Methods. Please indicate the number of times each experiment was performed for each experimental condition tested and clarify whether statistics were performed on the number of cells analyzed or on the number of experimental repetitions (the mean of the means). Also, indicate in the Experimental Methods the statistical test/s used to determine if the data are normally distributed (an underlying assumption when applying student's t test or ANOVA).

We apologise for the lack of clarity. This has been rectified in the Methods.

4. Check scale bars on image panels. Many panels showing 3 µm diameter beads have the beads appear considerably larger than that as judged by the scale bars. (e.g. Fig 1I, Fig 2G, etc).

Thank you for highlighting the errors for which we apologise; we have checked all scale bars and have amended the legends where necessary.

5. The numerical y-axes on several graphs are simple labeled "Phagocytosis" (e.g. Fig 1H,K, Fig 2B,H,I, etc.). Clarify what the units on the axis represent: e.g. if it is bead uptake per cell, explicitly state in figure or figure legend.

We apologise for the confusion and this has been amended.

6. Page 15, top. The statement that the D276k mutant is not acting as a dominant negative is difficult to understand at this point in the text without further explanation; although this is clearly elucidated in the Discussion.

We understand the issue, and decided to remove this remark from the Results, but retain the point in the Discussion.

7. Page 29, discussion of 'trigger' hypothesis of amplified Ca²⁺ release through IP₃Rs. Although the global Ca²⁺ signal was blocked by EGTA, this does not eliminate the possibility of coupling via a Ca²⁺ nanodomain if TPCs and IP₃Rs were in close apposition.

We thank the referee for making this point and we agree with them. Using previous equations [71] and a diffusion coefficient for Ca²⁺ of 233 µm²/s [12], we have calculated length constants for EGTA and BAPTA in macrophages (see Calculations, Materials and Methods) (~300 nm and 10 nm respectively). Therefore, we agree with the referee that EGTA would not be able to buffer Ca²⁺ if IP₃R and TPCs were very intimately associated, but that BAPTA would. **We now include these values in the Discussion and qualify our statements regarding triggering.**

8. Page 20. Heading "FcγR-mediated phagocytosis induces TPC-dependent dephosphorylation of dynamin". Reword, as it is the dephosphorylation of dynamin that activates phagocytosis.
Point taken, and we have now changed this to "FcγRIIA-mediated phagocytosis requires TPC-dependent dephosphorylation of dynamin-2." We hope this is acceptable.

References

- [1] Marie-Anais, F., et al. (2016). *Traffic (Copenhagen, Denmark)*, 17 487-499.
- [2] Westman, J., et al. (2019). *J Leukoc Biol*, 106 837–851.
- [3] Kar, P., et al. (2011). *The Journal of biological chemistry*, 286 14795-14803.
- [4] Kar, P., et al. (2014). *Curr Biol*, 24 1361-1368.
- [5] Kar, P., Parekh, A.B. (2015). *Molecular cell*, 58 232-243.
- [6] Kar, P., et al. (2016). *Molecular cell*, 64 746-759.
- [7] Bianchi, K., et al. (2004). *Biochimica et biophysica acta*, 1742 119-131.
- [8] Nelson, M.T., et al. (1995). *Science*, 270 633-637.
- [9] Sonkusare, S.K., et al. (2012). *Science*, 336 597-601.
- [10] Feng, B., Stemmer, P.M. (1999). *Biochemistry*, 38 12481-12489.
- [11] Klee, C.B., et al. (1998). *The Journal of biological chemistry*, 273 13367-13370.
- [12] Kasai, H., Petersen, O.H. (1994). *Trends in neurosciences*, 17 95-101.
- [13] Thillaiappan, N.B., et al. (2017). *Nat Commun*, 8 1505.
- [14] Churchill, G.C., Galione, A. (2000). *J. Biol. Chem.*, 275 38687-38692.
- [15] Lloyd-Evans, E., et al. (2008). *Nature medicine*, 14 1247-1255.
- [16] Guerrero-Hernández, A., et al., Sarco-Endoplasmic Reticulum Calcium Release Model Based on Changes in the Luminal Calcium Content, in: M.S. Islam (Ed.) *Calcium Signaling*, Springer International Publishing, Place Published, 2020, pp. 337-370.
- [17] Mogami, H., et al. (1998). *Embo. J.*, 17 435-442.
- [18] Yano, K., et al. (2004). *Biochem J*, 383 353-360.
- [19] Gerasimenko, J.V., et al. (1998). *Curr. Biol.*, 8 1335-1338.
- [20] Sherwood, M.W., et al. (2007). *Proceedings of the National Academy of Sciences of the United States of America*, 104 5674-5679.
- [21] Lagostena, L., et al. (2017). *Sci Rep*, 7 43900.
- [22] Pitt, S.J., et al. (2010). *The Journal of biological chemistry*, 285 35039 –35046.
- [23] Hofer, A.M., et al. (1998). *The Journal of cell biology*, 140 325-334.
- [24] Morgan, A.J., et al., Imaging approaches to measuring lysosomal calcium, in: F.M. Platt, N. Platt (Eds.) *Methods In Cell Biology: Lysosomes and Lysosomal Diseases*, Elsevier Inc., Academic Press, Place Published, 2015, pp. 159-195.
- [25] Christensen, K.A., et al. (2002). *J. Cell Sci.*, 115 599-607.
- [26] Morgan, A.J., et al. (2011). *Biochem J*, 439 349-374.
- [27] Steinberg, B.E., et al. (2010). *The Journal of cell biology*, 189 1171-1186.
- [28] Fineran, P., et al. (2016). *Wellcome Open Res*, 1 18.
- [29] Wang, X., et al. (2012). *Cell*, 151 372-383.
- [30] Garrity, A.G., et al. (2016). *Elife*, 5 e15887.
- [31] Samie, M., et al. (2013). *Developmental cell*, 26 511-524.
- [32] Kilpatrick, B.S., et al. (2016). *Journal of cell science*, 129 3859-3867.
- [33] Medina, D.L., et al. (2015). *Nat Cell Biol*, 17 288-299.
- [34] Zhang, X., et al. (2019). *Elife*, 8.
- [35] Ruas, M., et al. (2015). *The EMBO journal*, 34 1743-1758.
- [36] Ruas, M., et al. (2010). *Curr Biol*, 20 703-709.
- [37] Calcraft, P.J., et al. (2009). *Nature*, 459 596-600.
- [38] Zong, X., et al. (2009). *Pflugers Arch*, 458 891-899.
- [39] Brailoiu, E., et al. (2009). *The Journal of cell biology*, 186 201-209.
- [40] Gerndt, S., et al. (2020). *eLife*, 9 e54712.
- [41] Dong, X.-P., et al. (2010). *Nature Communications*, 1 1-11.
- [42] Grimm, C., et al. (2012). *The Journal of pharmacology and experimental therapeutics*, 342 236-244.
- [43] Morgan, A.J. (2016). *Biochemical Society transactions*, 44 546-553.
- [44] Galione, A. (2015). *Cell calcium*, 58 27-47.

- [45] Rietdorf, K., et al. (2011). *The Journal of biological chemistry*, 286 37058-37062.
- [46] Morgan, A.J., et al. (2013). *The Journal of cell biology*, 200 789-805.
- [47] Sanjurjo, C.I., et al. (2012). *Journal of cell science*, 126 289-300.
- [48] Atakpa, P., et al. (2018). *Cell Rep*, 25 3180-3193 e3187.
- [49] Bastian-Eugenio, C.E., et al. (2019). *Commun Biol*, 2 88.
- [50] Naraghi, M., Neher, E. (1997). *J Neurosci*, 17 6961-6973.
- [51] She, J., et al. (2018). *Nature*, 556 130-134.
- [52] She, J., et al. (2019). *Elife*, 8.
- [53] Fameli, N., et al. (2014). *FI000Research*, 3 93.
- [54] Nicholson-Fish, J.C., et al. (2016). *Neurochem Res*, 41 534-543.
- [55] Morton, A., et al. (2015). *Journal of neurochemistry*, 134 405-415.
- [56] Yamashita, T., et al. (2010). *Nat Neurosci*, 13 838-844.
- [57] Lai, M.M., et al. (1999). *The Journal of biological chemistry*, 274 25963-25966.
- [58] Chircop, M., et al. (2010). *Cell Mol Life Sci*, 67 3725-3737.
- [59] Patel, W., et al. (2019). *Cell Mol Life Sci*, 76 977-994.
- [60] Nicholson, M.W., et al. (2018). *Mol Psychiatry*, 23 1851-1867.
- [61] Sakai, H., et al. (2010). *American journal of physiology*, 299 C570-578.
- [62] He, Z., et al. (2008). *Traffic (Copenhagen, Denmark)*, 9 910-923.
- [63] Emal, D., et al. (2019). *Sci Rep*, 9 106.
- [64] Tourneur, E., et al. (2013). *PLoS Pathog*, 9 e1003152.
- [65] Bagur, R., Hajnoczky, G. (2017). *Molecular cell*, 66 780-788.
- [66] Wang, W., et al. (2015). *Proceedings of the National Academy of Sciences of the United States of America*, 112 E1373-1381.
- [67] Li, R.J., et al. (2016). *Elife*, 5.
- [68] Favia, A., et al. (2014). *Proceedings of the National Academy of Sciences of the United States of America*, 111 E4706-4715.
- [69] Schwartz, D.M., Muallem, S. (2019). *EMBO reports*, 20.
- [70] Aley, P.K., et al. (2010). *Proceedings of the National Academy of Sciences of the United States of America*, 107 19927-19932.
- [71] Kidd, J.F., et al. (1999). *The Journal of physiology*, 520 Pt 1 187-201.

Thank you for submitting a revised version of your manuscript. Please accept my apologies for the delay in getting back to you. Your study has now been seen by the original referees whose comments are shown below.

As you will see, while referee #2 and #3 find that their criticisms have been sufficiently addressed and recommend the manuscript for publication, reviewer #1 still feels that his/her original concerns remain unsettled. Hence, we ask you to better discuss and interpret your findings in the light of referee #1's comments and, wherever it is required, to tone down the main conclusions. In addition to resolving these remaining points, there are a few editorial issues concerning text and figures that I need you to address.

Referee #1:

I read in detail the revised version of EMBOJ-2019-104058R and the accompanying rebuttal letter. I found the rebuttal to be largely argumentative, invoking a number of hypothetical concepts in an effort to account for what I had considered inconsistencies or unexplained observations. Thus, the authors argue for the existence of "trigger models", local nanodomains, transient events not captured by imaging, hypothetical association of calcineurin with TPC, etc. In essence, they argue that generation of NAADP -which is never demonstrated- generates local calcium nanodomains with the unique ability to activate a putative pool of calcineurin -that is never validated- to activate the dephosphorylation of dynamin, which is deemed essential for dynamin to function in phagocytosis but, somehow, dispensable for ongoing endocytosis. I would therefore suggest that you either rely on the opinions of the other reviewers or, better still, engage an additional independent reviewer with expertise in both phagocytosis and calcium.

Referee #2:

The authors addressed all my concerns and I strongly support publication of this manuscript in EMBO.

Referee #3:

The authors have satisfactorily addressed all of the suggestions we made in our previous review of the manuscript. I recommend the paper for publication in EMBO J without need for further revision.

We thank Referees 2 and 3 for their strong support and endorsement of our work for publication. We are sorry that referee 1 remains dissatisfied with our responses, but we are strongly of the opinion that most of the 'inconsistencies' that the referee raised have either been addressed with new data, or countered by reference to a long-standing literature.

We are puzzled that in their most recent remarks, the referee refers to 'transient events not captured by imaging' when we have indeed imaged them using a TPC-GECL.

To move this forward, we would like to make the following points:

- (a) We concede that we have not measured NAADP levels in response to phagocytosis.
- We have, however, investigated NAADP both pharmacologically and genetically (NAADP/AM, Ned-19, TPC-KO, TPC-GECL); importantly, we showed that **both *bona fide* NAADP responses (using NAADP/AM) and FcR phagocytosis Ca²⁺ signals share a common profile of sensitivity to Ned-19, Bafilomycin A1 and TPC-KO.**
 - We feel this is compelling evidence for NAADP, but without NAADP measurements, **would you prefer us to remove NAADP from the manuscript title?** We would prefer to retain reference to NAADP throughout the manuscript as a conceptual framework that is entirely consistent with our approaches.
- (b) As we stated in our response and in the revised manuscript, we only offer a lysosomal pool of calcineurin as a *hypothetical model*. Determining how the Ca²⁺ nanodomains are selectively decoded is a study in itself and forms the focus of our ongoing work. Therefore, we feel it is already 'toned down'. We have, however, added a sentence in the Discussion highlighting that this awaits experimental confirmation to reinforce this point.
- (c) We still do not understand the referee's point about dynamin, particularly when we have not even measured *endocytosis*. Regardless, the rate of membrane internalization at phagocytosis (large particles) will surely swamp that of endocytosis (small vesicles), and phagocytosis will likely require more dynamin activation than does any endocytosis.

In summary, we are willing to tone down the role of NAADP, but the other remarks we do not feel warrant change (either because they were only hypotheses, or because we still disagree with referee 1).

We hope that our amendments satisfy EMBO and make it now suitable for publication.

I am pleased to inform you that your manuscript has been accepted for publication in the EMBO Journal.